# A Unified Framework for Provably Efficient Algorithms to Estimate Shapley Values

**Tyler Chen**[*]  **Akshay Seshadri**[*]  **Mattia J. Villani**[*]  **Pradeep Niroula**
**Shouvanik Chakrabarti   Archan Ray   Pranav Deshpande   Romina Yalovetzky**
**Marco Pistoia   Niraj Kumar**[†]
Global Technology Applied Research, JPMorganChase, New York, NY 10001, USA

## Abstract

Shapley values have emerged as a critical tool for explaining which features impact the decisions made by machine learning models. However, computing exact Shapley values is difficult, generally requiring an exponential (in the feature dimension) number of model evaluations. To address this, many model-agnostic randomized estimators have been developed, the most influential and widely used being the KernelSHAP method (Lundberg & Lee, 2017). While related estimators such as unbiased KernelSHAP (Covert & Lee, 2021) and LeverageSHAP (Musco & Witter, 2025) are known to satisfy theoretical guarantees, bounds for KernelSHAP have remained elusive. We describe a broad and unified framework that encompasses KernelSHAP and related estimators constructed using both with and without replacement sampling strategies. We then prove strong non-asymptotic theoretical guarantees that apply to all estimators from our framework. This provides, to the best of our knowledge, the first theoretical guarantees for KernelSHAP and sheds further light on tradeoffs between existing estimators. Through comprehensive benchmarking on small and medium dimensional datasets for Decision-Tree models, we validate our approach against exact Shapley values, consistently achieving low mean squared error with modest sample sizes. Furthermore, we make specific implementation improvements to enable scalability of our methods to high-dimensional datasets. Our methods, tested on datasets such MNIST and CIFAR10, provide consistently better results compared to the KernelSHAP library.

## 1 Introduction

Explaining the prediction of a machine learning model is as important as building the model itself, since it helps determine whether the model can be trusted to give meaningful predictions when deployed in real world [RSG16]. Such explanations of black-box decisions are all the more important in sensitive applications, such as medicine, finance, and law [Bur+16].

In the quest of explaining models, recent line of research has focused on developing *local explanation* methods with the objective to identify the degree of influence of each feature that a specific data point has on the model prediction. These include Explanation vectors [Bae+10], LIME [RSG16], and Shapley values [ŠK14]. When local methods are expressed as additive feature attribution methods, i.e., the feature influence linearly adds up to provide the model prediction, [LL17] provided game theoretic results guaranteeing that Shapley values provide a unique solution to additive feature attribution. For these reasons, it has emerged as a front-runner model agnostic explanation tool.

---

[*]Equal contribution. Email: `{akshay.seshadri, tyler.chen, mattia.villani}@jpmchase.com`.
[†]Principal Investigator. Email: `niraj.x7.kumar@jpmchase.com`.

39th Conference on Neural Information Processing Systems (NeurIPS 2025).

Shapley values have found relevance in other machine learning applications too. They have been used in measuring the global sensitivity analysis where for instance they have been used to partition the coefficient of determination quantity in linear regression [SNS16]. More concretely, Shapley values offer a general approach of answering the following question: *given a model $f$ trained on data-points with $d$ features, and evaluated on a test sample $q \in \mathbb{R}^d$, how does each feature of $q$ locally influence the final model decision $f(q)$?*

Consider the value function $v : 2^{[d]} \to \mathbb{R}$, where $v(S)$ depends on the output of the model on a test sample $q$ using only the subset of features corresponding to the elements of the subset $S$ of $[d] = \{1, \ldots, d\}$. For instance, given a baseline $q^{\text{base}}$, we may define $v(S) = f(q^{(S)})$ where $q_j^{(S)} = q_j$ if $j \in S$ and $q_j^{\text{base}}$ otherwise.[3] The Shapley value $\phi_j^*$ corresponding to the $j$-th feature contribution is defined as

$$\phi_j^* = \sum_{S \subseteq [d] \setminus \{j\}} \frac{|S|!(d - |S| - 1)!}{d!} (v(S \cup \{j\}) - v(S)) \tag{1.1}$$

which is the aggregate of the marginal contribution of feature $j$ to the model prediction achievable by the modified mean of all the subsets $S$ that do not contain the feature $j$. As $d$ increases, an exact solution quickly cannot be computed and estimation techniques are required. However, as Shapley values are increasingly used to interpret the model behavior, the quality of the estimator is of the utmost importance: an unfaithful explanation may lead to incorrect model interventions, business decisions or court judgments whenever model assessment is involved.

## 1.1 Fast Approximate Estimators

In general, computing (1.1) requires evaluating $v(S)$ on each of the *exponentially many* subsets of $[d]$. Each evaluation of $v(S)$ is costly, with the exact cost depending on the way $v(S)$ is defined. While this cost can be reduced for certain types of simple models [LEL18], an appealing aspect of Shapley values is the potential for model-agnostic explanations.

To make Shapley values computationally tractable for arbitrary models, multiple randomized estimators have been proposed. Such methods aim to approximate the Shapley values, while using a sub-exponential number of value-function evaluations; see [CGT09; WF20; OL21; MCFH22; Zha+24]. Perhaps the most popular is a method called *KernelSHAP*, which is implemented in the widely used `SHAP` library [LL17]. KernelSHAP and related estimators are the focus of this paper.

**Approximate Shapley Value Estimation.** In what follows, the all ones vector and zero vector are $\mathbf{1}$ and $\mathbf{0}$ respectively, and the $j$-th standard basis vector is $e_j$. Given a vector $a$, $\|a\|$ denotes its Euclidean norm, while for a matrix $A$, $\|A\|$ denotes its spectral norm. The key observation [CGKR88] used by KernelSHAP and related estimators is that the Shapley values are the solution to a certain constrained least squares problem

$$\phi^* = \operatorname*{argmin}_{\substack{\phi \in \mathbb{R}^d \\ \mathbf{1}^\mathsf{T} \phi = v([d]) - v(\emptyset)}} \|Z' \phi - b\|^2, \tag{1.2}$$

where $Z' = \sqrt{W} Z$, $b = \sqrt{W} v$, and[4]

- $Z$ is a $(2^d - 2) \times d$ binary matrix: $Z_{S,j} = 1$ if $j \in S$ and $Z_{S,j} = 0$ if $j \notin S$
- $W$ is a $(2^d - 2) \times (2^d - 2)$ diagonal matrix: $W_{S,S} = k(S) = (d-1)/(\binom{d}{|S|}|S|(d - |S|))$
- $v$ is a $2^d - 2$ length vector: $v_S = v(S) - v(\emptyset)$.

As with the definition of the Shapley values (1.1), the regression formulation (1.2) requires the knowledge of $v(S)$ for each $2^d$ subsets of $[d]$. To get around this cost, KernelSHAP (randomly) subsamples and reweights the rows of (1.2), and then outputs the solution to the (much smaller) constrained regression problem. The sampling of the $S$-th row $Z_S$ is done proportional to the kernel

---

[3]There are other established ways to do this including replacing a fixed baseline with an expectation over suitable inputs or even training the model with only the features in $S$ present [CSWJ18; LL17]. The precise choice is not important for us, as the methods discussed in this paper work for any value function.

[4]The matrices are indexed by $S \subseteq 2^{[d]} \setminus \{[d], \emptyset\}$.

weight $k(S)$, a choice made based on the observation that the objective function $\|\boldsymbol{Z}'\boldsymbol{\phi} - \boldsymbol{b}\|^2$ can be written as an expectation $\mathbb{E}[(\boldsymbol{Z}_S\boldsymbol{\phi} - \boldsymbol{v}_S)^2]$ with respect to this sampling distribution, as explained in Appendix B.1. Other practical improvements such as *paired-sampling* and *sampling without replacement* are also included in the implementation of KernelSHAP in the SHAP library.

A large number of subsequent works have built on KernelSHAP [CL20; LL17; AJL21; Zha+24; MW25; Jet+21; KZ22; Fum+24; KTLM24]. Of particular relevance to the present work are *unbiased KernelSHAP* [CL20] and *LeverageSHAP* [MW25] which, to the best of our knowledge, are the only extensions of KernelSHAP with theoretical convergence guarantees. The method of [CL20] is an unbiased variant of KernelSHAP for which an asymptotic variance analysis is given. It was however observed that this method tends to underperform compared to the original KernelSHAP in practice. The method of [MW25] is a regression-based estimator and satisfies strong non-asymptotic theoretical guarantees. Numerical experiments suggest that it may outperform KernelSHAP in most settings.

**High-Dimensional Estimators.** Additionally, several works have specifically focused on the challenges of computing Shapley values for high-dimensional data [AJL21; CSWJ18; Jet+21; Fry+20; HZFS24; Zha+24]. These use parametric approaches to the computation of Shapley values; however, they require overhead model pretraining. Building on [Fry+20], [HZFS24] develop a method for high-dimensional SHAP estimation using latent features. [CSWJ18] propose a specific approach for data structured on graphs; such approaches avoid computing SHAP for large dimensions leveraging inductive biases. Recently, [Zha+24] propose SimSHAP, an unbiased alternative to [Jet+21]. Methods for large language models, such as [Kok+21] have recently been develop; however, no algorithm at present is tailored for high dimensional settings while providing provable guarantees on sample efficiency.

## 1.2 Our Contribution

In this work, we present a novel and unified framework to analyze Shapley value estimators. Using tools from randomized linear algebra, we prove non-asymptotic sample complexity guarantees on the efficient behavior of the estimators, including KernelSHAP [LL17] and LeverageSHAP [MW25]. Specifically, we identify three main contributions of the present work:

- **Unified Framework:** We present a unified framework which encompasses many existing randomized estimators for Shapley values, including the widely used KernelSHAP method. Our framework is derived by rewriting the standard constrained regression formulation of the Shapley values as either an ordinary linear regression problem or a matrix-vector multiplication.

- **Provable Guarantees:** We prove non-asymptotic sample-complexity bounds for estimators within our framework constructed via both with and without replacement sampling strategies. *This immediately gives, for the first time to our knowledge, theoretical guarantees for KernelSHAP.* Our theory also provides insight into the relative performance of estimators such as LeverageSHAP and KernelSHAP, as well as a novel estimator built with kernel re-weighted $\ell_2$ distribution.

- **Shapley Value Estimation for High Dimensional Inputs:** We make specific implementation improvements to Shapley value computation that allow our methods to scale beyond all other theoretically grounded methods. We test these on image datasets (MNIST and CIFAR10) with consistently better results compared to KernelSHAP library.

These advancements promote trust in the estimation of Shapley values, enabling their usage in safety-critical applications. In Section 2, we develop the unified framework: defining the estimators and distributions in Section 2.1 and Section 2.2 respectively, and providing our main result on sample complexity guarantees in Section 2.3. In Section 3, we perform an extensive experimental evaluation of the described estimators, comparing their performance in Section 3.1, and showcasing their effectiveness in higher dimensional settings in Section 3.2.

## 2 A Unified Framework for Provable Shapley Value Estimation

The main theoretical contribution of our paper is a unified framework through which many existing estimators for Shapley value estimation can be understood. We provide *non-asymptotic theoretical*

*guarantees* for all methods within our framework, *including that of the widely used KernelSHAP method.*

Towards this end, it is useful to reformulate (1.2) in terms of an ordinary linear regression or a matrix-vector multiplication problem involving a matrix with orthonormal columns. The key observation herein is that any vector $\phi \in \mathbb{R}^d$ satisfying the constraint $\mathbf{1}^\mathsf{T}\phi = v([d]) - v(\emptyset)$ can be decomposed as the sum of a vector proportional to $\mathbf{1}$ (with proportionality constant $(v([d]) - v(\emptyset))/d$) and a vector orthogonal to $\mathbf{1}$. By converting (1.2) to an unconstrained problem, we will be able to more easily understand how popular Shapley value estimators can be studied through the lens of randomized numerical linear algebra.

**Theorem 2.1.** *Let $\boldsymbol{Q}$ be any fixed $d \times (d-1)$ matrix whose columns form an orthonormal basis for the space of vectors orthogonal to the all-ones vector (i.e. $\boldsymbol{Q}^\mathsf{T}\boldsymbol{Q} = \boldsymbol{I}$, $\boldsymbol{Q}^\mathsf{T}\mathbf{1} = \mathbf{0}$). Given $\lambda \in \mathbb{R}$, define*

$$\boldsymbol{U} := \sqrt{\frac{d}{d-1}}\boldsymbol{Z}'\boldsymbol{Q}, \qquad \alpha := \frac{v([d]) - v(\emptyset)}{d}, \qquad \boldsymbol{b}_\lambda := \sqrt{\frac{d}{d-1}}(\boldsymbol{b} - \lambda\boldsymbol{Z}'\mathbf{1}).$$

*Then, $\boldsymbol{U}^\mathsf{T}\boldsymbol{U} = \boldsymbol{I}$ and*

$$\phi^* = \boldsymbol{Q}\operatorname*{argmin}_{\boldsymbol{x} \in \mathbb{R}^{d-1}}\|\boldsymbol{U}\boldsymbol{x} - \boldsymbol{b}_\lambda\|^2 + \alpha\mathbf{1} = \boldsymbol{Q}\boldsymbol{U}^\mathsf{T}\boldsymbol{b}_\lambda + \alpha\mathbf{1}.$$

A similar formulation of the Shapley values in terms of unconstrained regression appears in [MW25]. Theorem 2.1, which is proved in Appendix A.3, goes beyond that of [MW25] in two key ways. First, we observe that by solving the unconstrained problem explicitly, we obtain the solution as the product of a matrix $\boldsymbol{Q}\boldsymbol{U}^\mathsf{T}$ and vector $\boldsymbol{b}_\lambda$. Second, we make the observation that there is complete freedom in the choice of $\lambda \in \mathbb{R}$. Together, these advancements allow us to develop a unifying framework for providing provable guarantees for a broad class of randomized estimators which encompasses many existing estimators [CL20; LL17; AJL21; Zha+24; MW25, etc.].

## 2.1 Randomized Estimators Within our Framework

We frame our exposition in the context of *randomized sketching*, a powerful technique which has been studied for decades in randomized numerical linear algebra [Woo+14; MT20].

In the context of Shapley value estimation, a sketching matrix is an $m \times (2^d - 2)$ matrix $\boldsymbol{S}$ where each row has exactly one nonzero entry and $\mathbb{E}[\boldsymbol{S}^\mathsf{T}\boldsymbol{S}] = \boldsymbol{I}$. We leave the exact choice of the distribution of $\boldsymbol{S}$ general, but discuss several natural choices in Section 2.2. Regardless of the distribution, since each of the $m$ rows of $\boldsymbol{S}$ has exactly one nonzero entry, computing $\boldsymbol{S}\boldsymbol{b}$ requires at most $m$ evaluations of $v(S)$. Thus, estimators which make use of $\boldsymbol{S}\boldsymbol{b}$ can be substantially more efficient to compute when $m \ll 2^d$.

Using the sketch $\boldsymbol{S}\boldsymbol{b}_\lambda$ (which can easily be computed from $\boldsymbol{S}\boldsymbol{b}$) in the formulations in Theorem 2.1 yields estimators based on *sketched regression* or on *approximate matrix-vector multiplication*.

1. **Sketched Regression:** Methods such as KernelSHAP[5] and LeverageSHAP can be viewed as sketched versions of the regression formulation of the Shapley values:

$$\phi_\lambda^\mathsf{R} := \boldsymbol{Q}\operatorname*{argmin}_{\boldsymbol{x} \in \mathbb{R}^{d-1}}\|\boldsymbol{S}(\boldsymbol{U}\boldsymbol{x} - \boldsymbol{b}_\lambda)\|^2 + \alpha\mathbf{1}.$$

   Given the sketching matrix $\boldsymbol{S}$, this regression (or least squares) estimator can be computed in $O(md^2 + mT_v)$ time, where $T_v$ is the time to evaluate an entry of $\boldsymbol{b}$.

2. **Approximate Matrix-Vector Multiplication:** Instead of approximating the regression problem, methods such as unbiased KernelSHAP approximate the closed-form solution $\boldsymbol{U}^\mathsf{T}\boldsymbol{b}_\lambda$ directly:

$$\phi_\lambda^\mathsf{M} := \boldsymbol{Q}\boldsymbol{U}^\mathsf{T}\boldsymbol{S}^\mathsf{T}\boldsymbol{S}\boldsymbol{b}_\lambda + \alpha\mathbf{1}.$$

   This estimator is *unbiased* (provided $\mathbb{E}[\boldsymbol{S}^\mathsf{T}\boldsymbol{S}] = \boldsymbol{I}$) and, given the sketching matrix $\boldsymbol{S}$, can be computed in $O(md + mT_v)$ time, where $T_v$ is the time to compute $v(S)$.

---

[5]At first glance it is not obvious that KernelSHAP, which solves an approximation to the *constrained* problem (1.2), can be expressed this way. However, a careful computation (see Appendix B.1) reveals that the KernelSHAP estimator is indeed a special case of the general regression estimator (with $\lambda = \alpha$).

We provide proofs that the estimators from [LL17; CL20; MW25] fit into our framework in Appendix B. Past works, especially [CL20], have used a Lagrangian framework to obtained closed-form solutions to their randomized estimators. While this is mathematically equivalent to our change of variable approach, as described in Appendix C, the expressions, which involve ratios of correlated random variables, are seemingly harder to analyze directly in the Lagrangian framework leading to previous difficulties in providing proofs of KernelSHAP [CL20].

## 2.2 Sampling Schemes for Sketching Matrix

The choice of $S$ plays a critical role in both the regression and matrix-vector multiplication estimators–which $m$ entries of $b$ are observed impacts what we learn about the Shapley values. However, model-agnostic estimators cannot make strong assumptions about the structure of $b$. The relative importance of the $i$-entry of $b$ can be encoded in a probability distribution $\mathcal{P}$ over subsets $S \subset 2^{[d]} \setminus \{[d], \emptyset\}$. This distribution is subsequently used to generate $S$ and hence sample the entries of $b$.[6] In the context of Shapley value estimation, it is common to use further optimizations such as *paired sampling* and *sampling without replacement*, which we explore empirically in Section 3.

Since the values of $b$ are costly to observe and are highly dependent on the given model, it is natural to choose the $\mathcal{P}$ based on $U$. Two popular choices are sampling based on the kernel weights (as done in KernelSHAP), and sampling based on the leverage scores of $U$ (as done in LeverageSHAP). We therefore analyze these distributions in our study, along with another distribution that interpolates between these two.

1. **Kernel Weight Sampling:** The KernelSHAP and unbiased KernelSHAP methods use $p_S \propto k(S)$. This is a heuristic choice based on the fact that expressions like $(Z')^\mathsf{T} Z'$ and $(Z')^\mathsf{T} b$ can be naturally written as the expectation of certain random variables with respect to this sampling distribution.

2. **Leverage Score / $\ell_2$-squared Sampling:** The LeverageSHAP method chooses sampling probabilities proportional to the *statistical leverage scores* of $U$. Since $U$ has orthonormal columns, the leverage score of the $S$th row of $U$ coincides with the squared row-norm $\|u_S\|^2$, which is widely used in the quantum-inspired algorithms framework [Tan19]. Leverage score sampling for sketched regression satisfies strong theoretical guarantees, which [MW25] use to prove guarantees about the LeverageSHAP estimator.

3. **Modified $\ell_2$ Sampling:** The modified row-norm sampling scheme is obtained by taking the usual geometric mean of kernel weights and leverage scores. The theoretical bounds we derive for these weights are never worse than the bounds for $\ell_2$-squared sampling in the worst-case (up to constant factors), but can be up to a factor of $\sqrt{d}$ better in some cases.

All the above distributions can be thought of special cases of a family of distributions that interpolate between kernel weights and leverage scores. Specifically, given a parameter $\tau \in [0, 1]$, we can consider the distribution

$$p_S^\tau \propto (k(S))^\tau (\|u_S\|^2)^{1-\tau}, \tag{2.1}$$

which is the weighted geometric mean of the kernel weights and the leverage scores (see (A.66) for the full expression). $\tau = 1$ gives kernel weight distribution, $\tau = 0$ gives leverage score sampling, while $\tau = 1/2$ gives modified $\ell_2$ sampling.

---

[6]The approaches we consider only take into account the relative importance of individual rows. Other approaches (e.g. based on Determinantal Point Processes/volume sampling) take into account the relative importance of entire sets of rows. This results in stronger theoretical guarantees for general regression problems, but such distributions are harder to sample from [DM21].

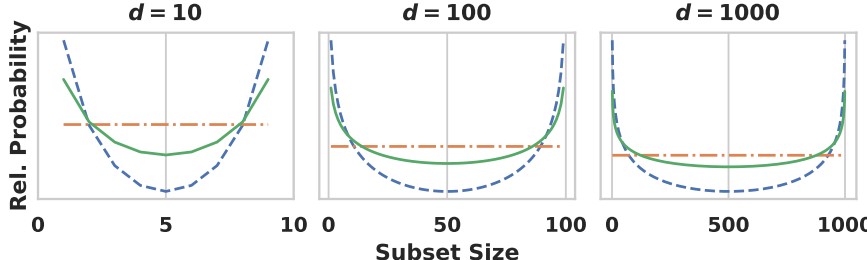

**Figure 1:** Comparison of the sampling probabilities described in Section 2.2. Kernel Weights (dashed), Leverage scores (dash-dot), and our proposed modified $\ell_2$-weights (solid), which are the geometric mean of the Kernel Weights and Leverage scores.

**Note:** In the remainder of the study, we interchangeably use the term (modified) row-norm sampling with (modified) $\ell_2$-norm sampling, and leverage score sampling with $\ell_2$-squared sampling.

## 2.3 Theoretical Guarantees of Shapley Value Approximation

We now provide bounds on the sketching dimension $m$ required to ensure a Shapley value estimator $\widehat{\phi} \in \{\phi_\lambda^{\mathrm{R}}, \phi_\lambda^{\mathrm{M}}\}$ satisfies a guarantee

$$\mathbb{P}\left[\left\|\phi^* - \widehat{\phi}\right\| < \varepsilon\right] > 1 - \delta, \tag{2.2}$$

for some accuracy parameter $\varepsilon > 0$ and failure probability parameter $\delta \in (0, 1)$.

While there are a number of ways to construct a $m \times 2^d - 2$ sketching matrix $S$ from sampling probabilities $\mathcal{P} = (p_S)_{S \subset 2^{[d]} \setminus \{[d], \emptyset\}}$. We analyze two common choices:

1. **With Replacement:** Each of the $m$ rows of $S$ are sampled independently. For a given row, a single entry is selected to be nonzero according to $\mathcal{P}$. The value of this nonzero entry is $1/\sqrt{mp_S}$, where $S$ is the index of the nonzero row; see Appendix A.2.

2. **Without Replacement:** For each subset $S$, we flip a coin that returns heads with probability $q_S$. If the coin is heads, we add a row to $S$, where the $S$-th entry of the row is nonzero and takes value $1/\sqrt{q_S}$. The probabilities $q_S$ are chosen based on the $\mathcal{P}$ so that, the dimension of the sketching matrix is equal, on average, to some target value $m$; see Appendix A.5.

To reduce the notational burden, we parameterize our bounds in terms of

$$\eta := \max_{S \in 2^{[d]} \setminus \{[d], \emptyset\}} \frac{\|u_S\|^2}{p_S}, \qquad \gamma(z) := \sum_{S \in 2^{[d]} \setminus \{[d], \emptyset\}} \frac{\|u_S\|^2}{p_S}(z_S)^2, \quad z \in \mathbb{R}^{2^d - 2}. \tag{2.3}$$

Our main theoretical result, which we prove in Appendix A.3 using techniques from randomized numerical linear algebra [Woo+14; Tro15; MT20], is the following:

**Theorem 2.2.** *Define $P_U := (I - UU^\mathsf{T})$, and fix $\lambda \in \mathbb{R}$. Let $m$ denote the sample complexity in the sampling with replacement scenario and the average sample complexity in the sampling without replacement scenario. Then, for the regression estimator,*

$$m = O\left(\frac{\gamma(P_U b_\lambda)}{\delta \varepsilon^2} + \eta \log\left(\frac{d}{\delta}\right)\right) \quad \text{guarantees} \quad \mathbb{P}\left[\|\phi^* - \phi_\lambda^{\mathrm{R}}\| < \varepsilon\right] > 1 - \delta,$$

*and for the matrix-vector multiplication estimator,*

$$m = O\left(\frac{\gamma(b_\lambda)}{\delta \varepsilon^2}\right) \quad \text{guarantees} \quad \mathbb{P}\left[\|\phi^* - \phi_\lambda^{\mathrm{M}}\| < \varepsilon\right] > 1 - \delta.$$

A direct computation reveals that $\gamma(P_U b_\lambda) \leq \eta \|P_U b_\lambda\|^2 \leq \eta \|b_\lambda\|^2$, where the first inequality is by the definition of $\eta$ and second inequality is due to the fact that $P_U$ is the orthogonal projector onto the column-span of $U$. However, for a particular $b_\lambda$, each of these inequalities may not be sharp.

In Table 1, we provide more refined bounds for the kernel weight, leverage score, and modified row-norm sampling probabilities from Section 2.2. More precise bounds are stated and derived in Appendix A.4, and we also give bounds for the family of distributions defined in (2.1) in Remark A.11. Importantly, the bounds for modified row-norm sampling are no worse than leverage scores, but can be up to a factor of $\sqrt{d}$ better in some cases. Furthermore, up to log factors, the bounds for kernel weights are no worse than both leverage scores and modified row-norm sampling, but can be a factor of $d/\log(d)$ or $\sqrt{d}/\log(d)$ better than leverage scores and modified row-norm sampling in some cases, respectively. These observations are formalized in Corollary A.9, and we construct an adversarial model demonstrating such an advantage in the sample complexity bounds in Appendix E. Intuitively, kernel weights and modified row-norm sampling place a larger importance on subsets of small/large size, as seen from Fig. 1. As a result, for models where the entries of the vector $\boldsymbol{b}_\lambda$ or $\boldsymbol{P}_U \boldsymbol{b}_\lambda$ are concentrated around subsets of small/large size, kernel weights or modified row-norm sampling would perform better than leverage score sampling, which is the key observation we use for constructing the adversarial model in Appendix E. It remains to be seen whether kernel weights or modified row-norm sampling scheme provides a sample complexity advantage over leverage scores for models used in practice (such as neural networks), and we leave this as an open question for future research.

| | $\gamma(\boldsymbol{P}_U \boldsymbol{b}_\lambda)$ | $\gamma(\boldsymbol{b}_\lambda)$ | $\eta$ |
|---|---|---|---|
| Kernel Weights | $d\log(d)\,\|\boldsymbol{H}\boldsymbol{P}_U\boldsymbol{b}_\lambda\|^2$ | $d\log(d)\,\|\boldsymbol{H}\boldsymbol{b}_\lambda\|^2$ | $d\log(d)$ |
| Leverage Scores | $d\,\|\boldsymbol{P}_U\boldsymbol{b}_\lambda\|^2$ | $d\,\|\boldsymbol{b}_\lambda\|^2$ | $d$ |
| Modified row-norms | $d\,\|\sqrt{\boldsymbol{H}}\boldsymbol{P}_U\boldsymbol{b}_\lambda\|^2$ | $d\,\|\sqrt{\boldsymbol{H}}\boldsymbol{b}_\lambda\|^2$ | $d$ |

**Table 1:** Bounds (big-$\Theta$) on parameters in Theorem 2.2 for the sampling weights from Section 2.2, derived in Corollary A.10. $\boldsymbol{H}$ is a diagonal matrix defined in Corollary A.10 satisfying $\lambda_{\min}(\boldsymbol{H}) = \Theta(1/\sqrt{d})$ and $\lambda_{\max}(\boldsymbol{H}) = \Theta(1)$, so that $\|\boldsymbol{H}\boldsymbol{x}\|/\|\boldsymbol{x}\| \in [\Theta(1/\sqrt{d}), \Theta(1)]$. Hence, the bounds for kernel sampling are within a $\log(d)$ factor of leverage score sampling in the worst case, but can be better by a factor $d/\log(d)$ in some cases. On the other hand, the bounds for modified $\ell_2$ sampling are never worse than leverage score sampling, but can be better by a factor of $\sqrt{d}$ in some cases (see Corollary A.9).

## 3 Experiments

Based on our framework, Appendix F describes the pseudo-code of the randomized estimators based on sampling with-replacement Algorithm 1 and without-replacement Algorithm 2. We evaluate these estimators across a range of synthetic and real world settings. Of primary interest is the mean squared error distance from the true Shapley value (normalized: $\mathtt{mse} = \mathbb{E}[\|\boldsymbol{\phi}^* - \widehat{\boldsymbol{\phi}}\|^2]/\|\boldsymbol{\phi}\|^2$); we explore the convergence of these estimators to the true Shapley Values. We set out to find the best strategy, but our findings reveal that each method has its own merits across different scenarios. A summary of the experiments is provided here, with details deferred to the following sections.

In the experiments that follow, [MW25] has been re-implemented to (a) allow the methods to be computed in high dimensions efficiently, and (b) to ensure a fair comparison between regression and matrix-vector multiplication method by fixing a single $\boldsymbol{SZ}$ for both estimators. We include results from our implementation of KernelSHAP (regression + kernel weights) as well as the implementation of KernelSHAP from the shap library. This particular implementation includes several additional heuristic optimizations.

We run experiments on eight popular tabular datasets from the shap library (up to $d = 101$) and two image datasets (MNIST $d = 784$, and CIFAR-10 $d = 3072$), details on each dataset are in Appendix G.2. In each dataset, we train an XG-Boost model [CG16] to compute the exact Shapley values using TreeExplainer class in shap [Lun+20]. We report a summary of the experimental findings while leaving detailed experiments to Appendix G.

Following [CL20; MW25], we run our experiments using paired sampling, a simple modification of the estimation procedure, which has been observed to improve empirical performance. In paired

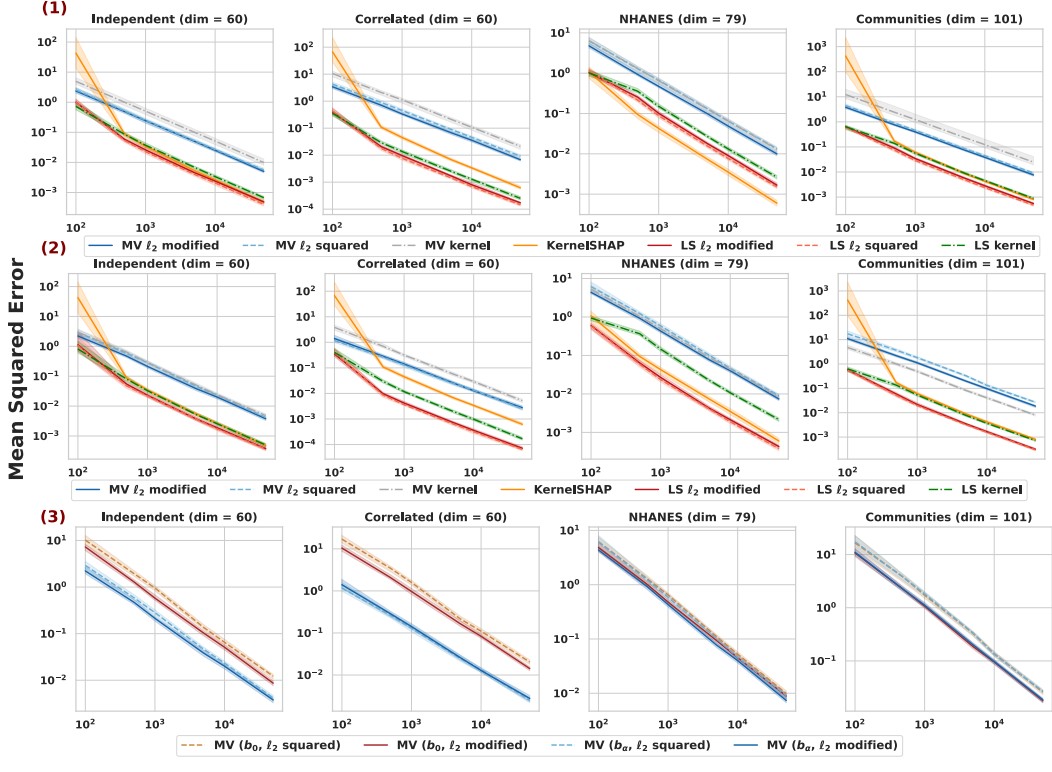

**Figure 2:** Comparison of performance across different estimators. In (1, top row) estimators use *with replacement* sampling strategies. In (2,3, central and bottom row) $SZ$ is sampled without replacement. In legends, **MV** refers to matrix-vector multiplication estimator and **LS** to regression (least squares) estimator. Dimensions of each datasets are reported with the titles.

sampling, when an index $S \subset 2^{[d]} \setminus \{[d], \emptyset\}$ is selected, the compliment $S^c = [d] \setminus S$ is also selected. Paired sampling is also used by default in implementation of KernelSHAP from the `shap` library.

We run our experiments on an AMD EPYC 7R13 processor with 48 cores per socket, 96 CPUs, and 183GB RAM.

## 3.1 Comparisons of Estimators

For each dataset, we choose the first data points of the train and the test sets, according to an $80/20$ split, as baseline, and query points for our Shapley estimators respectively. We choose $m = 10^3, 10^4/2, 10^4, ..., 10^6/2$ for larger datasets ($d > 12$) and pick specific values of $m$ for smaller datasets. We run the experiments on random seeds $0, .., 99$ (`numpy` and Python's `random`) for replicability of results. Exact Shapley values are computed with `TreeExplainer` on the same baseline; `KernelExplainer` is run without $\ell_1$ regularization. XG-boost regressors and classifiers are fit with 100 estimators and a maximum tree depth of 10. We highlight key observations in Fig. 2, where we plot median normalized mean squared errors for 100 random seeds, alongside interquantile ranges. Except when specified otherwise, we use $\lambda = \alpha$. Detailed results, variations and tables are in Appendix H.1. Our findings are as follows.

- **Matrix-Vector Multiplication vs. Regression Estimator.** We showcase the effectiveness of each Shapley value estimator in practice, reporting a comparison between the best performing distribution in Fig. 2 (1, with replacement; 2 without replacement). The clearest separations across methods appears in the comparison between matrix-vector multiplication and regression estimators. We find that *regression estimator tends to perform better* than matrix-vector multiplication estimator. KernelSHAP is generally positioned between these methods in the ranking. This is highlighted in Fig. 2.

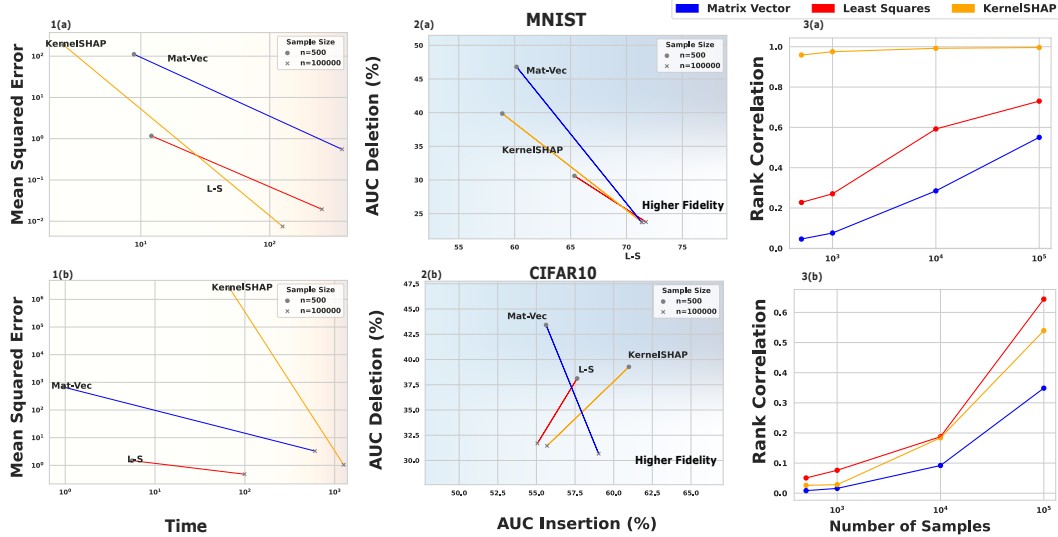

**Figure 3:** Comparison of estimators in image datasets: MNIST (top row) and CIFAR (bottom row). In the first column, (1, left column) performance of estimators is measured with mean squared error (normalized) from true Shapley value and time (in seconds). (2, center column) Area under the curve (AUC) calculation for insertion (x-axis) and deletion curves (y-axis) have been provided, computed on the top 100 features; reported as percentage under the curve. (3, right column) Spearman rank correlation for increasing number of samples.

- **Comparison Across Distributions.** Among the three sampling schemes evaluated, our results in Fig. 2 (1,2) indicate that the $\ell_2$-squared method outperforms modified $\ell_2$ marginally, while outperforming kernel distribution more significantly in both regression and matrix-vector multiplication approximations. As discussed in Section 2.2, the choice of sampling distribution affects the performance of the estimator; with the best choice being problem dependent. In Appendix E, we design a synthetic experiment where estimators based on modified $\ell_2$ and kernel distributions significantly outperform $\ell_2$-squared distribution based estimator.

- **With and Without Replacement Sampling Strategies.** Sampling strategies (with and without replacement) perform similarly for the matrix-vector multiplication estimator. For the regression estimator, sampling with replacement outperforms sampling without replacement on some datasets. However, sampling with replacement strategy is poorly suited for settings where $m > 2^d$, since it will perform worse than brute force computation of Shapley values.

- **Comparing $\lambda=0$ versus $\lambda=\alpha$ for the Matrix-Vector Multiplication Estimator.** We compare the performance of matrix-vector multiplication estimator using $\lambda = 0$ and $\lambda = \alpha$ (i.e., $\boldsymbol{b}_0$ vs $\boldsymbol{b}_\alpha$) in Fig. 2 (3). As noted in Appendix B, the unbiased KernelSHAP method of [CL20] uses $\lambda = 0$, while the other methods we have explored use $\lambda = \alpha$. We find that using $\lambda = \alpha$ in the estimator leads to better performance.

### 3.2 Provably Efficient Methods in High-Dimensions

For high dimensional datasets, we aim to compare the estimators across faithfulness measures, as well as mean squared error. We compute Shapley values on the first 10 data points from the test sets, using the first data point of the training set, using 80/20 splits. As before, we train a decision tree in order to be able to compare with exact Shapley value computed from the `TreeExplainer` class. For each method, we compare average normalized mean squared error across test points, computational costs and faithfulness of the explanations. Mean squared error is juxtaposed with time (in seconds) in Fig. 3 1(a,b) to emphasize computational tradeoffs between methods. Faithfulness via both area under the curve (AUC) of insertion and deletion curves in Fig. 3 2(a,b), and Spearman rank correlation between exact and estimated Shapley values (as reported in Fig. 3, 3(a,b)). Detailed experimental results with errors can be found in Appendix H.2.

**Algorithmic Innovations.** Approximating Shapley values in high dimensional problems is a challenge. There are two computational bottlenecks in [MW25]: (a) for distributions beyond $\ell_2$-squared, combinatorial terms $\binom{d}{k}$ will cause overflow/underflow for sufficiently large $d$ and middle $k$ (i.e., $k \sim d/2$), and (b) even if we are able to compute the binomial term, [MW25] bucket sampling procedure requires binomial sampling from a distribution with support of size $\binom{d}{k}$, which can be large. In our Algorithm 2, we overcome both issues for all distributions by (a) avoiding the computation of the combinatorial terms in the probability distributions and weights, and (b) using Poisson approximation of large binomials to avoid the large support problem. This allows an analysis of our estimators on CIFAR10.

**Estimator Performance.** In Fig. 3 part 1(a-b), experiments confirm that regression estimators generally requires less time and lead to better approximations for fixed number of samples compared to matrix-vector multiplication estimator. Indeed, this discrepancy is accentuated as the dimension size increases. The regression estimator produces accurate estimates even when the number of samples is small, improving on all other estimators.

**Faithfulness.** In Fig. 3 part 2(a-b) and 3(a-b); after 100k samples, we find that for MNIST, all estimators have similar fidelity, but KernelSHAP has very high rank correlation. This may be due to the fact that KernelSHAP first samples from buckets of size $1$ and $d$, a difference which may be beneficial in this setting. For CIFAR-10, there have been significant increases in rank correlation, showcasing the effectiveness of the estimators. In all settings, we find increased fidelity especially as the dimensionality of the problem increases. We note this could be problem dependent. We report AUC curves in Appendix H.3.

## 4 Discussion

We have provided a theoretical grounding for the use of randomized estimators in the context of Shapley value computation. We have achieved this by means of sample-efficient convergence guarantees for a broad family of estimators, including the popular estimator KernelSHAP and the recently introduced LeverageSHAP. Responsible use of explainable-AI methods involves an understanding of how estimators scale as sample complexity is increased, especially when computing the exact ground truth Shapley values are not computationally feasible. This work on unified framework provides a definitive step in this direction.

**Limitations.** Computing accurate Shapley values remains a challenge. As with past work, the theoretical bounds we derive for Shapley value estimators depend on quantities involving $\boldsymbol{b}_\lambda$ (e.g. $\|\boldsymbol{b}_\lambda\|$) which cannot be computed efficiently. As such, they cannot be instantiated by the user. Below, we give a prescription on how this limitation can be mitigated in practice, but leave a thorough study for future research. Also note that there are several approaches to sampling without replacement and the present work does not provide prescriptions on which to use; this is left to future work.

**Practical Prescription.** As our analysis reveals, the estimators converge in a predictable way with the number of samples ($m$) to the true Shapley value, at the rate of $\sim 1/\sqrt{m}$. Therefore, we can use the estimate from a larger value of $m$ to approximate the error at some $m_0 \ll m$. As long as $m \gg m_0$, the estimate using $m$ samples is a good proxy for the true Shapley values, relative to the error of the estimate using $m_0$ samples. We find that this method, while heuristic, gives a good estimate of the error in practice.

**Future Work.** This work promotes trust in the estimation of Shapley values, promoting a responsible use of the estimators in the explainable-AI community. Our theoretical contributions of a unified framework pave the way for development of tailored estimators depending on the observed entries of $\boldsymbol{b}_\lambda$, which can be used to adapt the sampling distribution accordingly. Developing such adaptive estimators, as well as their theoretical analysis, is left as an interesting direction for future research.

## Acknowledgements

The authors thank Rob Otter and Shaohan Hu for their support and valuable feedback on this project. We also acknowledge our colleagues at the Global Technology Applied Research Center of JPMorganChase, especially Sriram Yechan Gunja and Rajagopal Ganesan, for helpful discussions.

We would also like to thank R. Teal Witter and Christopher Musco for providing us the code for LeverageSHAP [MW25].

## Disclaimer

This paper was prepared for informational purposes by the Global Technology Applied Research center of JPMorgan Chase & Co. This paper is not a merchandisable/sellable product of the Research Department of JPMorgan Chase & Co. or its affiliates. Neither JPMorgan Chase & Co. nor any of its affiliates makes any explicit or implied representation or warranty and none of them accept any liability in connection with this paper, including, without limitation, with respect to the completeness, accuracy, or reliability of the information contained herein and the potential legal, compliance, tax, or accounting effects thereof. This document is not intended as investment research or investment advice, or as a recommendation, offer, or solicitation for the purchase or sale of any security, financial instrument, financial product or service, or to be used in any way for evaluating the merits of participating in any transaction.

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

# Supplementary Material for
## "A Unified Framework for Provably Efficient Algorithms to Estimate Shapley Values"

## A  Proofs of the Main Theoretical Results

### A.1  Notation

It will be convenient to switch from indexing rows by sets $S \subset 2^{[d]} \setminus \{[d], \emptyset\}$ and instead index by integers $i \in [2^d - 2]$. Given $d \in \mathbb{N}$, we fix an ordering of the subsets of $[d]$ according to the size of the subset. Subsets of the same size are ordered in any fixed way (since the sampling probabilities of all distributions we consider only depends on the subset size). We then identify $i \in [2^d - 2]$ to integers $(h, l)$ satisfying $h \in [d-1]$, $l \in [\binom{d}{h}]$ by

$$i = \sum_{j=1}^{h-1} \binom{d}{j} + l. \tag{A.1}$$

Unless mentioned otherwise, $e_1, \ldots, e_q$ are the standard basis vectors for $\mathbb{R}^q$. $\mathbf{0}$ and $\mathbf{1}$ are the vectors of all zeros and ones, respectively, while $I$ is the identity matrix. $\|\cdot\|$ denotes the Euclidean norm for vectors, while the spectral norm for matrices. $\|\cdot\|_\mathsf{F}$ denotes the Frobenius norm. Given a matrix $A$, $A^+$ denotes its Moore-Penrose pseudoinverse. Finally, given a matrix $U$ with orthonormal columns, we write $P_U = I - UU^\mathsf{T}$.

### A.2  Proofs from Randomized Numerical Linear Algebra

In this section, we provide bounds for approximate matrix-vector multiplication and sketched regression (least squares) as defined in Section 2.1. Our proofs follow standard techniques in randomized numerical linear algebra [Woo+14; MT20; Tro15], and are included to illustrate core concepts which may provide a useful starting point for proving theoretical guarantees for more complicated sketching distributions for Shapley value estimation. For simplicity, we analyze the simpler case that $S$ has independent rows; i.e. that

$$S = \frac{1}{\sqrt{m}} \sum_{j=1}^{m} \frac{1}{\sqrt{p_{I_j}}} e_j (e_{I_j})^\mathsf{T}, \tag{A.2}$$

where $I_1, \ldots, I_m$ are iid copies of a random variable $I$ for which $\mathbb{P}[I = k] = p_k$, $k \in [r]$ for some fixed $r \in \mathbb{N}$. Note that $\mathbb{E}[S^\mathsf{T} S] = I$.

**Proof Sketch:** Prior to diving into the technical details, we present a high-level overview of the strategy used in deriving sample complexity bounds for matrix-vector multiplication and regression estimators.

Given a matrix $U \in \mathbb{R}^{r \times q}$ satisfying $U^\mathsf{T} U = I$, fixed vectors $z, b \in \mathbb{R}^r$, our goal is to estimate $U^\mathsf{T} z$ and $y^* = \operatorname{argmin}_y \|Uy - b\|^2$.

1. **Approximate Matrix-Vector Multiplication:** Observe that $U^\mathsf{T} S^\mathsf{T} S z$ is an unbiased estimator of $U^\mathsf{T} z$. By computing the variance of this estimator and using Markov's inequality, we obtain bounds on the sample complexity of estimating $U^\mathsf{T} z$ to a given error (in $\ell_2$ norm) and confidence level (see Theorem A.1). Note, in particular, that the term $\gamma(z)$ (see (2.3)) appearing in the sample complexity is related to the variance of the estimator.

2. **Sketched Regression:** The sketched regression estimator is given by $\widehat{y} = \operatorname{argmin}_y \|S(Uy - b)\|^2$. To derive sample complexity bounds for estimating $y^*$ using $\widehat{y}$, we use two main observations. (I) Since $U^\mathsf{T} S^T S U$ is an unbiased estimator of $I$, we can compute the number of samples (using matrix Bernstein's inequality, Imported Theorem A.3)

to ensure that $\|\boldsymbol{U}^\mathsf{T}\boldsymbol{S}^\mathsf{T}\boldsymbol{S}\boldsymbol{U} - \boldsymbol{I}\| \leq 1/2$ holds with sufficiently high probability (see Theorem A.4). Straightforward algebra then gives $\|\widehat{\boldsymbol{y}} - \boldsymbol{y}^*\| \leq 2\|\boldsymbol{U}^\mathsf{T}\boldsymbol{S}^\mathsf{T}\boldsymbol{S}\boldsymbol{U}(\widehat{\boldsymbol{y}} - \boldsymbol{y}^*)\|$ with high probability. (II) Since $\boldsymbol{S}(\boldsymbol{U}\widehat{\boldsymbol{y}} - \boldsymbol{b})$ lies in the orthogonal complement of $\boldsymbol{SU}$, we have $\boldsymbol{U}^\mathsf{T}\boldsymbol{S}^\mathsf{T}(\boldsymbol{SU}\widehat{\boldsymbol{y}} - \boldsymbol{Sb}) = 0$. For a similar reason, we also have $\boldsymbol{U}^\mathsf{T}(\boldsymbol{U}\boldsymbol{y}^* - \boldsymbol{b}) = 0$. It follows that $\boldsymbol{U}^\mathsf{T}\boldsymbol{S}^\mathsf{T}\boldsymbol{SU}(\boldsymbol{y}^* - \widehat{\boldsymbol{y}}) = \boldsymbol{U}^\mathsf{T}\boldsymbol{S}^\mathsf{T}\boldsymbol{S}(\boldsymbol{U}\boldsymbol{y}^* - \boldsymbol{b})$, which is just a sketched matrix-vector multiplication estimator for $\boldsymbol{U}^\mathsf{T}(\boldsymbol{U}\boldsymbol{y}^* - \boldsymbol{b}) = 0$. Consequently, we can use Theorem A.1 to compute the sample complexity for bounding the error $\|\boldsymbol{U}^\mathsf{T}\boldsymbol{S}^\mathsf{T}\boldsymbol{SU}(\widehat{\boldsymbol{y}} - \boldsymbol{y}^*)\|$ with high probability. In particular, since $\boldsymbol{U}^\mathsf{T}(\boldsymbol{U}\boldsymbol{y}^* - \boldsymbol{b}) = (\boldsymbol{I} - \boldsymbol{U}\boldsymbol{U}^\mathsf{T})\boldsymbol{b} = \boldsymbol{P_U}\boldsymbol{b}$, this explains why we have $\gamma(\boldsymbol{P_U}\boldsymbol{b})$ instead of $\gamma(\boldsymbol{b})$ in the sample complexity for the sketched regression estimator in Theorem A.5.

Proofs for sampling without replacement, which follow a similar strategy, are described in Appendix A.5.

### A.2.1 Approximate Matrix-Vector Multiplication

We begin with a simple bound on approximate matrix-vector multiplication. This bound immediately gives provable guarantees for the Shapley estimator $\phi^\mathrm{M}$ defined in Section 2.1.

**Theorem A.1** (Matrix-Vector multiplication). *Let $\boldsymbol{U} \in \mathbb{R}^{r \times q}$ and $\boldsymbol{z} \in \mathbb{R}^r$. Let $\boldsymbol{S}$ be a $m \times r$ sketching matrix with iid rows drawn according to probability $\mathcal{P}$. Then, if*

$$m \geq \left(\gamma(\boldsymbol{z}) - \|\boldsymbol{U}^\mathsf{T}\boldsymbol{z}\|^2\right)\frac{1}{\delta\varepsilon^2}$$

*it holds that*

$$\mathbb{P}\left[\|\boldsymbol{U}^\mathsf{T}\boldsymbol{S}^\mathsf{T}\boldsymbol{S}\boldsymbol{z} - \boldsymbol{U}^\mathsf{T}\boldsymbol{z}\| \leq \varepsilon\right] \geq 1 - \delta.$$

*Proof.* Let $I_1, \ldots, I_m$ denote $m$ iid random variables that sample indices from $[r]$ according to probability $\mathcal{P}$. Let $\boldsymbol{u}_1, \ldots, \boldsymbol{u}_r$ be the columns of $\boldsymbol{U}^\mathsf{T}$, and define $\boldsymbol{X}_j = \boldsymbol{u}_{I_j}z_{I_j}/p_{I_j}$ for $j \in [m]$. Then, $\boldsymbol{X}_j$ are iid $d$-dimensional random vectors. It can be verified that $\mathbb{E}[\boldsymbol{X}_j] = \boldsymbol{U}^\mathsf{T}\boldsymbol{z}$ for all $j \in [m]$. Next, we calculate the variance of the random vector $\boldsymbol{X}_j$ for $j \in [m]$. Using (2.3), observe that

$$\mathbb{E}\left[\|\boldsymbol{X}_j\|^2\right] = \sum_{i=1}^r p_i \frac{\|\boldsymbol{u}_i\|^2}{p_i^2}z_i^2 = \gamma(\boldsymbol{z}), \tag{A.3}$$

so that $\mathrm{var}(\boldsymbol{X}_j) = \mathbb{E}[\|\boldsymbol{X}_j - \mathbb{E}[\boldsymbol{X}_j]\|^2] = \gamma(\boldsymbol{z}) - \|\boldsymbol{U}^\mathsf{T}\boldsymbol{z}\|^2$ for all $j \in [m]$. Since $\boldsymbol{X}_1, \ldots, \boldsymbol{X}_m$ are independent and $(1/m)\sum_{i=1}^m \boldsymbol{X}_i = \boldsymbol{U}^\mathsf{T}\boldsymbol{S}^\mathsf{T}\boldsymbol{S}\boldsymbol{z}$, we have

$$\mathbb{E}\left[\|\boldsymbol{U}^\mathsf{T}\boldsymbol{S}^\mathsf{T}\boldsymbol{S}\boldsymbol{z} - \boldsymbol{U}^\mathsf{T}\boldsymbol{z}\|^2\right] = \mathbb{E}\left[\left\|\frac{1}{m}\sum_{j=1}^m \boldsymbol{X}_j - \mathbb{E}[\boldsymbol{X}_j]\right\|^2\right] = \frac{1}{m}\mathrm{var}(\boldsymbol{X}_1). \tag{A.4}$$

Then, using the bound on $m$, the result follows by Markov's inequality. $\square$

**Remark A.2.** *When using Theorem A.1 to compute the sample complexity bound in the subsequent proofs in Appendix A, we use*

$$m = O\left(\frac{\gamma(\boldsymbol{z})}{\delta\varepsilon^2}\right) \tag{A.5}$$

*samples. This is, in general, an upper bound on the sample complexity required for approximating the matrix-vector product, and can be tightened by including the term $\|\boldsymbol{U}^\mathsf{T}\boldsymbol{z}\|^2$.*

### A.2.2 Subspace Embedding

Before we prove a bound for the sketched regression $\phi^\mathrm{R}$ from Section 2.1, we prove a subspace embedding guarantee.

We begin by recalling the following well-known matrix concentration inequality; see e.g., [Tro15, Theorem 6.6.1].

**Imported Theorem A.3** (Matrix Bernstein's inequality). *Let $\boldsymbol{X}_1, \ldots, \boldsymbol{X}_m$ be zero-mean, independent, $q \times q$ Hermitian random matrices. Then, if $\|\boldsymbol{X}_i\| \leq L$ for all $i \in [m]$, we have*

$$\mathbb{P}\left[\left\|\sum_{i=1}^m \boldsymbol{X}_i\right\| \geq \varepsilon\right] \leq q \exp\left(-\frac{\varepsilon^2}{2\|\sum_{i=1}^m \mathbb{E}[\boldsymbol{X}_i^2]\| + (2L/3)\varepsilon}\right). \tag{A.6}$$

*In particular, denoting $\boldsymbol{X} = m^{-1}\sum_{i=1}^m \boldsymbol{X}_i$ and $\|\sum_{i=1}^m \mathbb{E}[\boldsymbol{X}_i^2]\| = m\sigma^2$, if*

$$m \geq \left(\frac{2\sigma^2}{\varepsilon^2} + \frac{2L}{3\varepsilon}\right)\log\left(\frac{q}{\delta}\right),$$

*it holds that $\mathbb{P}[\|\boldsymbol{X}\| \leq \varepsilon] \geq 1 - \delta$.*

By subsampling sufficiently many rows/columns of a matrix, we can obtain an appropriate subspace embedding guarantee.

**Theorem A.4** (Subspace embedding). *Given an $r \times d$ matrix $\boldsymbol{U}$, let $\boldsymbol{u}_1, \ldots, \boldsymbol{u}_r \in \mathbb{R}^d$ denote the columns of $\boldsymbol{U}^\mathsf{T}$. Let $\boldsymbol{S}$ be a $m \times r$ sketching matrix with iid rows drawn according to probability $\mathcal{P}$. Then, if*

$$m \geq \frac{2}{\varepsilon^2}\left\|\sum_{i=1}^r \frac{\|\boldsymbol{u}_i\|^2}{p_i}\boldsymbol{u}_i\boldsymbol{u}_i^\mathsf{T} - (\boldsymbol{U}^\mathsf{T}\boldsymbol{U})^2\right\|\log\left(\frac{d}{\delta}\right) + \frac{4}{3\varepsilon}\max_{i \in [r]}\frac{\|\boldsymbol{u}_i\|^2}{p_i}\log\left(\frac{d}{\delta}\right)$$

*it holds that*

$$\mathbb{P}\left[\|\boldsymbol{U}^\mathsf{T}\boldsymbol{S}^\mathsf{T}\boldsymbol{S}\boldsymbol{U} - \boldsymbol{U}^\mathsf{T}\boldsymbol{U}\| \leq \varepsilon\right] \geq 1 - \delta.$$

*Proof.* First, we write $\boldsymbol{U}^\mathsf{T} = (\boldsymbol{u}_1 \cdots \boldsymbol{u}_r)$, where $\boldsymbol{u}_i \in \mathbb{R}^d$ is the $i$th column of $\boldsymbol{U}^\mathsf{T}$ for $i \in [r]$. Then, $\boldsymbol{U}^\mathsf{T}\boldsymbol{U} = \sum_{i=1}^r \boldsymbol{u}_i\boldsymbol{u}_i^\mathsf{T}$. Similarly, it can be verified that $\boldsymbol{U}^\mathsf{T}\boldsymbol{S}^\mathsf{T}\boldsymbol{S}\boldsymbol{U} = \sum_{i=1}^m \boldsymbol{u}_{I_i}\boldsymbol{u}_{I_i}^\mathsf{T}/(mp_{I_i})$, where $I_1, \ldots, I_m$ are the random variables defining the sketching matrix. It follows that $\mathbb{E}[\boldsymbol{U}^\mathsf{T}\boldsymbol{S}^\mathsf{T}\boldsymbol{S}\boldsymbol{U}] = \boldsymbol{U}^\mathsf{T}\boldsymbol{U}$. For convenience, define $\boldsymbol{X}_i = \boldsymbol{u}_{I_i}\boldsymbol{u}_{I_i}^\mathsf{T}/p_{I_i} - \boldsymbol{U}^\mathsf{T}\boldsymbol{U}$ for $i \in [m]$ and $\boldsymbol{X} = \sum_{i=1}^m \boldsymbol{X}_i/m$. Then, for all $i \in [m]$, we have

$$\left\|\boldsymbol{u}_{I_i}\boldsymbol{u}_{I_i}^\mathsf{T}/p_{I_i}\right\| \leq \max_{k \in [r]}\frac{\|\boldsymbol{u}_k\|^2}{p_k} =: L. \tag{A.7}$$

It follows from triangle inequality and Jensen's inequality that $\|\boldsymbol{X}_i\| \leq 2L$ for all $i \in [m]$. Furthermore, using the fact that $\boldsymbol{X}_1, \ldots, \boldsymbol{X}_m$ are iid with zero mean, and symmetric, we have

$$\sum_{i=1}^m \mathbb{E}[\boldsymbol{X}_i^2] = m\mathbb{E}[\boldsymbol{X}_1^2] = m\left(\sum_{i=1}^r \|\boldsymbol{u}_i\|^2\frac{\boldsymbol{u}_i\boldsymbol{u}_i^\mathsf{T}}{p_i} - (\boldsymbol{U}^\mathsf{T}\boldsymbol{U})^2\right). \tag{A.8}$$

Writing $\|\sum_{i=1}^m \mathbb{E}[\boldsymbol{X}_i^2]\| = m\sigma^2$, we have

$$\sigma^2 = \left\|\sum_{i=1}^r \|\boldsymbol{u}_i\|^2\frac{\boldsymbol{u}_i\boldsymbol{u}_i^\mathsf{T}}{p_i} - (\boldsymbol{U}^\mathsf{T}\boldsymbol{U})^2\right\|. \tag{A.9}$$

The result then follows from Imported Theorem A.3. $\qquad\square$

### A.2.3 Sketched Regression

Together, Theorems A.1 and A.4 give a bound on sketched regression.

**Theorem A.5** (Sketched Regression). *Suppose $\boldsymbol{U}$ has orthonormal columns and let $\boldsymbol{y}^* = \operatorname{argmin}_{\boldsymbol{y}}\|\boldsymbol{U}\boldsymbol{y} - \boldsymbol{b}\|^2$. Let $\boldsymbol{S}$ be a $m \times q$ sketching matrix with iid rows drawn according to probability $\mathcal{P}$. Define*

$$\widehat{\boldsymbol{y}} = \operatorname*{argmin}_{\boldsymbol{y}}\|\boldsymbol{S}\boldsymbol{U}\boldsymbol{y} - \boldsymbol{S}\boldsymbol{b}\|^2. \tag{A.10}$$

*Then, if*

$$m = O\left(\frac{\gamma((\boldsymbol{I} - \boldsymbol{U}\boldsymbol{U}^\mathsf{T})\boldsymbol{b})}{\delta\varepsilon^2} + \eta\log\left(\frac{d}{\delta}\right)\right),$$

*it holds that*

$$\mathbb{P}[\|\boldsymbol{y}^* - \widehat{\boldsymbol{y}}\| \leq \varepsilon] \geq 1 - \delta. \tag{A.11}$$

*Proof.* Since $\boldsymbol{y}^*$ is the solution of $\min_{\boldsymbol{y}}\|\boldsymbol{U}\boldsymbol{y} - \boldsymbol{b}\|^2$, $\boldsymbol{U}\boldsymbol{y}^* - \boldsymbol{b}$ lies in the orthogonal complement of the range of $\boldsymbol{U}$, and therefore, $\boldsymbol{U}^\mathsf{T}(\boldsymbol{U}\boldsymbol{y}^* - \boldsymbol{b}) = 0$. Then, taking $\boldsymbol{z} = \boldsymbol{U}\boldsymbol{y}^* - \boldsymbol{b} = (\boldsymbol{I} - \boldsymbol{U}\boldsymbol{U}^\mathsf{T})\boldsymbol{b}$ in Theorem A.1, we can infer that using

$$m = O\left(\frac{\gamma(\boldsymbol{z})}{\delta\varepsilon^2}\right) \geq \left(\sum_{i=1}^r \frac{\|\boldsymbol{u}_i\|^2}{p_i}(\boldsymbol{e}_i^\mathsf{T}(\boldsymbol{I} - \boldsymbol{U}\boldsymbol{U}^\mathsf{T})\boldsymbol{b})^2\right)\frac{8}{\delta\varepsilon^2}, \tag{A.12}$$

we have with probability exceeding $1 - \delta/2$,

$$\left\|\boldsymbol{U}^\mathsf{T}\boldsymbol{S}^\mathsf{T}\boldsymbol{S}(\boldsymbol{U}\boldsymbol{y}^* - \boldsymbol{b})\right\| \leq \frac{\varepsilon}{2}. \tag{A.13}$$

Next, note that $\eta = \max_i \|u_i\|^2/p_i$ so

$$\sum_{i=1}^r \frac{\|\boldsymbol{u}_i\|^2}{p_i}\boldsymbol{u}_i\boldsymbol{u}_i^\mathsf{T} \preceq \eta \sum_{i=1}^r \boldsymbol{u}_i\boldsymbol{u}_i^\mathsf{T} = \eta\boldsymbol{U}^\mathsf{T}\boldsymbol{U}, \tag{A.14}$$

where $\boldsymbol{X} \preceq \boldsymbol{Y}$ indicates $\boldsymbol{Y} - \boldsymbol{X}$ is positive semi-definite. If $p_i > \|\boldsymbol{u}_i\|^2$ for all $i \in [r]$, then $1 > \sum_{i=1}^r \|\boldsymbol{u}_i\|^2 = \|\boldsymbol{U}\|_\mathsf{F}^2 = d$ (since $\boldsymbol{U}^\mathsf{T}\boldsymbol{U} = \boldsymbol{I}$), which is a contradiction. Thus, we must have $p_i \leq \|\boldsymbol{u}_i\|^2$ for some $i \in [r]$, or equivalently, $\eta \geq 1$. Then, because $\boldsymbol{U}$ has orthonormal columns, we have

$$\left\|\sum_{i=1}^r \frac{\|\boldsymbol{u}_i\|^2}{p_i}\boldsymbol{u}_i\boldsymbol{u}_i^\mathsf{T} - (\boldsymbol{U}^\mathsf{T}\boldsymbol{U})^2\right\| \leq (\eta - 1)\|\boldsymbol{U}^\mathsf{T}\boldsymbol{U}\| = \eta - 1 \leq \eta. \tag{A.15}$$

Therefore, by Theorem A.4, if

$$m = O\left(\eta\log\left(\frac{d}{\delta}\right)\right) \tag{A.16}$$

$$\geq 8\left\|\sum_{i=1}^r \frac{\|\boldsymbol{u}_i\|^2}{p_i}\boldsymbol{u}_i\boldsymbol{u}_i^\mathsf{T} - (\boldsymbol{U}^\mathsf{T}\boldsymbol{U})^2\right\|\log\left(\frac{d}{\delta}\right) + \frac{8}{3}\max_{i\in[r]}\frac{\|\boldsymbol{u}_i\|^2}{p_i}\log\left(\frac{d}{\delta}\right), \tag{A.17}$$

then, with probability exceeding $1 - \delta/2$,

$$\left\|\boldsymbol{U}^\mathsf{T}\boldsymbol{S}^\mathsf{T}\boldsymbol{S}\boldsymbol{U} - \boldsymbol{I}\right\| \leq 1/2. \tag{A.18}$$

For the remainder of the proof, we condition on (A.13) and (A.18), which, by a union bound, simultaneously occur with probability exceeding $1 - \delta$.

Using the triangle inequality, submultiplicitivty, and (A.18),

$$\|\boldsymbol{y}^* - \widehat{\boldsymbol{y}}\| = \left\|(\boldsymbol{U}^\mathsf{T}\boldsymbol{S}^\mathsf{T}\boldsymbol{S}\boldsymbol{U})(\boldsymbol{y}^* - \widehat{\boldsymbol{y}}) + (\boldsymbol{I} - (\boldsymbol{U}^\mathsf{T}\boldsymbol{S}^\mathsf{T}\boldsymbol{S}\boldsymbol{U}))(\boldsymbol{y}^* - \widehat{\boldsymbol{y}})\right\| \tag{A.19}$$

$$\leq \left\|(\boldsymbol{U}^\mathsf{T}\boldsymbol{S}^\mathsf{T}\boldsymbol{S}\boldsymbol{U})(\boldsymbol{y}^* - \widehat{\boldsymbol{y}})\right\| + \left\|(\boldsymbol{I} - (\boldsymbol{U}^\mathsf{T}\boldsymbol{S}^\mathsf{T}\boldsymbol{S}\boldsymbol{U}))(\boldsymbol{y}^* - \widehat{\boldsymbol{y}})\right\| \tag{A.20}$$

$$\leq \left\|(\boldsymbol{U}^\mathsf{T}\boldsymbol{S}^\mathsf{T}\boldsymbol{S}\boldsymbol{U})(\boldsymbol{y}^* - \widehat{\boldsymbol{y}})\right\| + \left\|\boldsymbol{I} - (\boldsymbol{U}^\mathsf{T}\boldsymbol{S}^\mathsf{T}\boldsymbol{S}\boldsymbol{U})\right\|\|\boldsymbol{y}^* - \widehat{\boldsymbol{y}}\| \tag{A.21}$$

$$\leq \left\|(\boldsymbol{U}^\mathsf{T}\boldsymbol{S}^\mathsf{T}\boldsymbol{S}\boldsymbol{U})(\boldsymbol{y}^* - \widehat{\boldsymbol{y}})\right\| + \frac{1}{2}\|\boldsymbol{y}^* - \widehat{\boldsymbol{y}}\|, \tag{A.22}$$

and hence,

$$\|\boldsymbol{y}^* - \widehat{\boldsymbol{y}}\| \leq 2\left\|(\boldsymbol{U}^\mathsf{T}\boldsymbol{S}^\mathsf{T}\boldsymbol{S}\boldsymbol{U})(\boldsymbol{y}^* - \widehat{\boldsymbol{y}})\right\|. \tag{A.23}$$

Next, by the optimality of $\widehat{\boldsymbol{y}}$ we have that $(\boldsymbol{S}\boldsymbol{U})^\mathsf{T}(\boldsymbol{S}\boldsymbol{U}\widehat{\boldsymbol{y}} - \boldsymbol{S}\boldsymbol{b}) = \boldsymbol{0}$ and hence that $(\boldsymbol{U}^\mathsf{T}\boldsymbol{S}^\mathsf{T}\boldsymbol{S}\boldsymbol{U})\widehat{\boldsymbol{y}} = \boldsymbol{U}^\mathsf{T}\boldsymbol{S}^\mathsf{T}\boldsymbol{S}\boldsymbol{b}$. Therefore, by (A.13),

$$\left\|(\boldsymbol{U}^\mathsf{T}\boldsymbol{S}^\mathsf{T}\boldsymbol{S}\boldsymbol{U})(\boldsymbol{y}^* - \widehat{\boldsymbol{y}})\right\| = \left\|\boldsymbol{U}^\mathsf{T}\boldsymbol{S}^\mathsf{T}\boldsymbol{S}(\boldsymbol{U}\boldsymbol{y}^* - \boldsymbol{b})\right\| \leq \frac{\varepsilon}{2}. \tag{A.24}$$

Combining the above equations gives the result. $\qquad\square$

## A.3 Proofs for Shapley Value Estimators

As noted in [MW25, Lemma 3.3], the matrix $\boldsymbol{Z}'$ nearly has orthonormal columns.

**Lemma A.6.** *Let $c_d = ((d-1)H_{d-2} - (d-2))/d$, where $H_d = \sum_{i=1}^{d}(1/i)$ is the $d^{th}$ harmonic number. Then,*

$$(\boldsymbol{Z}')^{\mathsf{T}}\boldsymbol{Z}' = \frac{d-1}{d}\boldsymbol{I} + c_d\boldsymbol{1}\boldsymbol{1}^{\mathsf{T}}. \tag{A.25}$$

*Proof.* First, we note that $(\boldsymbol{Z}')^{\mathsf{T}}\boldsymbol{Z}' = \boldsymbol{Z}^{\mathsf{T}}\boldsymbol{W}\boldsymbol{Z}$ is a matrix of size $d \times d$. Let $i, j \in \{1, \ldots, d\}$. Then it follows from definition that,

$$[\boldsymbol{Z}^{\mathsf{T}}\boldsymbol{W}\boldsymbol{Z}]_{ij} = \sum_{S:i,j\in S} k(S) \tag{A.26}$$

We separately consider the case where $i = j$. From the above,

$$[\boldsymbol{Z}^{\mathsf{T}}\boldsymbol{W}\boldsymbol{Z}]_{ii} = \sum_{S:i\in S} k(S) \tag{A.27}$$

$$= \sum_{|S|=1}^{d-1} \frac{(d-1)\binom{d-1}{|S|-1}}{\binom{d}{|S|}|S|(d-|S|)} \tag{A.28}$$

$$= \frac{d-1}{d}\sum_{|S|=1}^{d-1}\frac{1}{d-|S|} = \frac{(d-1)H_{d-1}}{d}, \tag{A.29}$$

Similarly for $i \neq j$,

$$[\boldsymbol{Z}^{\mathsf{T}}\boldsymbol{W}\boldsymbol{Z}]_{ij} = \sum_{S:i,j\in S} k(S) \tag{A.30}$$

$$= \sum_{|S|=2}^{d-1} \frac{(d-1)\binom{d-2}{|S|-2}}{\binom{d}{|S|}|S|(d-|S|)} \tag{A.31}$$

$$= \frac{1}{d}\sum_{|S|=2}^{d-1}\frac{|S|-1}{d-|S|} \tag{A.32}$$

$$= \frac{1}{d}\sum_{|S|=2}^{d-1}\left(\frac{d-1}{d-|S|} - 1\right) \tag{A.33}$$

$$= \frac{(d-1)H_{d-2} - (d-2)}{d}. \tag{A.34}$$

Define $\boldsymbol{1}$ as the all ones vector in $\mathbb{R}^d$ and $\boldsymbol{I}$ as the identity matrix of size $d \times d$. The matrix $\boldsymbol{Z}^{\mathsf{T}}\boldsymbol{W}\boldsymbol{Z}$ can then be written as

$$\boldsymbol{Z}^{\mathsf{T}}\boldsymbol{W}\boldsymbol{Z} = \frac{d-1}{d}\boldsymbol{I} + \frac{(d-1)H_{d-2} - (d-2)}{d}\boldsymbol{1}\boldsymbol{1}^{\mathsf{T}}, \tag{A.35}$$

which is the desired result. $\qquad\square$

Next, we describe the conversion from the constrained problem (1.2) to an unconstrained problem. Our approach is closely related to [MW25, Lemma 3.1]. However, as noted in Section 2, our approach allows arbitrary $\lambda$ (where as [MW25] only allows $\lambda = \alpha$). In addition, on a more technical note, we state our results in terms of the argmin of a regression problem involving a full-rank matrix $\boldsymbol{U}$. The result of [MW25] is stated in terms of the argmin of a regression problem involving rank-deficient matrix, which is not uniquely defined. As such, their result implicitly assumes that the argmin returns one particular solution (the minimum norm solution); see Appendix B.3.

***Proof of Theorem 2.1.*** Using $\boldsymbol{Q}^{\mathsf{T}}\mathbf{1} = \mathbf{0}$, $\boldsymbol{Q}^{\mathsf{T}}\boldsymbol{Q} = \boldsymbol{I}$, and (A.25),

$$\boldsymbol{U}^{\mathsf{T}}\boldsymbol{U} = \frac{d}{d-1}\boldsymbol{Q}^{\mathsf{T}}(\boldsymbol{Z}')^{\mathsf{T}}\boldsymbol{Z}'\boldsymbol{Q} = \frac{d}{d-1}\boldsymbol{Q}^{\mathsf{T}}\left(\frac{d-1}{d}\boldsymbol{I} + c_d\mathbf{1}\mathbf{1}^{\mathsf{T}}\right)\boldsymbol{Q} = \boldsymbol{I}. \tag{A.36}$$

Therefore, $(\boldsymbol{U}^{\mathsf{T}}\boldsymbol{U})^{-1}\boldsymbol{U}^{\mathsf{T}} = \boldsymbol{U}^{\mathsf{T}}$ and so $\boldsymbol{Q}\operatorname{argmin}_{\boldsymbol{x}\in\mathbb{R}^{d-1}}\|\boldsymbol{U}\boldsymbol{x} - \boldsymbol{b}_\lambda\|^2 + \alpha\mathbf{1} = \boldsymbol{Q}\boldsymbol{U}^{\mathsf{T}}\boldsymbol{b}_\lambda + \alpha\mathbf{1}$.

It remains to show these formulations are equivalent to (1.2). Since $\boldsymbol{Q}^{\mathsf{T}}\mathbf{1} = \mathbf{0}$ and $\mathbf{1}^{\mathsf{T}}\mathbf{1} = d$, observe that

$$\{\boldsymbol{\phi} : \boldsymbol{\phi} \in \mathbb{R}^d, \mathbf{1}^{\mathsf{T}}\boldsymbol{\phi} = v([d]) - v(\emptyset)\} = \{\boldsymbol{Q}\boldsymbol{x} + \alpha\mathbf{1} : \boldsymbol{x} \in \mathbb{R}^{d-1}\}, \tag{A.37}$$

with the natural bijection $\boldsymbol{\phi} \leftrightarrow \boldsymbol{Q}\boldsymbol{x} + \alpha\mathbf{1}$ between $\boldsymbol{\phi}$ and $\boldsymbol{x}$. Thus, using the definitions of $\boldsymbol{U}$ and $\boldsymbol{b}_\alpha$,

$$\boldsymbol{\phi}^* = \operatorname*{argmin}_{\substack{\boldsymbol{\phi}\in\mathbb{R}^d \\ \mathbf{1}^{\mathsf{T}}\boldsymbol{\phi}=v([d])-v(\emptyset)}} \|\boldsymbol{Z}'\boldsymbol{\phi} - \boldsymbol{b}\|^2 \tag{A.38}$$

$$= \boldsymbol{Q}\operatorname*{argmin}_{\boldsymbol{x}\in\mathbb{R}^{d-1}}\|\boldsymbol{Z}'(\boldsymbol{Q}\boldsymbol{x} + \alpha\mathbf{1}) - \boldsymbol{b}\|^2 + \alpha\mathbf{1} \tag{A.39}$$

$$= \boldsymbol{Q}\operatorname*{argmin}_{\boldsymbol{x}\in\mathbb{R}^{d-1}}\|\boldsymbol{U}\boldsymbol{x} - \boldsymbol{b}_\alpha\|^2 + \alpha\mathbf{1}. \tag{A.40}$$

Now, since $\boldsymbol{Q}^{\mathsf{T}}\mathbf{1} = \mathbf{0}$,

$$\boldsymbol{U}^{\mathsf{T}}\boldsymbol{Z}'\mathbf{1} = \boldsymbol{Q}^{\mathsf{T}}(\boldsymbol{Z}')^{\mathsf{T}}\boldsymbol{Z}'\mathbf{1} = \boldsymbol{Q}^{\mathsf{T}}\left(\frac{d-1}{d}\boldsymbol{I} + c_d\mathbf{1}\mathbf{1}^{\mathsf{T}}\right)\mathbf{1} = \mathbf{0}. \tag{A.41}$$

Therefore, for any $\lambda$,

$$\operatorname*{argmin}_{\boldsymbol{x}\in\mathbb{R}^{d-1}}\|\boldsymbol{U}\boldsymbol{x} - \boldsymbol{b}_\lambda\|^2 = \operatorname*{argmin}_{\boldsymbol{x}\in\mathbb{R}^{d-1}}\|\boldsymbol{U}\boldsymbol{x} - (\boldsymbol{b} - \lambda\boldsymbol{Z}'\mathbf{1})\|^2 = \operatorname*{argmin}_{\boldsymbol{x}\in\mathbb{R}^{d-1}}\|\boldsymbol{U}\boldsymbol{x} - \boldsymbol{b}\|^2. \tag{A.42}$$

This gives the desired result. $\qquad\square$

Finally, we use Theorem 2.1 and the bounds from Appendix A.2 to prove our main approximation guarantee.

***Proof of Theorem 2.2.*** We analyze the estimators individually. Recall from Theorem 2.1 that

$$\boldsymbol{\phi}^* = \boldsymbol{Q}\operatorname*{argmin}_{\boldsymbol{x}\in\mathbb{R}^{d-1}}\|\boldsymbol{U}\boldsymbol{x} - \boldsymbol{b}_\lambda\|^2 + \alpha\mathbf{1} = \boldsymbol{Q}\boldsymbol{U}^{\mathsf{T}}\boldsymbol{b}_\lambda + \alpha\mathbf{1}. \tag{A.43}$$

We will use both of these formulations.

**Regression:** Observe,

$$\boldsymbol{\phi}_\lambda^{\mathrm{R}} = \boldsymbol{Q}\operatorname*{argmin}_{\boldsymbol{x}\in\mathbb{R}^{d-1}}\|\boldsymbol{S}(\boldsymbol{U}\boldsymbol{x} - \boldsymbol{b}_\lambda)\|^2 + \alpha\mathbf{1} = \boldsymbol{Q}(\boldsymbol{S}\boldsymbol{U})^+\boldsymbol{S}\boldsymbol{b}_\lambda + \alpha\mathbf{1}.$$

Now, since $\boldsymbol{Q}^{\mathsf{T}}\boldsymbol{Q} = \boldsymbol{I}$,

$$\|\boldsymbol{\phi}^* - \boldsymbol{\phi}_\lambda^{\mathrm{R}}\| = \|\boldsymbol{Q}\boldsymbol{U}^{\mathsf{T}}\boldsymbol{b}_\lambda - \boldsymbol{Q}(\boldsymbol{S}\boldsymbol{U})^+\boldsymbol{S}\boldsymbol{b}_\lambda\| = \|\boldsymbol{U}^{\mathsf{T}}\boldsymbol{b}_\lambda - (\boldsymbol{S}\boldsymbol{U})^+\boldsymbol{S}\boldsymbol{b}_\lambda\|. \tag{A.44}$$

By Theorem A.5, if

$$m = O\left(\frac{\gamma(\boldsymbol{P}_{\boldsymbol{U}}\boldsymbol{b}_\lambda)}{\delta\varepsilon^2} + \eta\log\left(\frac{d}{\delta}\right)\right), \tag{A.45}$$

then

$$\mathbb{P}\big[\|\boldsymbol{U}^{\mathsf{T}}\boldsymbol{b}_\lambda - (\boldsymbol{S}\boldsymbol{U})^+\boldsymbol{S}\boldsymbol{b}_\lambda\|^2 \le \varepsilon\big] \ge 1 - \delta.$$

**Matrix-Vector Multiplication:** By definition,

$$\boldsymbol{\phi}_\lambda^{\mathrm{M}} = \boldsymbol{Q}\boldsymbol{U}^{\mathsf{T}}\boldsymbol{S}^{\mathsf{T}}\boldsymbol{S}\boldsymbol{b}_\lambda + \alpha\mathbf{1}. \tag{A.46}$$

Then, since $\boldsymbol{Q}^{\mathsf{T}}\boldsymbol{Q} = \boldsymbol{I}$,

$$\|\boldsymbol{\phi}^* - \boldsymbol{\phi}_\lambda^{\mathrm{M}}\| = \|\boldsymbol{Q}\boldsymbol{U}^{\mathsf{T}}\boldsymbol{b}_\lambda - \boldsymbol{Q}\boldsymbol{U}^{\mathsf{T}}\boldsymbol{S}^{\mathsf{T}}\boldsymbol{S}\boldsymbol{b}_\lambda\| = \|\boldsymbol{U}^{\mathsf{T}}\boldsymbol{b}_\lambda - \boldsymbol{U}^{\mathsf{T}}\boldsymbol{S}^{\mathsf{T}}\boldsymbol{S}\boldsymbol{b}_\lambda\|. \tag{A.47}$$

By Theorem A.1, if

$$m = O\left(\frac{\gamma(\boldsymbol{b}_\lambda)}{\delta\varepsilon^2}\right) \ge \left(\sum_{i=1}^r \frac{\|\boldsymbol{u}_i\|^2}{p_i}((\boldsymbol{b}_\lambda)_i)^2 - \|\boldsymbol{U}^{\mathsf{T}}\boldsymbol{b}_\lambda\|^2\right)\frac{1}{\delta\varepsilon^2} \tag{A.48}$$

then

$$\mathbb{P}\big[\|\boldsymbol{U}^{\mathsf{T}}\boldsymbol{b}_\lambda - \boldsymbol{U}^{\mathsf{T}}\boldsymbol{S}^{\mathsf{T}}\boldsymbol{S}\boldsymbol{b}_\lambda\| < \varepsilon\big] > 1 - \delta. \tag{A.49}$$

This establishes the result. $\qquad\square$

## A.4 Fine-grained bounds for specific probability distributions

**Theorem A.7.** *Map the index $i \in [2^d - 2]$ to integers $(h, l)$ satisfying $h \in [d - 1]$, $l \in [\binom{d}{h}]$, as $i = \sum_{j=1}^{h-1} \binom{d}{j} + l$. Then, we have*

$$\|\boldsymbol{u}_i\|^2 = \|\boldsymbol{u}_{h,l}\|^2 = \frac{1}{\binom{d}{h}} \tag{A.50}$$

*for all $l \in [\binom{d}{h}]$ and all $h \in [d-1]$. Moreover,*

1. *($\boldsymbol{\ell_2}$-squared) For $h \in [d-1]$ and $l \in [\binom{d}{h}]$, we have*

$$p_{h,l} = \frac{\|\boldsymbol{u}_{h,l}\|^2}{\|\boldsymbol{U}\|_{\mathsf{F}}^2} = \frac{1}{(d-1)\binom{d}{h}},$$

$$\gamma(\boldsymbol{z}) = (d-1)\|\boldsymbol{z}\|^2 \qquad \text{and} \qquad \eta = d - 1.$$

2. *(**Kernel**) For $h \in [d-1]$ and $l \in [\binom{d}{h}]$, denoting $k(h) = (d-1)/(\binom{d}{h}(h(d-h)))$ we have*

$$p_{h,l} = \frac{k(h)}{\sum_{j=1}^{d-1} k(j)\binom{d}{j}} = \frac{1}{\binom{d}{h}} \frac{\frac{1}{h(d-h)}}{\sum_{j=1}^{d-1} \frac{1}{j(d-j)}},$$

$$\gamma(\boldsymbol{z}) = \frac{2}{d}\left(\sum_{h=1}^{d-1} \frac{1}{h}\right) \sum_{h=1}^{d-1} \sum_{l=1}^{\binom{d}{h}} h(d-h) z_{h,l}^2 \quad \text{and} \quad \eta \le \frac{d}{2} \sum_{h=1}^{d-1} \frac{1}{h}.$$

3. *(**Modified** $\ell_2$) For $h \in [d-1]$ and $l \in [\binom{d}{h}]$, we have*

$$p_{h,l} = \frac{\sqrt{k(h)}\|\boldsymbol{u}_{h,l}\|}{\sum_{j=1}^{d-1} \sum_{l=1}^{\binom{d}{j}} \sqrt{k(j)}\|\boldsymbol{u}_{j,l}\|} = \frac{1}{\binom{d}{h}} \frac{\frac{1}{\sqrt{h(d-h)}}}{\sum_{j=1}^{d-1} \frac{1}{\sqrt{j(d-j)}}},$$

$$\gamma(\boldsymbol{z}) = \left(\sum_{h=1}^{d-1} \frac{1}{\sqrt{h(d-h)}}\right) \sum_{h=1}^{d-1} \sum_{l=1}^{\binom{d}{h}} \sqrt{h(d-h)} z_{h,l}^2 \quad \text{and} \quad \eta \le \frac{d}{2} \sum_{h=1}^{d-1} \frac{1}{\sqrt{h(d-h)}}.$$

*Proof.* Denote $r = 2^d - 2$, and let $\boldsymbol{e}_1, \ldots, \boldsymbol{e}_r \in \mathbb{R}^d$ be the standard basis vectors. Since $\boldsymbol{u}_1, \ldots, \boldsymbol{u}_r$ are the columns of $\boldsymbol{U}^\mathsf{T}$, we can write $\boldsymbol{u}_i = \boldsymbol{U}^\mathsf{T} \boldsymbol{e}_i$ for all $i \in [r]$. It follows that

$$\|\boldsymbol{u}_i\|^2 = \boldsymbol{e}_i^\mathsf{T} \boldsymbol{U}\boldsymbol{U}^\mathsf{T} \boldsymbol{e}_i = \frac{d}{d-1} \boldsymbol{e}_i^\mathsf{T} \sqrt{\boldsymbol{W}} \boldsymbol{Z}\boldsymbol{P}\boldsymbol{Z}^\mathsf{T} \sqrt{\boldsymbol{W}} \boldsymbol{e}_i. \tag{A.51}$$

Now, we map $i$ to $(h, l)$ for appropriate integers $h \in [d-1]$ and $l \in [\binom{d}{h}]$, so that the subset $S_i \subset [d]$ is of size $h$ (according to the chosen ordering of subsets). Then, writing $k(h) = k(S_i)$, we have $\boldsymbol{Z}^\mathsf{T} \sqrt{\boldsymbol{W}} \boldsymbol{e}_{h,l} = \sqrt{k(h)} \boldsymbol{Z}^\mathsf{T} \boldsymbol{e}_{h,l} = \sqrt{k(h)} \boldsymbol{z}_{h,l}$, where $\boldsymbol{z}_{h,l}$ is a $d$-dimensional vector with 1 at entry $j$ if $j \in S_{h,l}$ and 0 otherwise. Substituting this in (A.51), using $\boldsymbol{P} = \boldsymbol{I} - (1/d)\boldsymbol{1}\boldsymbol{1}^\mathsf{T}$ and $|S_{h,l}| = \|\boldsymbol{z}_{h,l}\|_1 = \|\boldsymbol{z}_{h,l}\|^2 = h$, we obtain

$$\begin{aligned}
\|\boldsymbol{u}_i\|^2 &= \frac{d}{d-1} k(h) \left(\|\boldsymbol{z}_{h,l}\|^2 - \frac{1}{d}\|\boldsymbol{z}_{h,l}\|_1^2\right) \\
&= \frac{d}{d-1} \frac{d-1}{\binom{d}{h} h(h-d)} \left(h - \frac{h^2}{d}\right) \\
&= \frac{1}{\binom{d}{h}}.
\end{aligned} \tag{A.52}$$

1. It can be verified that $\sum_{i=1}^{2^d-1} \|\boldsymbol{u}_i\|^2 = \|\boldsymbol{U}_\mathsf{F}\|^2 = d - 1$. The result follow from the definition of $p_{h,l}$, and $\gamma$, $\eta$ in (2.3).

2. Noting that $k(S)$ depends only on the size of the subset $S \subseteq [d]$, $p_{h,l}$ in Item 2 is obtained by direct calculation. Observe that

$$\sum_{h=1}^{d-1} \frac{1}{h(d-h)} = \frac{1}{d} \sum_{h=1}^{d-1} \left( \frac{1}{h} + \frac{1}{d-h} \right) = \frac{2}{d} \sum_{h=1}^{d-1} \frac{1}{h}, \tag{A.53}$$

and therefore,

$$\frac{\|\boldsymbol{u}_{h,l}\|^2}{p_{h,l}} = \left( \frac{2}{d} \sum_{h=1}^{d-1} \frac{1}{h} \right) h(d-h), \tag{A.54}$$

for $h \in [d-1]$ and $l \in [\binom{d}{h}]$. Since $h(d-h) \leq d^2/4$ for $h \in [d-1]$, Item 2 follows from (2.3) by direct substitution.

3. We obtain $p_{h,l}$ in Item 3 by direct substitution. Since

$$\frac{\|\boldsymbol{u}_{h,l}\|^2}{p_{h,l}} = \left( \sum_{h=1}^{d-1} \frac{1}{\sqrt{h(d-h)}} \right) \sqrt{h(d-h)} \tag{A.55}$$

for $h \in [d-1]$ and $l \in [\binom{d}{h}]$, we obtain Item 3. $\qquad\square$

**Remark A.8.** *The sum over $1/h$ in Item 2 and over $1/\sqrt{h(d-h)}$ in Item 3 only mildly depend on d. Indeed,*

$$\sum_{h=1}^{d-1} \frac{1}{h} = \Theta(\log(d)) \qquad \text{and} \qquad \sum_{h=1}^{d-1} \frac{1}{\sqrt{h(d-h)}} = \Theta(1). \tag{A.56}$$

*This can be seen from the (well-known) bound*

$$\log(d) = \int_1^d \frac{1}{x} dx \leq \sum_{h=1}^{d-1} \frac{1}{h} = 1 + \sum_{h=2}^{d-1} \frac{1}{h} \leq 1 + \int_1^{d-1} \frac{1}{x} dx = 1 + \log(d-1), \tag{A.57}$$

*where the approximation with the integral uses the fact that $h \mapsto 1/h$ is a decreasing function. Similarly, since $\lceil (d-1)/2 \rceil \leq d/2$, we have*

$$\begin{aligned}
\sum_{h=1}^{d-1} \frac{1}{\sqrt{h(d-h)}} &\leq 2 \sum_{h=1}^{\lceil (d-1)/2 \rceil} \frac{1}{\sqrt{h(d-h)}} \\
&\leq 2 \left( \frac{1}{\sqrt{d-1}} + \int_1^{d/2} \frac{1}{\sqrt{x(d-x)}} dx \right) \\
&= 2 \left( \frac{1}{\sqrt{d-1}} + 2\arctan(\sqrt{d-1}) - \frac{\pi}{2} \right) \xrightarrow[d\to\infty]{} \pi,
\end{aligned} \tag{A.58}$$

*and since $\lfloor (d-1)/2 \rfloor \geq d/2 - 1$, we have*

$$\begin{aligned}
\sum_{h=1}^{d-1} \frac{1}{\sqrt{h(d-h)}} &\geq 2 \sum_{h=1}^{\lfloor (d-1)/2 \rfloor} \frac{1}{h(d-h)} \\
&\geq 2 \left( \int_1^{d/2} \frac{1}{\sqrt{x(d-x)}} dx \right) \\
&= 2 \left( 2\arctan(\sqrt{d-1}) - \frac{\pi}{2} \right) \xrightarrow[d\to\infty]{} \pi.
\end{aligned} \tag{A.59}$$

Theorem A.7 allows us to directly compare the values of $\gamma$ for the different sampling strategies we consider.

**Corollary A.9.** *Denote $\gamma_{\ell_2^2}$, $\gamma_{\text{ker}}$, $\gamma_{\text{m-}\ell_2}$ to be the expressions for $\gamma$ for $\ell_2$-squared Item 1, kernel Item 2, and modified $\ell_2$ Item 3 sampling schemes respectively. Then, for all $\boldsymbol{z} \in \mathbb{R}^{2^d-2}$, we have*

$$\Theta\left(\frac{\log(d)}{d}\right) \leq \frac{\gamma_{\text{ker}}(\boldsymbol{z})}{\gamma_{\ell_2^2}(\boldsymbol{z})} \leq \Theta(\log(d)), \tag{A.60}$$

$$\Theta\left(\frac{1}{\sqrt{d}}\right) \leq \frac{\gamma_{\text{m-}\ell_2}(\boldsymbol{z})}{\gamma_{\ell_2^2}(\boldsymbol{z})} \leq \Theta(1), \tag{A.61}$$

*and*

$$\Theta\left(\frac{\log(d)}{\sqrt{d}}\right) \leq \frac{\gamma_{\text{ker}}(\boldsymbol{z})}{\gamma_{\text{m-}\ell_2}(\boldsymbol{z})} \leq \Theta(\log(d)). \tag{A.62}$$

*Proof.* Since $d - 1 \leq h(d-h) \leq d^2/4$ for all $h \in [d-1]$, we have

$$(d-1)\|\boldsymbol{z}_{h,l}\|^2 \leq \sum_{h=1}^{d-1} \sum_{l=1}^{\binom{d}{h}} h(d-h) z_{h,l}^2 \leq (d^2/4)\|\boldsymbol{z}_{h,l}\|^2 \tag{A.63}$$

and

$$\sqrt{d-1}\|\boldsymbol{z}_{h,l}\|^2 \leq \sum_{h=1}^{d-1} \sum_{l=1}^{\binom{d}{h}} \sqrt{h(d-h)} z_{h,l}^2 \leq (d/2)\|\boldsymbol{z}_{h,l}\|^2. \tag{A.64}$$

Similarly, since $(\sqrt{d-1}/d)\sqrt{h(d-h)} \leq h(d-h)/d \leq (1/2)\sqrt{h(d-h)}$ for $h \in [d-1]$, we have

$$\sqrt{d-1} \sum_{h=1}^{d-1} \sum_{l=1}^{\binom{d}{h}} \sqrt{h(d-h)} z_{h,l}^2 \leq \sum_{h=1}^{d-1} \sum_{l=1}^{\binom{d}{h}} h(d-h) z_{h,l}^2 \leq \frac{d}{2} \sum_{h=1}^{d-1} \sum_{l=1}^{\binom{d}{h}} \sqrt{h(d-h)} z_{h,l}^2. \tag{A.65}$$

Then, (A.60), (A.61), and (A.62) follow from Theorem A.7 and Remark A.8. $\qquad\square$

These bounds suggest that kernel weights perform at most a log factor worse than leverage scores, while it can perform nearly $d$ better than leverage scores. On the other hand, the performance of modified $\ell_2$ weights is never worse than leverage scores (up to constant factors), but can nearly do $\sqrt{d}$ better than leverage scores. In Appendix E, we explicitly construct a toy model that demonstrates such an advantage. While these results are only upper bounds on the sample complexities, we also observe similar results in experiments. Using Theorem A.7, we can derive the values of $\gamma$ and $\eta$ listed in Table 1 for the different sampling strategies as follows.

**Corollary A.10.** *Define $\boldsymbol{H}$ to be a $(2^d - 2) \times (2^d - 2)$ dimensional diagonal matrix with diagonal entries*

$$\boldsymbol{H}_{(h,l),(h,l)} = \frac{\sqrt{h(d-h)}}{d}$$

*for $h \in [d-1]$ and $l \in [\binom{d}{h}]$, so that*

$$\lambda_{\min}(\boldsymbol{H}) = \Theta\left(\frac{1}{\sqrt{d}}\right) \quad \text{and} \quad \lambda_{\max}(\boldsymbol{H}) = \Theta(1).$$

*Then, we have the following expressions for $\gamma(\boldsymbol{z})$ and $\eta$ for all $\boldsymbol{z} \in \mathbb{R}^{2^d-2}$.*

    1. *($\boldsymbol{\ell_2}$-squared)*
$$\gamma(\boldsymbol{z}) = \Theta\left(d\|\boldsymbol{z}\|^2\right) \quad \text{and} \quad \eta = \Theta(d).$$

    2. *(Kernel)*
$$\gamma(\boldsymbol{z}) = \Theta\left(d\log(d)\|\boldsymbol{H}\boldsymbol{z}\|^2\right) \quad \text{and} \quad \eta = \Theta(d\log(d)).$$

    3. *(Modified $\boldsymbol{\ell_2}$)*
$$\gamma(\boldsymbol{z}) = \Theta\left(d\|\sqrt{\boldsymbol{H}}\boldsymbol{z}\|_2^2\right) \quad \text{and} \quad \eta = \Theta(d).$$

*Proof.* This follows from Theorem A.7, Remark A.8 and the definition of $\boldsymbol{H}$. $\qquad\square$

**Remark A.11.** *The distributions considered in Theorem A.7 are actually a special case of a family of distributions, obtained by interpolating between kernel weights and leverage scores. Specifically, given $\tau \in [0,1]$, we can consider the weighted geometric mean $(k(h))^\tau (\|\boldsymbol{u}_{h,l}\|^2)^{(1-\tau)}$ of $k(h)$ and $\|\boldsymbol{u}_{h,l}\|^2$ for $h \in [d-1]$ and $l \in [\binom{d}{h}]$. This gives rise to the distribution*

$$p_{h,l}^\tau = \frac{1}{\binom{d}{h}} \frac{\left(\frac{1}{h(d-h)}\right)^\tau}{\sum_{j=1}^{d-1}\left(\frac{1}{j(d-j)}\right)^\tau}. \tag{A.66}$$

*For $\tau = 0$, we get the leverage scores (or $\ell_2$-squared distribution), $\tau = 1$ gives the kernel weight distribution, and $\tau = 1/2$ gives the modified $\ell_2$ distribution.*

*Denoting*

$$\mathcal{N}_\tau = \sum_{j=1}^{d-1}\left(\frac{1}{j(d-j)}\right)^\tau \tag{A.67}$$

*to be the normalization factor, we have*

$$\frac{\|\boldsymbol{u}_{h,l}\|^2}{p_{h,l}^\tau} = (h(d-h))^\tau \, \mathcal{N}_\tau \tag{A.68}$$

*for $h \in [d-1]$ and $l \in [\binom{d}{h}]$. It follows that*

$$\eta_\tau = \begin{cases} \left(\frac{d^2}{4}\right)^\tau \mathcal{N}_\tau & \text{if } d \text{ is even} \\ \left(\frac{d^2-1}{4}\right)^\tau \mathcal{N}_\tau & \text{if } d \text{ is odd,} \end{cases} \tag{A.69}$$

*and*

$$\gamma_\tau(\boldsymbol{z}) = \mathcal{N}_\tau \sum_{h,l} (h(d-h))^\tau \, z_{h,l}^2 \tag{A.70}$$

*for $\boldsymbol{z} \in \mathbb{R}^{2^d-2}$.*

*Using similar arguments as in Remark A.8, we can show that*

$$\mathcal{N}_\tau = \begin{cases} \Theta(d^{1-2\tau}) & \text{if } 0 \leq \tau < 1 \\ \Theta\left(\frac{\log(d)}{d}\right) & \text{if } \tau = 1. \end{cases} \tag{A.71}$$

*Here, we used the fact that*

$$\int_1^{d/2} \frac{1}{(x(d-x))^\tau}\mathrm{d}x = d^{1-2\tau}(\mathrm{B}_{1/2}(1-\tau, 1-\tau) - \mathrm{B}_{1/d}(1-\tau, 1-\tau)) = \Theta(d^{1-2\tau}) \tag{A.72}$$

*for $0 \leq \tau < 1$, where $\mathrm{B}_z(a,b) = \int_0^z t^{a-1}(1-t)^{b-1}\mathrm{d}t$ is the incomplete beta function.*

*Therefore, we have*

$$\gamma_\tau(\boldsymbol{z}) = \Theta(d\|\boldsymbol{H}^\tau \boldsymbol{z}\|^2) \text{ and } \eta_\tau = \Theta(d) \tag{A.73}$$

*for $0 \leq \tau < 1$, and*

$$\gamma_\tau(\boldsymbol{z}) = \Theta(d\log(d)\|\boldsymbol{H}^\tau \boldsymbol{z}\|^2) \text{ and } \eta_\tau = \Theta(d\log(d)) \tag{A.74}$$

*for $\tau = 1$. For $0 \leq \tau < 1$, we do no worse than leverage score sampling. We remark that because the $\Theta$ notation hides constants, for a given dimension, one can choose an appropriate $\tau$ that minimizes these constants. It remains to see how such a strategy performs in practice.*

## A.5 Theoretical guarantees for sampling without replacement

In this section, we prove guarantees for matrix vector multiplication estimator and the regression estimator when the rows/columns are sampled without replacement.[7] We follow the strategy of [MW25] for sampling indices without replacement.

Let $U$ be an $r \times q$ dimensional matrix, with rows $u_1, \ldots, u_r \in \mathbb{R}^q$. To sample the rows of $U$ without replacement, we suppose that we have $r$ independent Bernoulli random variables $Y_1, \ldots, Y_r$, where $Y_i$ has mean $q_i > 0$ for $i \in [r]$. We interpret $Y_i = 1$ as having picked the $i$th row, and $Y_i = 0$ as not having picked the $i$th row. The expected number of samples (or rows) is $m_0 = \sum_{i=1}^r q_i$. Thus, on an average, we will sample $m_0$ rows, none of which are the same. Observe that while we can control the expected number of samples by choosing the probabilities $q_1, \ldots, q_r$, the actual number of samples $m$ we draw is random. If $i_1, \ldots, i_m$ are the (distinct) indices we pick, then the sketching matrix $S$ is $m \times r$ dimensional, with $j$th row having the element $1/\sqrt{q_{i_j}}$ at location $i_j$ and zero elsewhere for $j \in [m]$. Note that an important feature of such a sampling without replacement scheme is that the probabilities $q_1, \ldots, q_r$ need *not* sum to 1 because they independently determine whether or not a given row is picked.

### A.5.1 Approximate Matrix-Vector Multiplication

We derive the following guarantee for the matrix-vector multiplication estimator for sampling without replacement. Since the number of samples are not fixed, we instead calculate the estimation error for a fixed expected number of samples (which is determined by the probabilities $q_1, \ldots, q_r$).

**Theorem A.12** (Matrix-Vector multiplication, sampling without replacement). *Given a matrix $U \in \mathbb{R}^{r \times q}$ and a vector $z \in \mathbb{R}^r$, let $S$ be an $m \times r$ dimensional sketching matrix constructed by sampling rows of $U$ without replacement according to probabilities $q_1, \ldots, q_r$. Then, using an expected number of samples $\sum_{i=1}^r q_i$, we have*

$$\mathbb{P}\left[\left\|U^\mathsf{T} S^\mathsf{T} S z - U^\mathsf{T} z\right\| \leq \varepsilon\right] \geq 1 - \delta$$

*for*

$$\varepsilon = \sqrt{\frac{1}{\delta} \sum_{i=1}^r \left(\frac{1}{q_i} - 1\right) \|u_i\|^2 z_i^2}. \tag{A.75}$$

*Proof.* Let $Y_1, \ldots, Y_r$ be independent Bernoulli random variables with means $q_1, \ldots, q_r$ respectively. Then, the random variable

$$\widehat{X} = U^T S^T S z = \sum_{i=1}^r Y_i \frac{u_i z_i}{q_i} \tag{A.76}$$

is an unbiased estimator of $U^\mathsf{T} z$. Denote $\text{var}(\widehat{X}) = \mathbb{E}[\|\widehat{X} - \mathbb{E}[\widehat{X}]\|^2]$ to be variance of $\widehat{X}$. Then, since all $Y_1, \ldots, Y_r$ are independent, we have

$$\text{var}(\widehat{X}) = \sum_{i=1}^r \text{var}\left(Y_i \frac{u_i z_i}{q_i}\right) \tag{A.77}$$

$$= \sum_{i=1}^r \left(\frac{1}{q_i} - 1\right) \|u_i\|^2 z_i^2. \tag{A.78}$$

Since $\text{var}(\widehat{X}) = \mathbb{E}[\|\widehat{X} - \mathbb{E}[\widehat{X}]\|_2^2]$, by Markov's inequality, we have

$$\mathbb{P}\left[\|\widehat{X} - \mathbb{E}[\widehat{X}]\| \geq \varepsilon\right] \leq \frac{\text{var}(\widehat{X})^2}{\varepsilon^2} = \frac{1}{\varepsilon^2} \sum_{i=1}^r \left(\frac{1}{q_i} - 1\right) \|u_i\|^2 z_i^2. \tag{A.79}$$

Setting the right-hand-side of the above inequality equal to $\delta$ and solving for $\varepsilon$ gives us (A.75). $\square$

We can use the above result to derive the error bounds in terms of the function $\gamma(z)$ defined in (2.3).

---

[7]We note that the term "sampling without replacement" is perhaps a bit of a misnomer for this type of sampling scheme. Nevertheless, we use it in order to maintain consistency with [MW25].

**Corollary A.13.** *Let $\mathcal{P} = (p_1, \ldots, p_r)$ be a probability distribution on $[r]$ with $p_i > 0$ for all $i \in [r]$. Given a number $m_0 \in (0, r]$, let $c > 0$ be a constant for which $q_i = \min\{1, cp_i\}$ for $i \in [r]$ and $\sum_{i=1}^r q_i = m_0$. Then, given error $\varepsilon > 0$ and confidence level $1 - \delta \in (0, 1)$, if*

$$m_0 \geq \frac{\gamma(\boldsymbol{z})}{\delta\varepsilon^2},$$

*by sampling the rows of $\boldsymbol{U}$ without replacement according to probabilities $q_1, \ldots, q_r$, we have*

$$\mathbb{P}\big[\big\|\boldsymbol{U}^\mathsf{T}\boldsymbol{S}^\mathsf{T}\boldsymbol{S}\boldsymbol{z} - \boldsymbol{U}^\mathsf{T}\boldsymbol{z}\big\| \leq \varepsilon\big] \geq 1 - \delta.$$

*Proof.* First, note that by the continuity of $c \mapsto \min\{1, cp_i\}$ for all $i \in [r]$, given a real number $m_0 \in (0, r]$, there is always some $c > 0$ for which $\sum_{i=1}^r \min\{1, cp_i\} = m_0$ by the intermediate value theorem. Furthermore, $m_0 = \sum_{i=1}^r \min\{1, cp_i\} \leq c$, since $\sum_{i=1}^r p_i = 1$. Therefore, we have

$$
\begin{aligned}
\frac{1}{\delta}\sum_{i=1}^r \left(\frac{1}{q_i} - 1\right)\|\boldsymbol{u}_i\|^2 z_i^2 &\leq \frac{1}{\delta}\sum_{\substack{i=1 \\ q_i < 1}}^r \frac{\|\boldsymbol{u}_i\|^2 z_i^2}{q_i} \\
&= \frac{1}{c\delta}\sum_{\substack{i=1 \\ q_i < 1}}^r \frac{\|\boldsymbol{u}_i\|^2 z_i^2}{p_i} \\
&\leq \frac{1}{c\delta}\sum_{i=1}^r \frac{\|\boldsymbol{u}_i\|^2 z_i^2}{p_i} \\
&\leq \frac{\gamma(\boldsymbol{z})}{m_0\delta} \leq \varepsilon^2,
\end{aligned}
\tag{A.80}
$$

where we use the fact that $q_i = cp_i$ when $q_i < 1$ in the second line, the fact that the terms are non-negative in the third line, and the definition of $\gamma$ (see (2.3)) and $c \geq m_0$ in the third line. The result then follows from Theorem A.12. $\qquad\square$

**Remark A.14.** *We can derive a tighter bound on the expected sample complexity for sampling without replacement as*

$$m_0 \geq \frac{1}{\delta\varepsilon^2}\sum_{\substack{i=1 \\ q_i < 1}}^r \frac{\|\boldsymbol{u}_i\|^2 z_i^2}{p_i}.$$

*Intuitively, when $q_i = 1$, we (deterministically) choose the $i$th row of $\boldsymbol{U}$, and therefore, it should not add to the estimation error, which is then reflected in the average sample complexity. Thus, in practice, we may observe a somewhat smaller error for sampling without replacement on an average, compared to sampling with replacement.*

### A.5.2 Subspace Embedding

In this section, we derive a subspace embedding guarantee for sampling without replacement.

**Theorem A.15** (Subspace embedding). *Let $\boldsymbol{U}$ be an $r \times d$ matrix with rows $\boldsymbol{u}_1, \ldots, \boldsymbol{u}_r$, and let $\mathcal{P} = (p_1, \ldots, p_r)$ be a probability distribution on $[r]$ with $p_i > 0$ for all $i \in [r]$. Given a number $m_0 \in (0, r]$, let $c > 0$ be a constant for which $q_i = \min\{1, cp_i\}$ for $i \in [r]$ and $\sum_{i=1}^r q_i = m_0$. Then, if*

$$m_0 \geq \frac{2}{\varepsilon^2}\left\|\sum_{i=1}^r \frac{\|\boldsymbol{u}_i\|^2}{p_i}\boldsymbol{u}_i\boldsymbol{u}_i^\mathsf{T}\right\|\log\left(\frac{d}{\delta}\right) + \frac{2}{3\varepsilon}\max_{i\in[r]}\frac{\|\boldsymbol{u}_i\|^2}{p_i}\log\left(\frac{d}{\delta}\right),$$

*by sampling the rows of $\boldsymbol{U}$ without replacement according to probabilities $q_1, \ldots, q_r$, it holds that*

$$\mathbb{P}\big[\big\|\boldsymbol{U}^\mathsf{T}\boldsymbol{S}^\mathsf{T}\boldsymbol{S}\boldsymbol{U} - \boldsymbol{U}^\mathsf{T}\boldsymbol{U}\big\| \leq \varepsilon\big] \geq 1 - \delta.$$

*Proof.* Let $Y_1, \ldots, Y_r$ be independent Bernoulli random variables with means $q_1, \ldots, q_r$ respectively. For $i \in [r]$, define the random variable

$$\boldsymbol{X}_i = \frac{Y_i}{q_i}\boldsymbol{u}_i\boldsymbol{u}_i^\mathsf{T} - \boldsymbol{u}_i\boldsymbol{u}_i^\mathsf{T}.\tag{A.81}$$

If $q_i = 1$, then $Y_i = 1$, so that $\boldsymbol{X}_i = 0$. Therefore, we have

$$\|\boldsymbol{X}_i\| \leq \max_{\substack{i \in [r] \\ q_i < 1}} \left| \frac{Y_i}{q_i} - 1 \right| \|\boldsymbol{u}_i\|^2 \tag{A.82}$$

$$\leq \max_{\substack{i \in [r] \\ q_i < 1}} \frac{\|\boldsymbol{u}_i\|^2}{q_i} \tag{A.83}$$

$$= \frac{1}{c} \max_{\substack{i \in [r] \\ q_i < 1}} \frac{\|\boldsymbol{u}_i\|^2}{p_i} \tag{A.84}$$

$$\leq \frac{1}{c} \max_{i \in [r]} \frac{\|\boldsymbol{u}_i\|^2}{p_i} \tag{A.85}$$

$$\leq \frac{1}{m_0} \max_{i \in [r]} \frac{\|\boldsymbol{u}_i\|^2}{p_i} =: \frac{L}{m_0} \tag{A.86}$$

for all $i \in [r]$. Here, the third line follows from the fact that $q_i = cp_i$ when $q_i < 1$, while the last line follows from the fact that $m_0 = \sum_{i=1}^r \min\{1, cp_i\} \leq c$ since $\sum_{i=1}^r p_i = 1$.

Next, note that $\mathbb{E}[\boldsymbol{X}_i] = 0$ and $\sum_{i=1}^r \boldsymbol{X}_i = \boldsymbol{U}^\mathsf{T} \boldsymbol{S}^\mathsf{T} \boldsymbol{S} \boldsymbol{U} - \boldsymbol{U}^\mathsf{T} \boldsymbol{U}$. Furthermore, we have

$$\mathbb{E}[\boldsymbol{X}_i^2] = \left( q_i \left( 1 - \frac{1}{q_i} \right)^2 + (1 - q_i) \right) \|\boldsymbol{u}_i\|^2 \boldsymbol{u}_i \boldsymbol{u}_i^\mathsf{T} = \frac{(1 - q_i)}{q_i} \|\boldsymbol{u}_i\|^2 \boldsymbol{u}_i \boldsymbol{u}_i^\mathsf{T} \tag{A.87}$$

for all $i \in [r]$. Therefore, we have

$$\sum_{i=1}^r \mathbb{E}[\boldsymbol{X}_i^2] \preceq \sum_{\substack{i=1 \\ q_i < 1}}^r \frac{\|\boldsymbol{u}_i\|^2}{q_i} \boldsymbol{u}_i \boldsymbol{u}_i^\mathsf{T} \tag{A.88}$$

$$= \frac{1}{c} \sum_{\substack{i=1 \\ q_i < 1}}^r \frac{\|\boldsymbol{u}_i\|^2}{p_i} \boldsymbol{u}_i \boldsymbol{u}_i^\mathsf{T} \tag{A.89}$$

$$\preceq \frac{1}{c} \sum_{i=1}^r \frac{\|\boldsymbol{u}_i\|^2}{p_i} \boldsymbol{u}_i \boldsymbol{u}_i^\mathsf{T} \tag{A.90}$$

$$\preceq \frac{1}{m_0} \sum_{i=1}^r \frac{\|\boldsymbol{u}_i\|^2}{p_i} \boldsymbol{u}_i \boldsymbol{u}_i^\mathsf{T}. \tag{A.91}$$

It follows that

$$\left\| \sum_{i=1}^r \mathbb{E}[\boldsymbol{X}_i^2] \right\| \leq \frac{1}{m_0} \left\| \sum_{i=1}^r \frac{\|\boldsymbol{u}_i\|^2}{p_i} \boldsymbol{u}_i \boldsymbol{u}_i^\mathsf{T} \right\| =: \frac{\sigma^2}{m_0}. \tag{A.92}$$

The result then follows from Imported Theorem A.3. $\qquad\square$

### A.5.3 Sketched Regression

We now combine approximate matrix-vector multiplication guarantee (Corollary A.13) and subspace embedding guarantee (Theorem A.15) to obtain guarantee for the sketched regression estimator constructed by sampling without replacement.

**Theorem A.16** (Sketched Regression). *Suppose $\boldsymbol{U}$ has orthonormal columns and let $\boldsymbol{y}^* = \operatorname{argmin}_{\boldsymbol{y}} \|\boldsymbol{U}\boldsymbol{y} - \boldsymbol{b}\|^2$. Let $\mathcal{P} = (p_1, \ldots, p_r)$ be a probability distribution on $[r]$ with $p_i > 0$ for all $i \in [r]$. Given a number $m_0 \in (0, r]$, let $c > 0$ be a constant for which $q_i = \min\{1, cp_i\}$ for $i \in [r]$ and $\sum_{i=1}^r q_i = m_0$. Let $\boldsymbol{S}$ be a $m \times q$ sketching matrix obtained by sampling rows of $\boldsymbol{U}$ without replacement according to probabilities $q_1, \ldots, q_r$. Define*

$$\widehat{\boldsymbol{y}} = \operatorname*{argmin}_{\boldsymbol{y}} \|\boldsymbol{S}\boldsymbol{U}\boldsymbol{y} - \boldsymbol{S}\boldsymbol{b}\|^2.$$

*Then, if*

$$m_0 = O\left(\frac{\gamma(\boldsymbol{P_U b})}{\delta \varepsilon^2} + \eta \log\left(\frac{d}{\delta}\right)\right),$$

*it holds that*

$$\mathbb{P}[\|\boldsymbol{y}^* - \widehat{\boldsymbol{y}}\| \leq \varepsilon] \geq 1 - \delta.$$

*Proof.* We closely follow the proof of Theorem A.5. Since $\boldsymbol{y}^*$ is the solution of $\min_{\boldsymbol{y}}\|\boldsymbol{Uy} - \boldsymbol{b}\|^2$, $\boldsymbol{Uy}^* - \boldsymbol{b}$ lies in the orthogonal complement of the range of $\boldsymbol{U}$, and therefore, $\boldsymbol{U}^\mathsf{T}(\boldsymbol{Uy}^* - \boldsymbol{b}) = 0$. Then, taking $\boldsymbol{z} = \boldsymbol{Uy}^* - \boldsymbol{b} = (\boldsymbol{I} - \boldsymbol{UU}^\mathsf{T})\boldsymbol{b}$ in Corollary A.13, we can infer that using

$$m_0 = O\left(\frac{\gamma(\boldsymbol{z})}{\delta \varepsilon^2}\right), \tag{A.93}$$

we have with probability exceeding $1 - \delta/2$,

$$\left\|\boldsymbol{U}^\mathsf{T}\boldsymbol{S}^\mathsf{T}\boldsymbol{S}(\boldsymbol{Uy}^* - \boldsymbol{b})\right\| \leq \frac{\varepsilon}{2}. \tag{A.94}$$

Next, note that $\eta = \max_i \|u_i\|^2/p_i$, so that

$$\sum_{i=1}^r \frac{\|\boldsymbol{u}_i\|^2}{p_i} \boldsymbol{u}_i \boldsymbol{u}_i^\mathsf{T} \preceq \eta \sum_{i=1}^r \boldsymbol{u}_i \boldsymbol{u}_i^\mathsf{T} = \eta \boldsymbol{U}^\mathsf{T}\boldsymbol{U}. \tag{A.95}$$

Then, because $\boldsymbol{U}$ has orthonormal columns, we have

$$\left\|\sum_{i=1}^r \frac{\|\boldsymbol{u}_i\|^2}{p_i} \boldsymbol{u}_i \boldsymbol{u}_i^\mathsf{T}\right\| \leq \eta\|\boldsymbol{U}^\mathsf{T}\boldsymbol{U}\| = \eta. \tag{A.96}$$

Therefore, by Theorem A.15, if

$$m_0 = O\left(\eta \log\left(\frac{d}{\delta}\right)\right) \tag{A.97}$$

$$\geq 8\left\|\sum_{i=1}^r \frac{\|\boldsymbol{u}_i\|^2}{p_i} \boldsymbol{u}_i \boldsymbol{u}_i^\mathsf{T}\right\| \log\left(\frac{d}{\delta}\right) + \frac{4}{3} \max_{i \in [r]} \frac{\|\boldsymbol{u}_i\|^2}{p_i} \log\left(\frac{d}{\delta}\right), \tag{A.98}$$

then, with probability exceeding $1 - \delta/2$,

$$\left\|\boldsymbol{U}^\mathsf{T}\boldsymbol{S}^\mathsf{T}\boldsymbol{S}\boldsymbol{U} - \boldsymbol{I}\right\| \leq 1/2. \tag{A.99}$$

The remainder of the proof is the same as that of Theorem A.5. □

# B   Description of past estimators

In this section we provide more details on how several existing estimators fit into the unified framework described in Section 2.

## B.1   KernelSHAP

KernelSHAP makes use of a subsampled and reweighted version of the constrained regression formulation (1.2) of the Shapley values. Specifically, denoting $\boldsymbol{Z}_S$ to be the $S$-th row of $\boldsymbol{Z}$, observe that

$$\|\boldsymbol{Z}'\boldsymbol{\phi} - \boldsymbol{b}\|^2 = \sum_{S \in 2^{[d]}\setminus\{[d],\emptyset\}} k(S)(\boldsymbol{Z}_S\boldsymbol{\phi} - \boldsymbol{v}_S)^2$$

$$= \left[\sum_{S \in 2^{[d]}\setminus\{[d],\emptyset\}} k(S)\right] \mathbb{E}[(\boldsymbol{Z}_{S'}\boldsymbol{\phi} - \boldsymbol{v}_{S'})^2], \tag{B.1}$$

where in the last equation $S'$ is a random variable for which $\mathbb{P}[S' = S] \propto k(S)$ for $S \subseteq [d]$, $S \neq \emptyset, [d]$. Note that

$$\underset{\substack{\phi \in \mathbb{R}^d \\ \mathbf{1}^\top \phi = v([d]) - v(\emptyset)}}{\operatorname{argmin}} \|\mathbf{Z}'\phi - \mathbf{b}\|^2 = \underset{\substack{\phi \in \mathbb{R}^d \\ \mathbf{1}^\top \phi = v([d]) - v(\emptyset)}}{\operatorname{argmin}} \mathbb{E}[(\mathbf{Z}_{S'}\phi - \mathbf{v}_{S'})^2],$$

because the minima of a function $f$ coincide with the minima of $\zeta f$ for $\zeta > 0$.

The KernelSHAP estimator [LL17] is then defined as

$$\phi^{\text{KS}} = \underset{\substack{\phi \in \mathbb{R}^d \\ \mathbf{1}^\top \phi = v([d]) - v(\emptyset)}}{\operatorname{argmin}} \frac{1}{m} \sum_{i=1}^{m} (\mathbf{Z}_{S_i}\phi - \mathbf{v}_{S_i})^2, \tag{B.2}$$

where $S_i$ are iid copies of $S'$.

As noted by [MW25], this can be viewed as a constrained sketched regression problem

$$\phi^{\text{KS}} = \underset{\substack{\phi \in \mathbb{R}^d \\ \mathbf{1}^\top \phi = v([d]) - v(\emptyset)}}{\operatorname{argmin}} \|\mathbf{S}(\mathbf{Z}'\phi - \mathbf{b})\|^2 \tag{B.3}$$

Performing the same change of variables as in the proof of Theorem 2.1 we find that

$$\phi^{\text{KS}} = \mathbf{Q} \underset{\mathbf{x} \in \mathbb{R}^{d-1}}{\operatorname{argmin}} \|\mathbf{S}(\mathbf{Z}'(\mathbf{Q}\mathbf{x} + \alpha\mathbf{1}) - \mathbf{b})\|^2 + \alpha\mathbf{1} \tag{B.4}$$

$$= \mathbf{Q} \underset{\mathbf{x} \in \mathbb{R}^{d-1}}{\operatorname{argmin}} \|\mathbf{S}(\mathbf{Z}'\mathbf{Q}\mathbf{x} - (\mathbf{b} - \alpha\mathbf{Z}'\mathbf{1}))\|^2 + \alpha\mathbf{1} \tag{B.5}$$

$$= \mathbf{Q} \underset{\mathbf{x} \in \mathbb{R}^{d-1}}{\operatorname{argmin}} \|\mathbf{S}(\mathbf{U}\mathbf{x} - \mathbf{b}_\alpha)\|^2 + \alpha\mathbf{1}. \tag{B.6}$$

## B.2 Unbiased KernelSHAP

In [CL20], the authors observe that the Shapley values can be expressed as

$$\phi^* = \mathbf{A}^{-1}\left(\mathbf{f} - \mathbf{1}\frac{\mathbf{1}^\top \mathbf{A}^{-1}\mathbf{f} - v([d]) + v(\emptyset)}{\mathbf{1}^\top \mathbf{A}^{-1}\mathbf{1}}\right) \tag{B.7}$$

where

$$\mathbf{A} = \mathbf{Z}^\top \mathbf{W}\mathbf{Z}, \qquad \mathbf{f} = \mathbf{Z}^\top \mathbf{W}\mathbf{b}. \tag{B.8}$$

They then introduce the *unbiased KernelSHAP* estimator

$$\phi^{\text{uKS}} = \mathbf{A}^{-1}\left(\widehat{\mathbf{f}} - \mathbf{1}\frac{\mathbf{1}^\top \mathbf{A}^{-1}\widehat{\mathbf{f}} - v([d]) + v(\emptyset)}{\mathbf{1}^\top \mathbf{A}^{-1}\mathbf{1}}\right), \qquad \widehat{\mathbf{f}} = \mathbf{Z}^\top \sqrt{\mathbf{W}}\mathbf{S}^\top \mathbf{S}\sqrt{\mathbf{W}}\mathbf{b}. \tag{B.9}$$

Expanding, we see that

$$\phi^{\text{uKS}} = \mathbf{A}^{-1}\widehat{\mathbf{f}} - \frac{\mathbf{A}^{-1}\mathbf{1}\mathbf{1}^\top \mathbf{A}^{-1}}{\mathbf{1}^\top \mathbf{A}^{-1}\mathbf{1}}\widehat{\mathbf{f}} + \mathbf{A}^{-1}\mathbf{1}\frac{v([d]) + v(\emptyset)}{\mathbf{1}^\top \mathbf{A}^{-1}\mathbf{1}} \tag{B.10}$$

Since $[\mathbf{Q}, d^{-1/2}\mathbf{1}]$ form an orthonormal basis for $\mathbb{R}^d$,

$$\mathbf{A}^{-1} = \left(\frac{d-1}{d}\mathbf{Q}\mathbf{Q}^\top + (d - 1 + dc_d)\frac{\mathbf{1}\mathbf{1}^\top}{d}\right)^{-1} = \frac{d}{d-1}\mathbf{Q}\mathbf{Q}^\top + (d - 1 + dc_d)^{-1}\frac{\mathbf{1}\mathbf{1}^\top}{d}. \tag{B.11}$$

Using this, we see that

$$\mathbf{A}^{-1}\mathbf{1} = (d - 1 + dc_d)^{-1}\mathbf{1}, \qquad \mathbf{1}^\top \mathbf{A}^{-1}\mathbf{1} = d(d - 1 + dc_d)^{-1}. \tag{B.12}$$

We now compute

$$\mathbf{A}^{-1}\widehat{\mathbf{f}} = \frac{d}{d-1}\mathbf{Q}\mathbf{Q}^\top\widehat{\mathbf{f}} + (d - 1 + dc_d)^{-1}\frac{\mathbf{1}\mathbf{1}^\top}{d}\widehat{\mathbf{f}}, \tag{B.13}$$

$$\frac{\mathbf{A}^{-1}\mathbf{1}\mathbf{1}^\top \mathbf{A}^{-1}}{\mathbf{1}^\top \mathbf{A}^{-1}\mathbf{1}} = \frac{(d - 1 + dc_d)^{-2}\mathbf{1}\mathbf{1}^\top}{d(d - 1 + dc_d)^{-1}} = \frac{(d - 1 + dc_d)^{-1}\mathbf{1}\mathbf{1}^\top}{d}, \tag{B.14}$$

and

$$\mathbf{A}^{-1}\mathbf{1}\frac{v([d]) + v(\emptyset)}{\mathbf{1}^\top \mathbf{A}^{-1}\mathbf{1}} = \frac{(d - 1 + dc_d)^{-1}\mathbf{1}}{d(d - 1 + dc_d)^{-1}} = \frac{\mathbf{1}}{d}. \tag{B.15}$$

Combining these equations we have

$$\phi^{\text{uKS}} = \frac{d}{d-1}\mathbf{Q}\mathbf{Q}^\top \mathbf{Z}^\top \sqrt{\mathbf{W}}\mathbf{S}^\top \mathbf{S}\sqrt{\mathbf{W}}\mathbf{b} + \frac{v([d]) - v(\emptyset)}{d}\mathbf{1}. \tag{B.16}$$

### B.3 LeverageSHAP

In [MW25], the authors show the typical formulation of the Shapley values (1.2) can be rewritten as an unconstrained problem

$$\phi^* = \operatorname*{argmin}_{\phi \in \mathbb{R}^d} \|A\phi - b_\alpha\|^2 + \alpha\mathbf{1}, \tag{B.17}$$

where

$$A = Z'P, \qquad P := I - d^{-1}\mathbf{1}\mathbf{1}^\mathsf{T} = QQ^\mathsf{T}.^8 \tag{B.18}$$

They then describe a randomized estimator *LeverageSHAP* of the form

$$\phi^{\mathrm{LS}} = \operatorname*{argmin}_{\phi \in \mathbb{R}^d} \|S(A\phi - b_\alpha)\|^2 + \alpha\mathbf{1}. \tag{B.19}$$

Theoretical guarantees are given for the case where $S$ is drawn according to the leverage scores of $A$.

## C   Equivalence between Lagrangian and Change of Variable Framework

We consider,

$$\phi^{\mathrm{R}} = \operatorname*{argmin}_{\substack{\phi \in \mathbb{R}^d \\ \mathbf{1}^\mathsf{T}\phi = \alpha}} \|C\phi - y\|^2. \tag{C.1}$$

where $\alpha = (v([d]) - v(\emptyset))/d$, and $C = Z'$ and $y = b$ for solving the constrained least squares exactly, while $C = SZ'$ and $y = Sb$ for approximately methods such that $E[S^\mathsf{T}S] = I$. Define $M = C^\mathsf{T}C$ and $g = C^\mathsf{T}y$. Next, we write the unconstrained solution of the above least squares as,

$$\phi^u = \operatorname*{argmin}_{\phi \in \mathbb{R}^d} \|C\phi - y\|^2 = M^+g \tag{C.2}$$

**Lagrangian method:**   In order to solve (C.1), the Lagrangian method writes,

$$\mathcal{L}(\phi, \lambda) = \frac{1}{2}\phi^\mathsf{T}M\phi - g^\mathsf{T}\phi + \lambda(\mathbf{1}^\mathsf{T}\phi - \alpha) \tag{C.3}$$

with the following KKT conditions,

1. $M\phi = g + \lambda\mathbf{1} = 0 \rightarrow \phi = M^+(g - \lambda\mathbf{1})$
2. $\mathbf{1}^\mathsf{T}\phi = \alpha \rightarrow \lambda = \frac{\mathbf{1}^\mathsf{T}M^+g - \alpha d}{\mathbf{1}^\mathsf{T}M^+\mathbf{1}}$

This results in the final solution to be,

$$\phi^{\mathrm{R}} = \phi^u - M^+\mathbf{1}\frac{\mathbf{1}^\mathsf{T}\phi^u - \alpha d}{\mathbf{1}^\mathsf{T}M^+\mathbf{1}} \tag{C.4}$$

**Change of Variable Method:**   As discussed in Appendix A.3, an alternative method to solve the constrained least squares is using the change of variable to explicitly enforce the constraint. Specifically, we re-parameterize $\phi$ as,

$$\phi = \alpha\mathbf{1} + Qx \tag{C.5}$$

where $Q \in \mathbb{R}^{d \times (d-1)}$ is a matrix with columns forming an orthonormal basis for the null space of $\mathbf{1}^\mathsf{T}$, i.e., $\mathbf{1}^\mathsf{T}Q = 0$ and $Q^\mathsf{T}Q = I$.

Plugging $\phi$ into the objective results in,

$$\operatorname*{argmin}_{x} \|C(\alpha\mathbf{1} + Qx) - y\|^2 = \operatorname*{argmin}_{x} \|CQx - (y - \alpha C\mathbf{1})\|^2 \tag{C.6}$$

Solving this results in,

$$x^* = (Q^\mathsf{T}M^\mathsf{T}Q)^+Q^TM(\phi^u - \alpha\mathbf{1}) \tag{C.7}$$

This the final solution is,

$$\phi^{\mathrm{R}} = u + Q(QM^\mathsf{T}Q)^+Q^TM(\phi^u - \alpha\mathbf{1}) \tag{C.8}$$

---

[8]Since $\mathbf{1}$ is in the null-space of $A$, all of $\{\phi + c\mathbf{1} : c \in \mathbb{R}\}$ produce the same objective value (and hence the argmin is an infinite set), it should be understood as the minimum norm solution; i.e. for which $\phi + c\mathbf{1}$ is orthogonal to $\mathbf{1}$.

**Equivalence of the methods:** The second term in (C.8) can be seen as a projection of the vector $\phi^u - \alpha\mathbf{1}$ into the span of $Q$ (or alternatively on the null space of $\mathbf{1}^\top$) with the projection matrix,

$$P = Q(QM^\top Q)^+ Q^T M \tag{C.9}$$

Next, we can rewrite (C.8) as,

$$\phi^{\text{R}} = \alpha\mathbf{1} + P(\phi^u - \alpha\mathbf{1}) = \phi^u - (I - P)(\phi^u - \alpha\mathbf{1}) \tag{C.10}$$

From the geometric intuition, $I - P$ can be seen as a metric-projection in the $M$-norm[9] into the orthogonal complement $Q$, or alternatively in the span of $M^+\mathbf{1}$. Such a projection in the $M$-norm for any vector $z$ is

$$(I - P)(z) = M^+\mathbf{1}\frac{\mathbf{1}^\top z}{\mathbf{1}^\top M^+\mathbf{1}}. \tag{C.11}$$

Thus, plugging in this (C.11) results in

$$\phi^{\text{R}} = \phi^u - M^+\mathbf{1}\frac{\mathbf{1}^\top(\phi^u - \alpha\mathbf{1})}{\mathbf{1}^\top M^+\mathbf{1}}, \tag{C.12}$$

thus recovering (C.4) by noting that $\mathbf{1}^\top\mathbf{1} = d$.

# D    Ratio of mean squared errors

In Appendix A.2, we saw that $\gamma(z)$ and $\eta$ (see (2.3)) give *upper* bounds on the sample complexity of matrix-vector multiplication and regression estimators for sampling with replacement. In this section, we study the ratio of mean squared errors for different sampling strategies these estimators in the finite-sample/asymptotic regime. We find that this ratio is determined by $\gamma$ for both these estimators, as summarized below.

**Theorem D.1** (Ratio of mean squared errors)**.** *Given an $r \times q$ matrix $U$ with orthonormal columns, and an $r$-dimensional vector $b$, suppose that we want to estimate $U^\top b$ using matrix-vector multiplication estimator (see Theorem A.1) and $\operatorname{argmin}_y\|Uy - b\|^2$ using a regression estimator (see Theorem A.5). Given a sampling distribution $\mathcal{P}$ over $[r]$ and a fixed number of samples $m$, denote $X_m^{\text{M}}(\mathcal{P})$, $X_m^{\text{R}}(\mathcal{P})$ to be the matrix-vector multiplication estimator and regression estimator for $U^\top b$, respectively.*

*Given two sampling distributions $\mathcal{P}_1$ and $\mathcal{P}_2$, denote $\gamma_1$ and $\gamma_2$ to be the values of $\gamma$ as defined in (2.3) with respect to distributions $\mathcal{P}_1$ and $\mathcal{P}_2$, respectively. Fix the number of samples $m \in \mathbb{N}$. Then, we have the following results.*

1. *(**Matrix-Vector Multiplication**)*

$$\frac{\mathbb{E}[\|X_m^{\text{M}}(\mathcal{P}_1) - U^\top b\|^2]}{\mathbb{E}[\|X_m^{\text{M}}(\mathcal{P}_2) - U^\top b\|^2]} = \frac{\gamma_1(b) - \|U^\top b\|^2}{\gamma_2(b) - \|U^\top b\|^2}. \tag{D.1}$$

2. *(**Regression**) If for $i = 1, 2$, $\mathbb{E}[\|X_m^{\text{R}}(\mathcal{P}_i) - U^\top b\|^2] \neq 0$ and*

$$\frac{\sqrt{\mathbb{E}[\|X_m^{\text{R}}(\mathcal{P}_i) - U^\top b\|^4]}}{\mathbb{E}[\|X_m^{\text{R}}(\mathcal{P}_i) - U^\top b\|^2]} = O(1), \tag{D.2}$$

   *we have*

$$\frac{\mathbb{E}[\|X_m^{\text{R}}(\mathcal{P}_1) - U^\top b\|^2]}{\mathbb{E}[\|X_m^{\text{R}}(\mathcal{P}_2) - U^\top b\|^2]} = \left(1 \pm O\left(\frac{1}{\sqrt{m}}\right)\right)\frac{\gamma_1(P_U b)}{\gamma_2(P_U b)}, \tag{D.3}$$

   *where $x = (a \pm b)$ means $x \in [a - b, a + b]$.*

*Proof.* Let $S$ be an $m \times r$ sketch matrix (for sampling with replacement) as defined in (A.2) with respect to the distribution $\mathcal{P}$. Then, the matrix-vector multiplication estimator is $X_m^{\text{M}}(\mathcal{P}) = U^\top S^\top Sb$, while the regression estimator is $X_m^{\text{R}}(\mathcal{P}) = \operatorname{argmin}_y\|SUy - Sb\|^2$.

---

[9]where $M$-norm is defined as $\|v\|_M = v^\top Mv$ for all $v \in \mathbb{R}^d$

1. **Matrix-Vector Multiplication:** Since $\mathbb{E}[\|\boldsymbol{X}_m^{\mathrm{M}}(\mathcal{P}) - \boldsymbol{U}^\mathsf{T}\boldsymbol{b}\|^2]$ is the variance of $\boldsymbol{X}_m^{\mathrm{M}}$ using $m$ samples, from (A.4), we have

$$\mathbb{E}[\|\boldsymbol{X}_m^{\mathrm{M}}(\mathcal{P}_i) - \boldsymbol{U}^\mathsf{T}\boldsymbol{b}\|^2] = \frac{1}{m}(\gamma_i(\boldsymbol{b}) - \|\boldsymbol{U}^\mathsf{T}\boldsymbol{b}\|^2) \tag{D.4}$$

for $i = 1, 2$, from which we obtain (D.1).

2. **Regression:** Observe that $\operatorname{argmin}_{\boldsymbol{y}}\|\boldsymbol{U}\boldsymbol{y} - \boldsymbol{b}\|^2 = \boldsymbol{U}^\mathsf{T}\boldsymbol{b}$. Furthermore, since

$$\|\boldsymbol{X}_m^{\mathrm{R}}(\mathcal{P}_i) - \boldsymbol{U}^\mathsf{T}\boldsymbol{b}\| = \|\boldsymbol{U}^\mathsf{T}\boldsymbol{S}^\mathsf{T}\boldsymbol{S}\boldsymbol{U}(\boldsymbol{X}_m^{\mathrm{R}}(\mathcal{P}_i) - \boldsymbol{U}^\mathsf{T}\boldsymbol{b}) + (\boldsymbol{I} - \boldsymbol{U}^\mathsf{T}\boldsymbol{S}^\mathsf{T}\boldsymbol{S}\boldsymbol{U})(\boldsymbol{X}_m^{\mathrm{R}}(\mathcal{P}_i) - \boldsymbol{U}^\mathsf{T}\boldsymbol{b})\|, \tag{D.5}$$

we have from triangle and reverse-triangle inequalities,

$$\left|\|\boldsymbol{X}_m^{\mathrm{R}}(\mathcal{P}_i) - \boldsymbol{U}^\mathsf{T}\boldsymbol{b}\| - \|\boldsymbol{U}^\mathsf{T}\boldsymbol{S}^\mathsf{T}\boldsymbol{S}\boldsymbol{U}(\boldsymbol{X}_m^{\mathrm{R}}(\mathcal{P}_i) - \boldsymbol{U}^\mathsf{T}\boldsymbol{b})\|\right| \leq \|(\boldsymbol{I} - \boldsymbol{U}^\mathsf{T}\boldsymbol{S}^\mathsf{T}\boldsymbol{S}\boldsymbol{U})(\boldsymbol{X}_m^{\mathrm{R}}(\mathcal{P}_i) - \boldsymbol{U}^\mathsf{T}\boldsymbol{b})\|. \tag{D.6}$$

For simplicity, denote $A_i = \|\boldsymbol{X}_m^{\mathrm{R}}(\mathcal{P}_i) - \boldsymbol{U}^\mathsf{T}\boldsymbol{b}\|$, $B_i = \|\boldsymbol{U}^\mathsf{T}\boldsymbol{S}^\mathsf{T}\boldsymbol{S}\boldsymbol{U}(\boldsymbol{X}_m^{\mathrm{R}}(\mathcal{P}_i) - \boldsymbol{U}^\mathsf{T}\boldsymbol{b})\|$, and $C_i = \|(\boldsymbol{I} - \boldsymbol{U}^\mathsf{T}\boldsymbol{S}^\mathsf{T}\boldsymbol{S}\boldsymbol{U})(\boldsymbol{X}_m^{\mathrm{R}}(\mathcal{P}_i) - \boldsymbol{U}^\mathsf{T}\boldsymbol{b})\|$. Then, we have $|A_i^2 - B_i^2| \leq (A_i + B_i)C_i$, from which it follows that

$$|\mathbb{E}[A_i^2] - \mathbb{E}[B_i^2]| \leq \mathbb{E}[(A_i + B_i)C_i]. \tag{D.7}$$

Now, observe that $B_i \leq \|\boldsymbol{U}^\mathsf{T}\boldsymbol{S}^\mathsf{T}\boldsymbol{S}\boldsymbol{U}\|A_i$ and $C_i \leq \|\boldsymbol{I} - \boldsymbol{U}^\mathsf{T}\boldsymbol{S}^\mathsf{T}\boldsymbol{S}\boldsymbol{U}\|A_i$. Moreover, we have $\|\boldsymbol{U}^\mathsf{T}\boldsymbol{S}^\mathsf{T}\boldsymbol{S}\boldsymbol{U}\| \leq \eta_i$, where $\eta_i$ is defined in (2.3) (and depends on the distribution $\mathcal{P}_i$). Therefore,

$$|\mathbb{E}[A_i^2] - \mathbb{E}[B_i^2]| \leq (1 + \eta_i)\mathbb{E}[\|\boldsymbol{I} - \boldsymbol{U}^\mathsf{T}\boldsymbol{S}^\mathsf{T}\boldsymbol{S}\boldsymbol{U}\|A_i^2] \leq (1 + \eta_i)\sqrt{\mathbb{E}[\|\boldsymbol{I} - \boldsymbol{U}^\mathsf{T}\boldsymbol{S}^\mathsf{T}\boldsymbol{S}\boldsymbol{U}\|^2]}\sqrt{\mathbb{E}[A_i^4]}, \tag{D.8}$$

where we used Cauchy-Schwarz inequality in the last step. Now, note that $\sqrt{\mathbb{E}[A_i^4]} \geq \mathbb{E}[A_i^2]$ by Jensen's inequality, and thus, $\sqrt{\mathbb{E}[A_i^4]}/\mathbb{E}[A_i^2] = O(1)$ implies $\sqrt{\mathbb{E}[A_i^4]}/\mathbb{E}[A_i^2] = \Theta(1)$. It follows from (A.9) and (A.15) that

$$\left|\frac{\mathbb{E}[B_i^2]}{\mathbb{E}[A_i^2]} - 1\right| \leq (1 + \eta_i)\sqrt{\frac{\eta_i}{m}}\frac{\sqrt{\mathbb{E}[A_i^4]}}{\mathbb{E}[A_i^2]} = \Theta\left(\frac{1}{\sqrt{m}}\right). \tag{D.9}$$

Thus, for large enough $m$ (using $(1 - x)^{-1} = 1 + O(x)$ for $x \ll 1$), we have

$$\frac{\mathbb{E}[A_1^2]}{\mathbb{E}[B_1^2]} = 1 \pm O\left(\frac{1}{\sqrt{m}}\right) \tag{D.10}$$

and

$$\frac{\mathbb{E}[B_2^2]}{\mathbb{E}[A_2^2]} = 1 \pm O\left(\frac{1}{\sqrt{m}}\right), \tag{D.11}$$

which implies

$$\frac{\mathbb{E}[A_1^2]}{\mathbb{E}[A_2^2]} = \left(1 \pm O\left(\frac{1}{\sqrt{m}}\right)\right)\frac{\mathbb{E}[B_1^2]}{\mathbb{E}[B_2^2]}. \tag{D.12}$$

Then, denoting $\boldsymbol{y}^* = \operatorname{argmin}_{\boldsymbol{y}}\|\boldsymbol{U}\boldsymbol{y} - \boldsymbol{b}\|^2$, from (A.24) and (A.4), we obtain

$$\frac{\mathbb{E}[B_1^2]}{\mathbb{E}[B_2^2]} = \frac{\mathbb{E}[\|\boldsymbol{U}^\mathsf{T}\boldsymbol{S}^\mathsf{T}\boldsymbol{S}(\boldsymbol{U}\boldsymbol{y}^* - \boldsymbol{b})\|^2]}{\mathbb{E}[\|\boldsymbol{U}^\mathsf{T}\boldsymbol{S}^\mathsf{T}\boldsymbol{S}(\boldsymbol{U}\boldsymbol{y}^* - \boldsymbol{b})\|^2]} \tag{D.13}$$

$$= \frac{\gamma_1(\boldsymbol{U}\boldsymbol{y}^* - \boldsymbol{b}) - \|\boldsymbol{U}^\mathsf{T}(\boldsymbol{U}\boldsymbol{y}^* - \boldsymbol{b})\|^2}{\gamma_2(\boldsymbol{U}\boldsymbol{y}^* - \boldsymbol{b}) - \|\boldsymbol{U}^\mathsf{T}(\boldsymbol{U}\boldsymbol{y}^* - \boldsymbol{b})\|^2} \tag{D.14}$$

$$= \frac{\gamma_1(\boldsymbol{P}_{\boldsymbol{U}}\boldsymbol{b})}{\gamma_2(\boldsymbol{P}_{\boldsymbol{U}}\boldsymbol{b})}, \tag{D.15}$$

where in the last step, we use the fact that $\boldsymbol{U}\boldsymbol{y}^* - \boldsymbol{b} = \boldsymbol{P}_{\boldsymbol{U}}\boldsymbol{b}$ and $\boldsymbol{U}^\mathsf{T}\boldsymbol{P}_{\boldsymbol{U}}\boldsymbol{b} = 0$. $\qquad\square$

Informally, (D.2) says that (the square-root of) the fourth "central moment" is comparable to the mean squared error of the estimator. This requirement actually holds for the simple statistical task of

estimating the mean of a scalar random variable. Indeed, if $X_1, \ldots, X_m$ are iid copies of a random variable $X$ with $\mathbb{E}[X^4] < \infty$, then $\widehat{X} = \sum_{i=1}^{m} X_i / m$ is an unbiased estimator of $\mathbb{E}[X]$ satisfying

$$\frac{\sqrt{\mathbb{E}[(\widehat{X} - \mathbb{E}[X])^4]}}{\mathbb{E}[(\widehat{X} - \mathbb{E}[X])^2]} = \Theta(1) \tag{D.16}$$

for all $m$. Motivated by this observation, we expect (D.2) to hold in practice, though this may be difficult to verify rigorously. Also note that while (D.3) gives an expression for ratio of mean squared errors for the regression estimator in the finite-sample regime, the number of samples needs to be large enough so that we can ignore the correction term.

Now, we specialize Theorem D.1 to Shapley value estimation.

**Corollary D.2.** *Let $\phi^*$ denote the true Shapley value vector and $\alpha$ as in Theorem 2.1. Given $\lambda \in \mathbb{R}$, define $b_\lambda$ as in Theorem 2.2. For $i = 1, 2$, given $m \in \mathbb{N}$ samples from the sampling distribution $\mathcal{P}_i$, denote $\phi_\lambda^M(\mathcal{P}_i)$ and $\phi_\lambda^R(\mathcal{P}_i)$ to be the matrix-vector multiplication estimator and regression estimator, respectively. Then, for all $\lambda \in \mathbb{R}$, we have the following results.*

1. (**Matrix-Vector Multiplication**)

$$\frac{\mathbb{E}[\|\phi_\lambda^M(\mathcal{P}_1) - \phi^*\|^2]}{\mathbb{E}[\|\phi_\lambda^M(\mathcal{P}_2) - \phi^*\|^2]} = \frac{\gamma_1(b_\lambda) - \|\phi^* - \alpha\mathbf{1}\|^2}{\gamma_2(b_\lambda) - \|\phi^* - \alpha\mathbf{1}\|^2}. \tag{D.17}$$

2. (**Regression**) *If for $i = 1, 2$, $\mathbb{E}[\|\phi_\lambda^R(\mathcal{P}_i) - \phi^*\|^2] \neq 0$ and*

$$\frac{\sqrt{\mathbb{E}[\|\phi_\lambda^R(\mathcal{P}_i) - \phi^*\|^4]}}{\mathbb{E}[\|\phi_\lambda^R(\mathcal{P}_i) - \phi^*\|^2]} = O(1), \tag{D.18}$$

*we have*

$$\frac{\mathbb{E}[\|\phi_\lambda^R(\mathcal{P}_1) - \phi^*\|^2]}{\mathbb{E}[\|\phi_\lambda^R(\mathcal{P}_2) - \phi^*\|^2]} = \left(1 \pm O\left(\frac{1}{\sqrt{m}}\right)\right) \frac{\gamma_1(\boldsymbol{P}_U b_\lambda)}{\gamma_2(\boldsymbol{P}_U b_\lambda)}. \tag{D.19}$$

*Proof.* Denote $\boldsymbol{S}$ to be $m \times 2^d - 2$ sketching matrix obtained by sampling with replacement according to appropriate sampling probability. Let $\boldsymbol{U}$ and $\boldsymbol{Q}$ be defined as in Theorem 2.1.

1. From (A.47), we know that $\|\phi_\lambda^M - \phi^*\| = \|\boldsymbol{U}^\mathsf{T}\boldsymbol{S}^\mathsf{T}\boldsymbol{S}b_\lambda - \boldsymbol{U}^\mathsf{T}b_\lambda\|$. Furthermore, from Theorem 2.1, we have $\|\boldsymbol{U}^\mathsf{T}b_\lambda\| = \|\boldsymbol{Q}\boldsymbol{U}^\mathsf{T}b_\lambda\| = \|\phi^* - \alpha\mathbf{1}\|$. Then, the result follows from Theorem D.1.

2. From (A.44), we have $\|\phi_\lambda^R - \phi^*\| = \|(\boldsymbol{S}\boldsymbol{U})^+ \boldsymbol{S}b_\lambda - \boldsymbol{U}^\mathsf{T}b_\lambda\| = \min_{\boldsymbol{y}}\|\boldsymbol{S}\boldsymbol{U}\boldsymbol{y} - \boldsymbol{S}b_\lambda\|$. Then, the result follows from Theorem D.1. $\qquad\square$

The results of this section shows that while the theoretical guarantees derived in Theorem 2.2 only give upper bounds on the sample complexity, the quantity $\gamma$ appearing in this theorem in fact determines the finite-sample/asymptotic behavior of the mean squared errors, as shown in Corollary D.2. Therefore, as long as our metric of performance is the mean squared error, we can directly compare the performance of different sampling schemes by comparing the corresponding values of $\gamma$.

# E  Adversarial example

In this section, we develop an adversarial example that help us separate the performance (in terms of the mean squared error) of $\ell_2$-squared sampling, kernel weight sampling, and modified $\ell_2$ sampling. The main intuition for construction such adversarial examples comes from Corollary A.9 and Corollary D.2, where we compare the value of $\gamma(\boldsymbol{z})$ (see (2.3)) for different sampling strategies. The vector $\boldsymbol{z}$ is either equal to $b_\lambda$ or $(\boldsymbol{I} - \boldsymbol{U}\boldsymbol{U}^T)b_\lambda$ as in Theorem 2.2. For ease of comparison, in our adversarial example, we will construct a model for which $b_\lambda = (\boldsymbol{I} - \boldsymbol{U}\boldsymbol{U}^T)b_\lambda$, and the lower bounds in Corollary A.9 are saturated up to constant factors. For simplicity, we fix $\lambda = (v([d]) - v(\emptyset))/d = \alpha$, as done in previous studies [LL17; MW25].

We now construct an example for which we can provably show better theoretical guarantees for modified $\ell_2$ sampling and kernel weight sampling compared to $\ell_2$-squared sampling. To that end,

define the function $f\colon \mathbb{R}^d \to \mathbb{R}$, which is our model acting on $d$-dimensional input data, as $f(x) = g(\sum_{i=1}^d h(x_i))$, where $g$ and $h$ are real-valued functions to be chosen below. While there is a reasonable freedom in defining the functions $h$ and $g$, we choose these judiciously in order to theoretically compute the Shapley values. Given a parameter $\epsilon_0 \in (0,1)$, we define

$$h(x) = \begin{cases} 1 & \text{if } x > \epsilon_0 \\ 0 & \text{otherwise.} \end{cases} \tag{E.1}$$

Furthermore, given parameters $n \in \mathbb{N}$ (independent of $d$) and $\xi, \chi \in \mathbb{R}$, we define $g\colon \mathbb{R} \to \mathbb{R}$ as

$$g(x) = \begin{cases} \xi\left(\frac{x}{d}\right)^2 + \chi x & \text{if } 1 \le x \le n \text{ or } d - n \le x \le d - 1 \\ \chi x & \text{otherwise.} \end{cases} \tag{E.2}$$

Then, we have the following result.

**Proposition E.1.** *For the model $f(\boldsymbol{x}) = g(\sum_{i=1}^d h(x_i))$, where $h$ is given in (E.1) and $g$ is given in (E.2), baseline $\boldsymbol{y} = \boldsymbol{0}$, and explicand $\boldsymbol{x} = \boldsymbol{1}$, we have $\boldsymbol{\phi}^* = \chi\boldsymbol{1}$ and*

$$\begin{aligned}
\gamma_{\ell_2^2}((\boldsymbol{I} - \boldsymbol{U}\boldsymbol{U}^\mathsf{T})\boldsymbol{b}_\lambda) &= \gamma_{\ell_2^2}(\boldsymbol{b}_\lambda) = \Theta(d) \\
\gamma_{\text{ker}}((\boldsymbol{I} - \boldsymbol{U}\boldsymbol{U}^\mathsf{T})\boldsymbol{b}_\lambda) &= \gamma_{\text{ker}}(\boldsymbol{b}_\lambda) = \Theta(\log(d)) \\
\gamma_{\text{m-}\ell_2}((\boldsymbol{I} - \boldsymbol{U}\boldsymbol{U}^\mathsf{T})\boldsymbol{b}_\lambda) &= \gamma_{\text{m-}\ell_2}(\boldsymbol{b}_\lambda) = \Theta(\sqrt{d}).
\end{aligned} \tag{E.3}$$

*Proof.* First, we compute $\boldsymbol{b}_\lambda$ and show that $(\boldsymbol{I} - \boldsymbol{U}\boldsymbol{U}^\mathsf{T})\boldsymbol{b}_\lambda = \boldsymbol{b}_\lambda$. For a given subset $S$ of $[d]$, define $\boldsymbol{x}^S \in \mathbb{R}^d$ as $x_i^S = x_i$ if $i \in S$ and $x_i^S = y_i$ if $y \notin S$. Then, from the definition of $f$, it follows that for all $S \subseteq [d]$, we have $v(S) = f(\boldsymbol{x}^S) = g(|S|)$. By construction, we have $v([d]) = \chi d$ and $v(\emptyset) = 0$. Since $v(S)$ depends only on the size of the subset $S$, by (1.1), we have that $\boldsymbol{\phi}^* = \phi_0\boldsymbol{1}$ for some constant $\phi_0^* \in \mathbb{R}$. Then, the constraint $\boldsymbol{1}^T\boldsymbol{\phi}^* = v([d]) - v(\emptyset)$ gives $\phi_0^* = (v[d] - v(\emptyset))/d = \chi$. Thus, for this example, we have $\lambda = \alpha = \chi$. Since $\boldsymbol{\phi}^* = \chi\boldsymbol{1} = \alpha\boldsymbol{1}$, from Theorem 2.1, we have $\boldsymbol{Q}\boldsymbol{U}^\mathsf{T}\boldsymbol{b}_\lambda = \boldsymbol{0}$, and therefore, $\boldsymbol{U}\boldsymbol{U}^\mathsf{T}\boldsymbol{b}_\lambda = (\boldsymbol{U}\boldsymbol{Q}^\mathsf{T})(\boldsymbol{Q}\boldsymbol{U}^\mathsf{T})\boldsymbol{b}_\lambda = \boldsymbol{0}$. It follows that $(\boldsymbol{I} - \boldsymbol{U}\boldsymbol{U}^T)\boldsymbol{b}_\lambda = \boldsymbol{b}_\lambda$.

Next, we compute $\|\boldsymbol{b}_\lambda\|^2$, $\|\boldsymbol{H}\boldsymbol{b}_\lambda\|^2$, and $\|\sqrt{\boldsymbol{H}}\boldsymbol{b}_\lambda\|^2$ (see Corollary A.10). Since $v$ depends only on the size of the subset and $\lambda = \chi$, we obtain

$$\|\boldsymbol{b}_\lambda\|^2 = \frac{d}{d-1}\sum_{h=1}^{d-1}\frac{d-1}{h(d-h)}(g(h) - \lambda h)^2 = \frac{\xi^2}{d^3}\left(\sum_{h=1}^n \frac{h^3}{d-h} + \sum_{h=d-n}^{d-1}\frac{h^3}{d-h}\right) = \Theta(1) \quad \text{(E.4)}$$

since $n$ is a constant independent of $d$. Similarly, we have

$$\|\boldsymbol{H}\boldsymbol{b}_\lambda\|^2 = \frac{d}{d-1}\sum_{h=1}^{d-1}\frac{d-1}{h(d-h)}\frac{h(d-h)}{d^2}(g(h) - \lambda h)^2 = \frac{\xi^2}{d^5}\left(\sum_{h=1}^n h^4 + \sum_{h=d-n}^{d-1}h^4\right) = \Theta\left(\frac{1}{d}\right). \tag{E.5}$$

We also have

$$\begin{aligned}
\|\sqrt{\boldsymbol{H}}\boldsymbol{b}_\lambda\|^2 &= \frac{d}{d-1}\sum_{h=1}^{d-1}\frac{d-1}{h(d-h)}\frac{\sqrt{h(d-h)}}{d}(g(h) - \lambda h)^2 \\
&= \frac{\xi^2}{d^4}\left(\sum_{h=1}^n \frac{h^{3.5}}{\sqrt{d-h}} + \sum_{h=d-n}^{d-1}\frac{h^{3.5}}{\sqrt{d-h}}\right) = \Theta\left(\frac{1}{\sqrt{d}}\right).
\end{aligned} \tag{E.6}$$

Therefore, by Corollary A.10, we have

$$\begin{aligned}
\gamma_{\ell_2^2}(\boldsymbol{b}_\lambda) &= \Theta(d\|\boldsymbol{b}_\lambda\|^2) = \Theta(d), \\
\gamma_{\text{ker}}(\boldsymbol{b}_\lambda) &= \Theta(d\log(d)\|\boldsymbol{H}\boldsymbol{b}_\lambda\|^2) = \Theta(\log(d)), \\
\gamma_{\text{m-}\ell_2}(\boldsymbol{b}_\lambda) &= \Theta(d\|\sqrt{\boldsymbol{H}}\boldsymbol{b}_\lambda\|^2) = \Theta(\sqrt{d}).
\end{aligned} \tag{E.7}$$

$\square$

We remark that the adversarial model constructed in this section is a specific toy example meant to illustrate the advantage of modified $\ell_2$ and kernel sampling. One can construct many such adversarial examples for which modified $\ell_2$ and kernel gives better performance than both leverage scores. We can now translate these results into statements concerning the mean squared error for the different sampling schemes.

**Corollary E.2.** *Denote $\mathcal{P}_{\ell_2^2}$, $\mathcal{P}_{\text{ker}}$, and $\mathcal{P}_{\text{m-}\ell_2}$ to be the sampling distributions for $\ell_2^2$-squared, kernel, and modified $\ell_2$ weights, respectively. Then, for the model $f(\boldsymbol{x}) = g(\sum_{i=1}^{d} h(x_i))$, where $h$ is given in (E.1) and $g$ is given in (E.2), baseline $\boldsymbol{y} = \boldsymbol{0}$, and explicand $\boldsymbol{x} = \boldsymbol{1}$, we have (using $m$ samples, drawn with replacement)*

$$
\begin{aligned}
\frac{\mathbb{E}[\|\phi_\lambda^{\text{M}}(\mathcal{P}_{\ell_2^2}) - \phi^*\|^2]}{\mathbb{E}[\|\phi_\lambda^{\text{M}}(\mathcal{P}_{\text{ker}}) - \phi^*\|^2]} &= \frac{\gamma_{\ell_2^2}(\boldsymbol{b}_\lambda)}{\gamma_{\text{ker}}(\boldsymbol{b}_\lambda)} = \Theta\left(\frac{d}{\log(d)}\right), \\
\frac{\mathbb{E}[\|\phi_\lambda^{\text{M}}(\mathcal{P}_{\text{m-}\ell_2}) - \phi^*\|^2]}{\mathbb{E}[\|\phi_\lambda^{\text{M}}(\mathcal{P}_{\text{ker}}) - \phi^*\|^2]} &= \frac{\gamma_{\text{m-}\ell_2}(\boldsymbol{b}_\lambda)}{\gamma_{\text{ker}}(\boldsymbol{b}_\lambda)} = \Theta\left(\frac{\sqrt{d}}{\log(d)}\right), \\
\frac{\mathbb{E}[\|\phi_\lambda^{\text{M}}(\mathcal{P}_{\ell_2^2}) - \phi^*\|^2]}{\mathbb{E}[\|\phi_\lambda^{\text{M}}(\mathcal{P}_{\text{m-}\ell_2}) - \phi^*\|^2]} &= \frac{\gamma_{\text{m-}\ell_2}(\boldsymbol{b}_\lambda)}{\gamma_{\text{ker}}(\boldsymbol{b}_\lambda)} = \Theta\left(\sqrt{d}\right), \\
\frac{\mathbb{E}[\|\phi_\lambda^{\text{R}}(\mathcal{P}_{\ell_2^2}) - \phi^*\|^2]}{\mathbb{E}[\|\phi_\lambda^{\text{R}}(\mathcal{P}_{\text{ker}}) - \phi^*\|^2]} &\approx \frac{\gamma_{\ell_2^2}(\boldsymbol{b}_\lambda)}{\gamma_{\text{ker}}(\boldsymbol{b}_\lambda)} = \Theta\left(\frac{d}{\log(d)}\right) \quad \text{for large enough } m, \\
\frac{\mathbb{E}[\|\phi_\lambda^{\text{R}}(\mathcal{P}_{\text{m-}\ell_2}) - \phi^*\|^2]}{\mathbb{E}[\|\phi_\lambda^{\text{R}}(\mathcal{P}_{\text{ker}}) - \phi^*\|^2]} &\approx \frac{\gamma_{\text{m-}\ell_2}(\boldsymbol{b}_\lambda)}{\gamma_{\text{ker}}(\boldsymbol{b}_\lambda)} = \Theta\left(\frac{\sqrt{d}}{\log(d)}\right) \quad \text{for large enough } m, \\
\frac{\mathbb{E}[\|\phi_\lambda^{\text{R}}(\mathcal{P}_{\ell_2^2}) - \phi^*\|^2]}{\mathbb{E}[\|\phi_\lambda^{\text{R}}(\mathcal{P}_{\text{m-}\ell_2}) - \phi^*\|^2]} &\approx \frac{\gamma_{\ell_2^2}(\boldsymbol{b}_\lambda)}{\gamma_{\text{ker}}(\boldsymbol{b}_\lambda)} = \Theta\left(\sqrt{d}\right) \quad \text{for large enough } m.
\end{aligned}
\tag{E.8}
$$

*The expressions for the ratio of mean squared errors for the regression estimator hold under the technical assumption (D.18) stated in Corollary D.2.*

*Proof.* This follows by directly substituting the results of Proposition E.1 in Corollary D.2. □

This example shows that modified $\ell_2$ gives an advantage over leverage scores by a factor of $\sqrt{d}$. On the other hand, kernel weights give a factor of $d/\log(d)$ advantage over leverage scores, while a factor of $\sqrt{d}/\log(d)$ advantage over modified $\ell_2$. These saturate the lower bounds in Corollary A.9. Since we have the analytical expressions for $\gamma$ for the adversarial example studied in this section, in Fig. 4, we plot the ratio of $\gamma$ for different the sampling distributions using these expressions.

# F   Methodology

In this section, we describe our estimators algorithmically. The unified theoretical framework can directly be implemented into an algorithmic framework, which we depict in Fig. 5. The general procedure to generate the Shapley values in our framework requires three choices: (1) a sampling distributions on the index-sizes, (2) a strategy for sampling (with replacement, without replacement) and (3) an approximation method (least squares or matrix-vector). Least squares and matrix-vector estimation are reported in Section 2. The missing detail is *how the sampling procedure is implemented* (this is the middle column in Fig. 5). We report this in Algorithm 1 for with replacement sampling and in Algorithm 2.

## F.1   With Replacement Estimators

Sampling with replacement to generate the sketch is a computationally efficient procedure that performs well in practice. However, if the number of samples $m > 2^d$, the estimator will fail to compute exact Shapley values in general. We report the sampling procedure as implemented in our experimental evaluations in Algorithm 1.

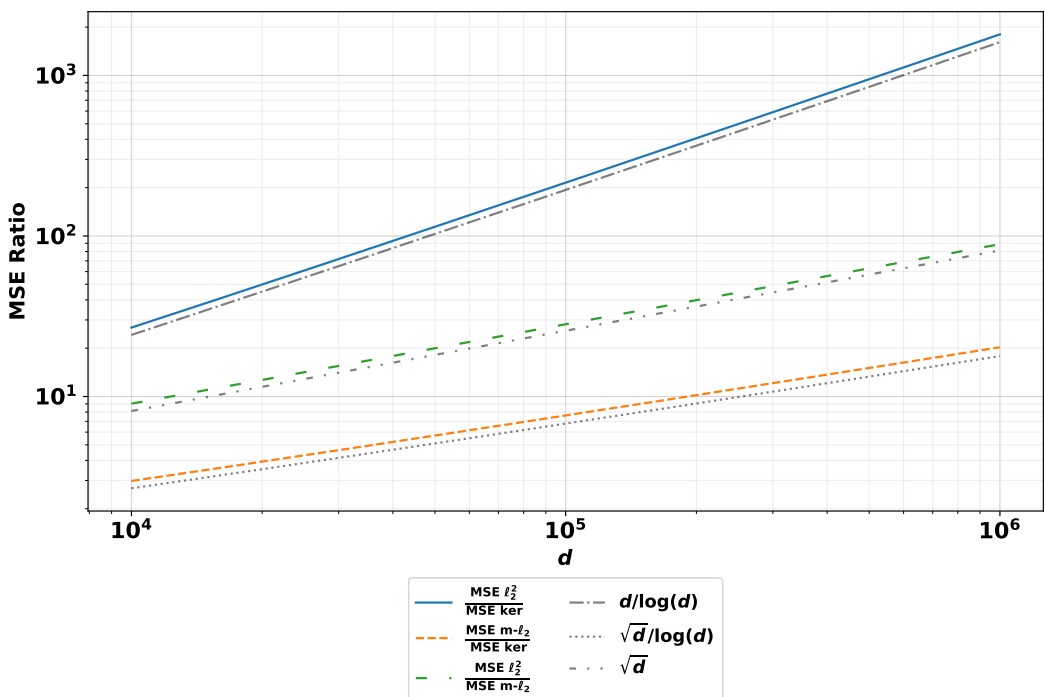

**Figure 4:** Ratio of mean-squared errors (MSE) as a function of the dimension for different sampling strategies for the adversarial model in Appendix E (computed analytically from expressions for $\gamma$). The matrix-vector multiplication estimator and regression estimator have (almost) the same MSE ratio for this model (see Corollary E.2). For $\ell_2$-squared v/s kernel (solid) and modified $\ell_2$ vs kernel (dashed), kernel weights give an advantage by a factor of $\tilde{O}(d)$ and $\tilde{O}(\sqrt{d})$ respectively. On the other hand, for modified $\ell_2$ v/s $\ell_2$-squared (long dashed), modified $\ell_2$ outperforms $\ell_2$-squared by a factor of $O(\sqrt{d})$.

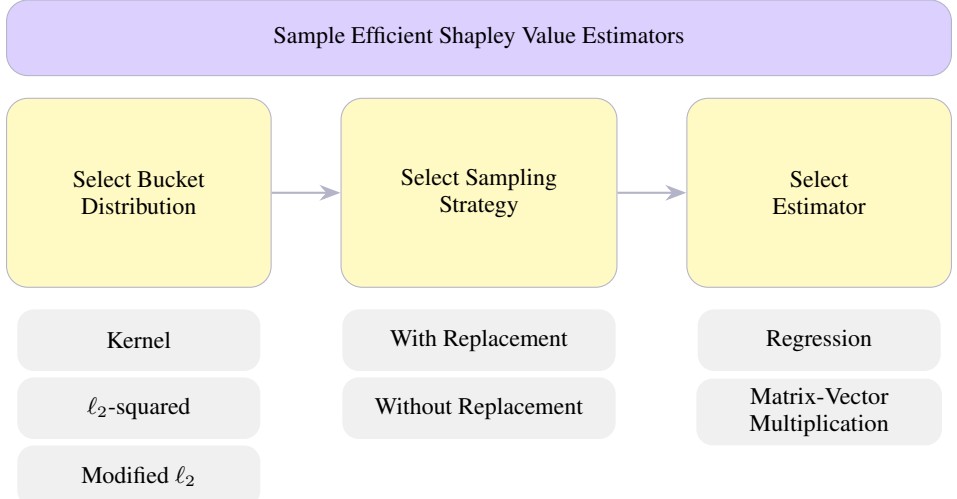

**Figure 5:** The unified framework for estimating Shapley values with the proposed class of estimators. First, we define a distribution to apply to each *bucket* (i.e., to the selection of the bit vector to select - $p_i$ is the probability of sampling an item from bucket/coalition of size (or bit vector with Hamming weight) $i \in [d]$. Then we select a sampling strategy (with or without replacement). Finally, we select the estimation strategy. If we limit ourselves to $\ell_2$-squared and modified, and kernel distribution, this provides a total of $3 \times 2 \times 2 = 12$ estimators.

**Algorithm 1** Sampling with Replacement (paired sampling)

---

**Require:** $d$: number of features $m$: number of samples expected by user, $p$ choice of distribution on $d-1$ buckets, maxval maximum value before Poisson approximation is applied (our algorithm uses $10^{10}$) .

**Ensure:** $SZ \in \mathbb{R}^{m \times d}, \tilde{W} \in \mathbb{R}^{m \times m}$: sub-sampled $Z$ matrix and weights

1: Redefine $p_i \leftarrow 2p_i$ for $i = 1, \ldots, \lfloor \frac{d-1}{2} \rfloor$.
2: First sample with replacement $\lfloor \frac{m}{2} \rfloor$ from indices $i = 1, ..., \lceil \frac{d-1}{2} \rceil$, $j \in [m]$ each with probability $p_1, ..., p_{\lceil \frac{d-1}{2} \rceil}$. Denote $m_i$ to be the number of times we observe the index (bucket) $i$ during sampling.
3: For each $i = 1, \ldots, \lceil \frac{d-1}{2} \rceil$, construct $\boldsymbol{b}_j \in \{0,1\}^d$ by sampling with replacement $m_i$ bitstrings of size $i$, which is equivalent to sampling without replacement $i$ indices from $[d]$, uniformly; and generate, the complement $\bar{\boldsymbol{b}}_{i,j} = 1 - \boldsymbol{b}_{i,j}$.
4: Construct $\boldsymbol{w}_j = ((d-1)p_j)^{-1}$ and return:

$$SZ = \texttt{stack}([\boldsymbol{b}_j, \bar{\boldsymbol{b}}_j]_{j \in \tilde{m}}), \tilde{W} = \frac{1}{m}\text{diag}([\boldsymbol{w}_j, \boldsymbol{w}_j]_{j \in \tilde{m}}).$$

---

### F.2 Without Replacement Estimators

At first glance, sampling based on the without replacement sampling scheme described in Section 2.3 (see also Appendix A.5) requires flipping $2^{[d]} - 2$ coins. However, for the sampling distributions described in Section 2.2, given the size of a subset, the probability of picking any given subset of that size is constant. This observation is used in [MW25] to avoid flipping exponentially many coins. In particular, one can determine which coins are heads by first determining how many heads there will be of a given subset size, and then picking the resulting subsets of this size uniformly at random. We describe a variant of [MW25, Algorithm 2] in Algorithm 2.

## G  Experimental Details

We use publicly available datasets for reproducibility; choosing particularly those available through the shap for their popularity, ease of use and for a direct comparison with [MW25].

### G.1  Training Details

In this subsection, we detail the experimental design choices and hyperparameter for low Section 3.1 and Section 3.2, including implementation details, to promote reprodicibility.

#### G.1.1  Low Dimensional Experiments

We refer to low dimensional experiments to the content of Section 3.1. For each dataset we train a decision tree from the xgboost. Specifically, we use the XGBRegressor class with 100 estimators and maximum depth of 10 for each task. We replacing missing values with the mean for that feature in the dataset. Note that the goal is not to achieve competitive performance but to rapidly train a model where the Shapley values can be computed exactly and efficiently. The train test splits are ordered 80/20 splits for all datasets; we pick as query and baseline points the first data points of the test and train datasets respectively.

#### G.1.2  High Dimensional Experiments

We refer to high dimensional experiments to the content of Section 3.2. For the two classification tasks, we train a RandomForestClassifier from the sci-kit learn library. The random forest has maximum depth 15 and random state 42 for both tasks (MNIST and CIFAR-10). For both datasets, we pick as query and baseline points the first data points of the test and train datasets respectively.

For MNIST we use train test splits (80/20) with random state 42 using the train_test_split method on sci-kit learn. We achieve a test accuracy of 96.3%.

**Algorithm 2** Sampling without Replacement (paired sampling, modified from [MW25])

---

**Require:** $d$: number of features $m$: number of samples expected by user, $p$ choice of distribution on $d$ buckets, maxval maximum value before Poisson approximation is applied (our algorithm uses $10^{10}$) .

**Ensure:** $SZ \in \mathbb{R}^{m \times d}, \tilde{W} \in \mathbb{R}^{m \times m}$: sub-sampled $Z$ matrix and weights

1: Redefine $p_i \leftarrow 2p_i$ for $i = 1, \ldots, \lfloor \frac{d-1}{2} \rfloor$.
2: Choose $\alpha$ such that

$$\lfloor \frac{m}{2} \rfloor = \sum_{i=1}^{\lceil \frac{d-1}{2} \rceil} \min\left( \binom{d}{i}, \alpha p_i \right)$$

using binary search algorithm.
3: **if** $\binom{d}{i} < $ maxval **then** let

$$m_i \leftarrow \text{Binomial}\left( \binom{d}{i}, \min(1, \alpha p_i) \right)$$

4: **else** let

$$m_i \leftarrow \text{Poisson}(\alpha p_i),$$

and let $\tilde{m} = \sum_{i=1}^{\lceil \frac{d-1}{2} \rceil} m_i$.
5: **end if**
6: Construct $j \in [m_i]$ bitstring arrays of $\boldsymbol{b}_j \in \{0,1\}^d$ of size $i \in [\lceil \frac{d-1}{2} \rceil]$ without replacement (e.g. using Fisher Yates shuffling or Algorithms 2,3 in [MW25]). If there is a middle bucket (i.e., $d$ is odd), fix $\boldsymbol{b}_{\lceil \frac{d-1}{2} \rceil, j} = 1$ and sample without replacement from the remaining bitstrings. Then, generate the complement $\bar{\boldsymbol{b}}_{i,j} = 1 - \boldsymbol{b}_{i,j}$.
7: Construct $\boldsymbol{w}_j = \left( (d-1) \min(\binom{d}{j}, \alpha p_j) \right)^{-1}$ and return:

$$SZ = \text{stack}([\boldsymbol{b}_j, \bar{\boldsymbol{b}}_j]_{j \in \tilde{m}}), \tilde{W} = \text{diag}([\boldsymbol{w}_j, \boldsymbol{w}_j]_{j \in \tilde{m}}).$$

---

For the CIFAR-10 dataset, we use standard train test split from the `torchvision` library. We preprocess the input data by normalising. We achieve low accuracy $44.98\%$, as should be expected using basic tree-based models for CIFAR-10; however, using said models enables the exact computation of Shapley values, which we consider a more important aspect for the paper. We used paired, with replacement estimators with modified $\ell_2$ and $\boldsymbol{b}_\alpha$

## G.2 Datasets

In this subsection, we briefly describe the datasets used for the experiments, for completeness.

### G.2.1 `shap` Datasets

**Adult.** Demographic information about individuals collected from the 1994 U.S. Census database. It is used to predict whether a person earns more than $\$50,000$ dollars per year based on individual attributes: age, work, class, education, etc.,.

**California.** The California Housing dataset is a linear regression tasks containing information collected from the 1990 U.S. Census. This includes data on housing prices as targets, and median income, housing age, and average number of rooms as the input features.

**Communities.** Communities and crime dataset studies the relationship between community characteristics and crime rates, including socio-economic, law enforcement and demographic factors - in the United States. This is a regression task.

**Diabetes.** The Diabetes dataset is used to predict onset of diabetes as diagnostic measurements. It includes factors like age, blood pressure, and body mass index.

**Independent and Correlated** Datasets that are used to study the behavior of the algorithm under the assumption of feature independence and correlation respectively. The target is a linear regressor of the features.

**IRIS.** This classic dataset in the field consists of 150 samples of iris flowers, with three different species: Iris setosa, iris versicolor and Iris virginica. Each has four features, describing anatomical sizes of the plant. This is a classification task.

**NHANES** The National Health and Nutrition Examination Survey (NHANES) is a program designed to assess health and nutritional status of citizens of the United States. Based on interview and physical examination data, it predicts survival times based on medical features (regression).

### G.2.2 Image Datasets

**MNIST** The MNIST dataset is a collection of handwritten digets (0-9); the classic task is to classify into their respective value. The dataset has $28 \times 28 (= 748)$ dimensions. This is an incredibly popular dataset used for training and testing image classification algorithms.

**CIFAR-10** CIFAR-10 dataset is a classification task dataset where a collection of 60,000 images, with $32 \times 32 \times 3 (= 3072)$ dimensions, are mapped to target clas: cars, airplanes, birds, cats, deer, dogs, frogs, horses, ships and trucks. The standard split for this dataset is 10,000 test images and 50,000 training images, which we use in our experiments. This is considered a relatively challenging dataset for boosted trees and feedforward neural networks; good performance is achieved, however, for convolutional neural networks.

### G.3 Adjustments for Classification Tasks

In the computation of Shapley values for classification tasks, a slight adjustment is needed. While the output of the classifier is ultimately a single value $f(x) \in [c]$, for $c \in \mathbb{N}$ classes, computing Shapley on a value function that predicts classes would be incorrect: in genreal, the classes should not be considered an ordered set. Therefore, we compute Shapley values on the probabilities for each class.

Successively, the mean squared error (normalized) is computed on the vectorized output of the Shapley computation. For example, for a classification task with $c$ classes and a $d$-dimensional input space, the Shapley values will be in $\hat{\phi} \in \mathbb{R}^{c \times d}$. Therefore, to compute the mean squared error (normalized), we vectorize the matrices of Shapley values and compute as usual.

In high dimensional experiments, the average is taken across test points. We also provide evaluation details with the purpose of increasing transparency and promoting reproducibility of experiments.

## H  Extended Experimental Results

The goal of this section is to report the numerical results from our experiments. We first report the extended experiments from Section 3.1, followed by experiments in Section 3.2. Importantly, we share tables and plots containing our results.

### H.1  Low Dimensional Experiments

In low dimensional experiments, as described in Appendix G and Section 3.1, we compute the mean squared error (normalized by norm of exact Shapley values) as the median of 100 random seeds (0-99). We also report the average and interquartile ranges for each of the experiments. These results are summarized in Table 2 (median), Table 3 (lower quantile), Table 4 (upper quantile), and Table 5 ($b_0$ values). In each of those tables we report the values for selected number of samples: for IRIS we report $m = 10$; Adult, California and Diabetes we show results for $m = 64$; for Communities, Correlated, Independent, and NHANES we report $m = 50000$.

### H.2  High Dimensional Experiments

For high dimensional experiments, as described in Appendix G and Section 3.2, we compute the the mean squared error (normalized by norm of exact Shapley values) as the average of 10 test points from the respective datasets. We also report quantiles and median in Table 6.

## H.3 Faithfulness Experiments

For each experiment in the high dimensional setting, we compute the insertion and deletion curves as reported in Table 7 (insertion and deletion AUC), Fig. 6 (insertion and deletion curves for MNIST), and Fig. 7 (insertion and deletion curves for CIFAR-10).

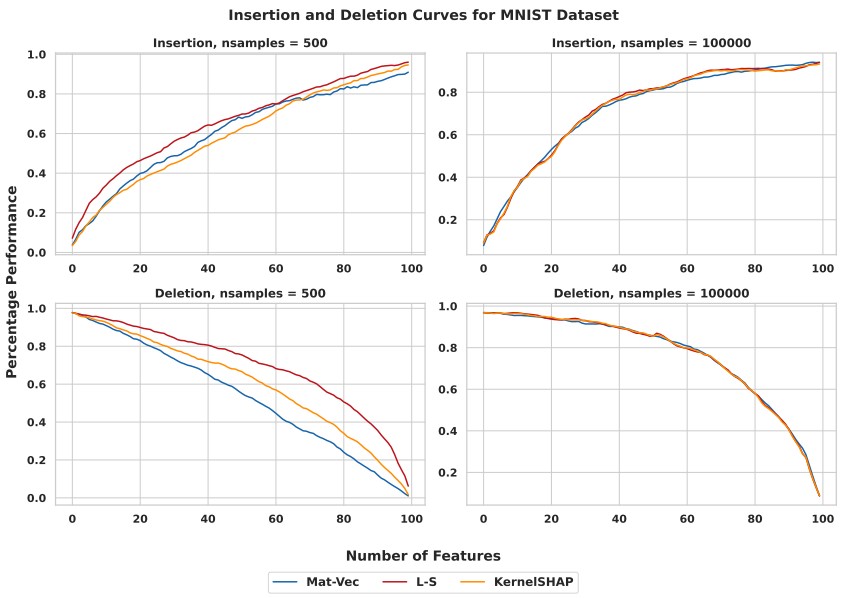

**Figure 6:** Insertion and Deletion Plots for MNIST Dataset, for varying number of Samples. As expected, the three methods converge to towards the same curve as the plots increase.

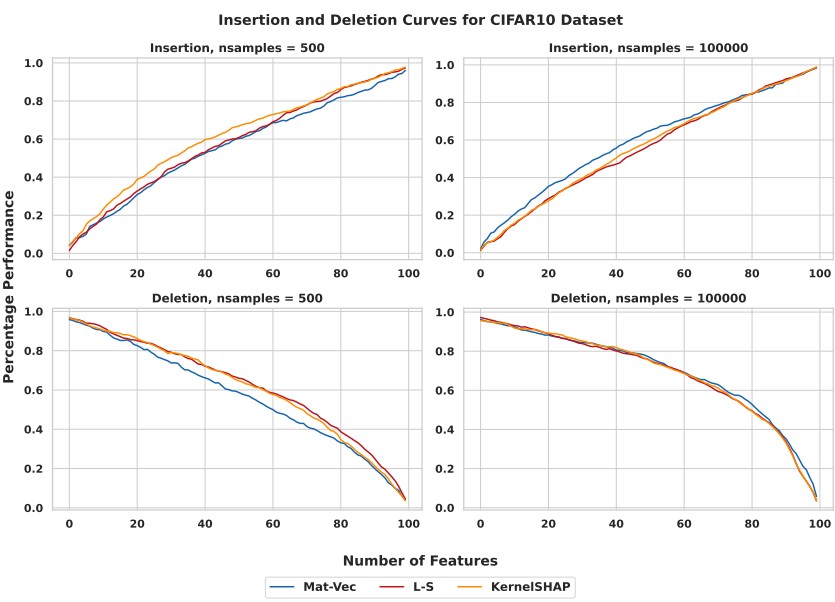

**Figure 7:** Insertion and Deletion Plots for MNIST Dataset, for varying number of Samples. As expected, the three methods converge to towards the same curve as the plots increase.

For any Shapley value, the insertion curve is computed by adding features, in order of importance, to an empty vector and computing the prediction. The expectation is that the most important features (the first features to be added) contribute most to reconstructing the original prediction. Hence, a good feature attribution method will maximise the area under this curve.

Conversely, the deletion curve is computed by removing features (replacing them by 0), in order of importance, to the original test point, and computing the prediction. The expectation is that the most important features (the first features to be removed) will deteriorate the performance rapidly. Hence, a good feature attribution method will minimize the area under the curve.

We compute this curve for each test point and average across the curves (as reported in Fig. 6 and Fig. 7). We limit the computation to the top 100 features for both datasets and report the values in percentages. Note that for MNIST, this is a complete ordering of all features (64), whereas for CIFAR-10, this is only a fraction of the 3072 dimensions. These are commonly used faithfulness measures from the literature: the higher the insertion AUC / the lower the deletion AUC, the higher the faithfulness of the model.

Moreover, for each test point, we compute the Spearman correlation rank from the scipy stats library, on the Shapley values, summed (in absolute value) across classes. We report the results in Table 7. This is a measure of agreement between the true Shapley values and the estimated Shapley values. The higher the Spearman rank correlation, the better the faithfulness of the approximation. Note, however, that due to the presence of many small values (near zero Shapley values) in image classification tasks, this measure may overemphasize incongruence between features.

| Dataset | Approximation | Sampling | With Replacement | Without Replacement |
|---|---|---|---|---|
| Adult | Kernel | kernel | | 0.0221 |
| | Matrix-Vec | kernel | 0.577 | 0.0904 |
| | | $\ell_2$-squared | 0.208 | 0.0793 |
| | | modified $\ell_2$ | 0.2 | 0.0888 |
| | Least Squares | kernel | 0.00652 | 0.0016 |
| | | $\ell_2$-squared | 0.00509 | 0.00146 |
| | | modified $\ell_2$ | 0.00572 | **0.00136** |
| California | Kernel | kernel | | 0.0165 |
| | Matrix-Vec | kernel | 1.03 | 0.151 |
| | | $\ell_2$-squared | 0.218 | 0.148 |
| | | modified $\ell_2$ | 0.254 | 0.136 |
| | Least Squares | kernel | 0.00419 | 0.00234 |
| | | $\ell_2$-squared | 0.0039 | 0.00222 |
| | | modified $\ell_2$ | 0.00427 | **0.00193** |
| Communities | Kernel | kernel | | 0.000810 |
| | Matrix-Vec | kernel | 0.009871 | 0.007950 |
| | | $\ell_2$-squared | 0.009174 | 0.026858 |
| | | modified $\ell_2$ | 0.007645 | 0.018672 |
| | Least Squares | kernel | 0.000887 | 0.000719 |
| | | $\ell_2$-squared | 0.000479 | **0.000301** |
| | | modified $\ell_2$ | 0.000554 | 0.000315 |
| Correlated | Kernel | kernel | | 0.000620 |
| | Matrix-Vec | kernel | 0.007933 | 0.005208 |
| | | $\ell_2$-squared | 0.009155 | 0.002495 |
| | | modified $\ell_2$ | 0.006804 | 0.002772 |
| | Least Squares | kernel | 0.000252 | 0.000169 |
| | | $\ell_2$-squared | 0.000146 | **0.000064** |
| | | modified $\ell_2$ | 0.000172 | 0.000072 |
| Diabetes | Kernel | kernel | | 0.0163 |
| | Matrix-Vec | kernel | 1.47 | 0.218 |
| | | $\ell_2$-squared | 0.235 | 0.179 |
| | | modified $\ell_2$ | 0.274 | 0.2 |
| | Least Squares | kernel | 0.00183 | 0.0106 |
| | | $\ell_2$-squared | **0.00155** | 0.00889 |
| | | modified $\ell_2$ | 0.0016 | 0.00983 |
| Independent | Kernel | kernel | | 0.000480 |
| | Matrix-Vec | kernel | 0.006640 | 0.004847 |
| | | $\ell_2$-squared | 0.005631 | 0.004184 |
| | | modified $\ell_2$ | 0.005024 | 0.003773 |
| | Least Squares | kernel | 0.000673 | 0.000502 |
| | | $\ell_2$-squared | 0.000401 | **0.000370** |
| | | modified $\ell_2$ | 0.000487 | 0.000377 |
| IRIS | Kernel | kernel | | 0.0222 |
| | Matrix-Vec | kernel | 0.471 | 0.218 |
| | | $\ell_2$-squared | 0.359 | 0.229 |
| | | modified $\ell_2$ | 0.366 | 0.254 |
| | Least Squares | kernel | 3.28e-05 | 1.64e-05 |
| | | $\ell_2$-squared | 3.28e-05 | **1.45e-05** |
| | | modified $\ell_2$ | 3.28e-05 | 1.54e-05 |
| NHANES | Kernel | kernel | | 0.000597 |
| | Matrix-Vec | kernel | 0.011273 | 0.008429 |
| | | $\ell_2$-squared | 0.013369 | 0.009053 |
| | | $\ell_2$-modified | 0.009980 | 0.007377 |
| | Least Squares | kernel | 0.002637 | 0.002153 |
| | | $\ell_2$-squared | 0.001505 | 0.00221 |
| | | modified $\ell_2$ | 0.001674 | **0.000428** |

**Table 2:** Mean Squared Error for Different Sampling and Approximation Methods Across Various Datasets (best relative MSE performance marked in bold).

| Dataset | Approximation | Sampling | With Replacement | Without Replacement |
|---|---|---|---|---|
| Adult | Kernel | kernel | | 0.0128 |
| | Matrix-Vec | kernel | 0.322 | 0.0635 |
| | | $\ell_2$-squared | 0.124 | 0.0526 |
| | | modified $\ell_2$ | 0.101 | 0.0584 |
| | Least Squares | kernel | 0.00414 | 0.000931 |
| | | $\ell_2$-squared | 0.00316 | 0.000895 |
| | | modified $\ell_2$ | 0.00305 | 0.000871 |
| California | Kernel | kernel | | 0.0118 |
| | Matrix-Vec | kernel | 0.595 | 0.0904 |
| | | $\ell_2$-squared | 0.15 | 0.0939 |
| | | modified $\ell_2$ | 0.156 | 0.0905 |
| | Least Squares | kernel | 0.00271 | 0.0017 |
| | | $\ell_2$-squared | 0.00281 | 0.0015 |
| | | modified $\ell_2$ | 0.00245 | 0.00136 |
| Communities | Kernel | kernel | | 0.000735 |
| | Matrix-Vec | kernel | 0.009144 | 0.007282 |
| | | $\ell_2$-squared | 0.008249 | 0.025044 |
| | | modified $\ell_2$ | 0.006825 | 0.016340 |
| | Least Squares | kernel | 0.000785 | 0.000659 |
| | | $\ell_2$-squared | 0.000436 | 0.000276 |
| | | modified $\ell_2$ | 0.000504 | 0.000284 |
| Correlated | Kernel | kernel | | 0.000547 |
| | Matrix-Vec | kernel | 0.006926 | 0.004395 |
| | | $\ell_2$-squared | 0.008373 | 0.002188 |
| | | modified $\ell_2$ | 0.006078 | 0.002340 |
| | Least Squares | kernel | 0.000219 | 0.000149 |
| | | $\ell_2$-squared | 0.000129 | 0.000058 |
| | | modified $\ell_2$ | 0.000149 | 0.000062 |
| Diabetes | Kernel | kernel | | 0.012 |
| | Matrix-Vec | kernel | 0.983 | 0.153 |
| | | $\ell_2$-squared | 0.148 | 0.124 |
| | | modified $\ell_2$ | 0.169 | 0.134 |
| | Least Squares | kernel | 0.00127 | 0.00742 |
| | | $\ell_2$-squared | 0.000874 | 0.00667 |
| | | modified $\ell_2$ | 0.00126 | 0.00799 |
| Independent | Kernel | kernel | | 0.000435 |
| | Matrix-Vec | kernel | 0.005925 | 0.004309 |
| | | $\ell_2$-squared | 0.004942 | 0.003723 |
| | | modified $\ell_2$ | 0.004517 | 0.003284 |
| | Least Squares | kernel | 0.000581 | 0.000431 |
| | | $\ell_2$-squared | 0.000353 | 0.000322 |
| | | modified $\ell_2$ | 0.000423 | 0.000333 |
| IRIS | Kernel | kernel | | 0.0108 |
| | Matrix-Vec | kernel | 0.241 | 0.123 |
| | | $\ell_2$-squared | 0.187 | 0.127 |
| | | modified $\ell_2$ | 0.192 | 0.101 |
| | Least Squares | kernel | 8.21e-06 | 1.64e-05 |
| | | $\ell_2$-squared | 1.15e-05 | 1.25e-05 |
| | | modified $\ell_2$ | 8.11e-06 | 1.44e-05 |
| NHANES | Kernel | kernel | | 0.000505 |
| | Matrix-Vec | kernel | 0.010318 | 0.007504 |
| | | $\ell_2$-squared | 0.011346 | 0.008037 |
| | | modified $\ell_2$ | 0.008947 | 0.006688 |
| | Least Squares | kernel | 0.002385 | 0.001926 |
| | | $\ell_2$-squared | 0.001368 | 0.00221 |
| | | modified $\ell_2$ | 0.001540 | 0.000375 |

**Table 3:** (Lower Quantile) Mean Squared Error for Different Sampling and Approximation Methods Across Various Datasets

| Dataset | Approximation | Sampling | With Replacement | Without Replacement |
|---|---|---|---|---|
| Adult | Kernel | kernel | | 0.0291 |
| | Matrix-Vec | kernel | 0.968 | 0.144 |
| | | $\ell_2$-squared | 0.276 | 0.115 |
| | | modified $\ell_2$ | 0.298 | 0.128 |
| | Least Squares | kernel | 0.0111 | 0.00225 |
| | | $\ell_2$-squared | 0.00936 | 0.00238 |
| | | modified $\ell_2$ | 0.0091 | 0.00211 |
| California | Kernel | kernel | | 0.0269 |
| | Matrix-Vec | kernel | 1.64 | 0.274 |
| | | $\ell_2$-squared | 0.293 | 0.242 |
| | | modified $\ell_2$ | 0.376 | 0.236 |
| | Least Squares | kernel | 0.00597 | 0.00313 |
| | | $\ell_2$-squared | 0.00542 | 0.00317 |
| | | modified $\ell_2$ | 0.00586 | 0.00331 |
| Communities | Kernel | kernel | | 0.000877 |
| | Matrix-Vec | kernel | 0.010869 | 0.008674 |
| | | $\ell_2$-squared | 0.009949 | 0.029046 |
| | | modified $\ell_2$ | 0.008383 | 0.019998 |
| | Least Squares | kernel | 0.000968 | 0.000787 |
| | | $\ell_2$-squared | 0.000522 | 0.000335 |
| | | modified $\ell_2$ | 0.000597 | 0.000346 |
| Correlated | Kernel | kernel | | 0.000721 |
| | Matrix-Vec | kernel | 0.009207 | 0.005980 |
| | | $\ell_2$-squared | 0.010407 | 0.002896 |
| | | modified $\ell_2$ | 0.007889 | 0.003279 |
| | Least Squares | kernel | 0.000294 | 0.000195 |
| | | $\ell_2$-squared | 0.000159 | 0.000073 |
| | | modified $\ell_2$ | 0.000184 | 0.000081 |
| Diabetes | Kernel | kernel | | 0.023 |
| | Matrix-Vec | kernel | 2.08 | 0.358 |
| | | $\ell_2$-squared | 0.318 | 0.296 |
| | | modified $\ell_2$ | 0.382 | 0.315 |
| | Least Squares | kernel | 0.00249 | 0.0146 |
| | | $\ell_2$-squared | 0.00237 | 0.0144 |
| | | modified $\ell_2$ | 0.0024 | 0.0141 |
| Independent | Kernel | kernel | | 0.000551 |
| | Matrix-Vec | kernel | 0.007710 | 0.005508 |
| | | $\ell_2$-squared | 0.006441 | 0.004724 |
| | | modified $\ell_2$ | 0.005656 | 0.004196 |
| | Least Squares | kernel | 0.000745 | 0.000558 |
| | | $\ell_2$-squared | 0.000463 | 0.000413 |
| | | modified $\ell_2$ | 0.000529 | 0.000444 |
| IRIS | Kernel | kernel | | 0.0453 |
| | Matrix-Vec | kernel | 0.851 | 0.593 |
| | | $\ell_2$-squared | 0.531 | 0.411 |
| | | modified $\ell_2$ | 0.538 | 0.504 |
| | Least Squares | kernel | 3.28e-05 | 2.76e-05 |
| | | $\ell_2$-squared | 3.28e-05 | 0.000131 |
| | | modified $\ell_2$ | 3.28e-05 | 3.3e-05 |
| NHANES | Kernel | kernel | | 0.000704 |
| | Matrix-Vec | kernel | 0.012395 | 0.009490 |
| | | $\ell_2$-squared | 0.014504 | 0.009863 |
| | | modified $\ell_2$ | 0.010979 | 0.008374 |
| | Least Squares | kernel | 0.002917 | 0.002366 |
| | | $\ell_2$-squared | 0.001668 | 0.00221 |
| | | modified $\ell_2$ | 0.001888 | 0.000471 |

**Table 4:** (Upper Quantile) Mean Squared Error for Different Sampling and Approximation Methods Across Various Datasets

| Dataset | Samples | Distribution | $Q_1$ | Median MSE | $Q_3$ |
|---|---|---|---|---|---|
| Adult | 64 | kernel | 0.0574 | 0.0839 | 0.144 |
| | | $\ell_2$-squared | 0.0633 | 0.0977 | 0.143 |
| | | modified $\ell_2$ | 0.0568 | 0.0907 | 0.127 |
| California | 64 | kernel | 0.102 | 0.165 | 0.288 |
| | | $\ell_2$-squared | 0.095 | 0.129 | 0.241 |
| | | modified $\ell_2$ | 0.098 | 0.15 | 0.233 |
| Communities | 50000 | kernel | 0.000765 | 0.000851 | 0.000920 |
| | | $\ell_2$-squared | 0.000555 | 0.000600 | 0.000672 |
| | | modified $\ell_2$ | 0.000595 | 0.000637 | 0.000695 |
| Correlated | 50000 | kernel | 0.000495 | 0.000566 | 0.000644 |
| | | $\ell_2$-squared | 0.000402 | 0.000448 | 0.000528 |
| | | modified $\ell_2$ | 0.000431 | 0.000493 | 0.000562 |
| Diabetes | 64 | kernel | 0.216 | 0.36 | 0.509 |
| | | $\ell_2$-squared | 0.328 | 0.432 | 0.562 |
| | | modified $\ell_2$ | 0.29 | 0.393 | 0.577 |
| Independent | 50000 | kernel | 0.000435 | 0.000510 | 0.000541 |
| | | $\ell_2$-squared | 0.000368 | 0.000426 | 0.000487 |
| | | modified $\ell_2$ | 0.000407 | 0.000446 | 0.000494 |
| IRIS | 10 | kernel | 0.122 | 0.443 | 0.593 |
| | | $\ell_2$-squared | 0.0883 | 0.361 | 0.473 |
| | | modified $\ell_2$ | 0.101 | 0.414 | 0.54 |
| NHANES | 50000 | kernel | 0.002350 | 0.002696 | 0.002915 |
| | | $\ell_2$-squared | 0.001704 | 0.001868 | 0.002084 |
| | | modified $\ell_2$ | 0.001845 | 0.002055 | 0.002203 |

**Table 5:** Values for $b_0$ in Fig. 2 (3) Comparison: Quantile Values for Different Datasets and Sampling Methods (least squares estimator without replacement, paired).

| Dataset | Samples | Approximation | Time | $Q_1$ | Median MSE | $Q_3$ |
|---|---|---|---|---|---|---|
| MNIST | 500 | Matrix-Vector | $7.140 \times 10^{-1}$ | $2.625 \times 10^4$ | $2.668 \times 10^4$ | $2.681 \times 10^4$ |
| | | Least Squares | $1.850 \times 10^0$ | $6.087 \times 10^{-2}$ | $6.144 \times 10^{-2}$ | $6.188 \times 10^{-2}$ |
| | | KernelSHAP | $7.390 \times 10^{-1}$ | $9.235 \times 10^1$ | $1.309 \times 10^2$ | $2.147 \times 10^2$ |
| | 1000 | Matrix-Vector | $9.270 \times 10^{-1}$ | $8.394 \times 10^3$ | $8.491 \times 10^3$ | $8.566 \times 10^3$ |
| | | Least Squares | $1.960 \times 10^0$ | $5.879 \times 10^{-2}$ | $5.914 \times 10^{-2}$ | $5.949 \times 10^{-2}$ |
| | | KernelSHAP | $8.920 \times 10^{-1}$ | $1.745 \times 10^0$ | $1.784 \times 10^0$ | $1.816 \times 10^0$ |
| | 10000 | Matrix-Vector | $5.540 \times 10^0$ | $2.385 \times 10^2$ | $2.393 \times 10^2$ | $2.414 \times 10^2$ |
| | | Least Squares | $7.710 \times 10^0$ | $4.889 \times 10^{-2}$ | $4.939 \times 10^{-2}$ | $4.964 \times 10^{-2}$ |
| | | KernelSHAP | $2.940 \times 10^0$ | $7.866 \times 10^{-2}$ | $7.979 \times 10^{-2}$ | $8.058 \times 10^{-2}$ |
| | 100000 | Matrix-Vector | $5.240 \times 10^1$ | $1.048 \times 10^1$ | $1.058 \times 10^1$ | $1.063 \times 10^1$ |
| | | Least Squares | $6.790 \times 10^1$ | $3.071 \times 10^{-2}$ | $3.117 \times 10^{-2}$ | $3.150 \times 10^{-2}$ |
| | | KernelSHAP | $2.840 \times 10^1$ | $7.374 \times 10^{-3}$ | $7.472 \times 10^{-3}$ | $7.593 \times 10^{-3}$ |
| CIFAR10 | 500 | Matrix-Vector | $1.990 \times 10^1$ | $1.411 \times 10^3$ | $1.476 \times 10^3$ | $3.191 \times 10^3$ |
| | | Least Squares | $9.810 \times 10^2$ | $1.440 \times 10^0$ | $1.478 \times 10^0$ | $1.494 \times 10^0$ |
| | | KernelSHAP | $1.390 \times 10^2$ | $3.112 \times 10^4$ | $1.114 \times 10^5$ | $6.247 \times 10^5$ |
| | 1000 | Matrix-Vector | $2.180 \times 10^1$ | $5.755 \times 10^2$ | $9.362 \times 10^2$ | $1.314 \times 10^3$ |
| | | Least Squares | $1.270 \times 10^3$ | $2.025 \times 10^0$ | $2.047 \times 10^0$ | $2.094 \times 10^0$ |
| | | KernelSHAP | $1.350 \times 10^2$ | $2.411 \times 10^4$ | $1.153 \times 10^5$ | $1.033 \times 10^6$ |
| | 10000 | Matrix-Vector | $5.690 \times 10^1$ | $9.168 \times 10^1$ | $1.031 \times 10^2$ | $1.172 \times 10^2$ |
| | | Least Squares | $1.210 \times 10^3$ | $1.000 \times 10^1$ | $1.035 \times 10^1$ | $1.059 \times 10^1$ |
| | | KernelSHAP | $2.820 \times 10^2$ | $2.846 \times 10^1$ | $2.870 \times 10^1$ | $2.897 \times 10^1$ |
| | 100000 | Matrix-Vector | $3.190 \times 10^2$ | $1.014 \times 10^1$ | $1.068 \times 10^1$ | $1.104 \times 10^1$ |
| | | Least Squares | $3.030 \times 10^3$ | $4.168 \times 10^{-1}$ | $4.252 \times 10^{-1}$ | $4.295 \times 10^{-1}$ |
| | | KernelSHAP | $1.860 \times 10^3$ | $1.050 \times 10^0$ | $1.053 \times 10^0$ | $1.053 \times 10^0$ |

**Table 6:** Performance Metrics for MNIST and CIFAR-10 Datasets Using Different Methods ($\ell_2$-squared estimator, without replacement, paired sampling with $b_0$).

| Dataset | Samples | Approximation | Deletion AUC | Insertion AUC | Rank. Corr. |
|---|---|---|---|---|---|
| MNIST | 500 | Matrix-Vector | 0.758 | 0.721 | 0.632 |
| | | Least Squares | 0.758 | 0.718 | 0.737 |
| | | KernelSHAP | 0.601 | 0.589 | 0.959 |
| | 1000 | Matrix-Vector | 0.758 | 0.719 | 0.639 |
| | | Least Squares | 0.761 | 0.718 | 0.742 |
| | | KernelSHAP | 0.707 | 0.649 | 0.975 |
| | 10000 | Matrix-Vector | 0.762 | 0.723 | 0.655 |
| | | Least Squares | 0.762 | 0.717 | 0.753 |
| | | KernelSHAP | 0.762 | 0.710 | 0.992 |
| | 100000 | Matrix-Vector | 0.756 | 0.715 | 0.674 |
| | | Least Squares | 0.758 | 0.719 | 0.767 |
| | | KernelSHAP | 0.762 | 0.714 | 0.996 |
| CIFAR10 | 500 | Matrix-Vector | 0.581 | 0.548 | 0.008 |
| | | Least Squares | 0.604 | 0.535 | 0.033 |
| | | KernelSHAP | 0.607 | 0.610 | 0.026 |
| | 1000 | Matrix-Vector | 0.560 | 0.523 | 0.020 |
| | | Least Squares | 0.584 | 0.541 | 0.061 |
| | | KernelSHAP | 0.613 | 0.612 | 0.028 |
| | 10000 | Matrix-Vector | 0.544 | 0.510 | 0.069 |
| | | Least Squares | 0.661 | 0.571 | 0.185 |
| | | KernelSHAP | 0.629 | 0.555 | 0.184 |
| | 100000 | Matrix-Vector | 0.521 | 0.485 | 0.239 |
| | | Least Squares | 0.679 | 0.558 | 0.662 |
| | | KernelSHAP | 0.685 | 0.557 | 0.540 |

**Table 7:** AUC and Rank Correlation for MNIST and CIFAR10 Datasets Using Different Methods ($\ell_2$-squared estimator, without replacement, paired sampling with $b_0$).

