# OpenReview forum: "A Unified Framework for Provably Efficient Algorithms to Estimate Shapley Values"
_NeurIPS.cc/2025/Conference — NeurIPS 2025 poster_

### Official Review · Reviewer_TpwW · 2025-06-29

**Clarity:** 4
**Significance:** 3
**Originality:** 3
**Rating:** 5
**Confidence:** 3

**Summary:**

This paper introduces a unified framework for estimating Shapley values. The authors start from the Shapley value computation as a constrained regression problem, and then reformulate it into an unconstrained linear regression problem. This reformulation, when solved with additional randomized sketching, allows them to encompass several existing estimators, including KernelSHAP, LeverageSHAP, and unbiased KernelSHAP, within a single theoretical framework. Using tools from randomized linear algebra, they derive non-asymptotic sample complexity bounds valid for all estimators in their framework.

In their experiments, they compare a large number of variants (least-squares regression versus matrix-vector operations, different sampling strategies, and tuning parameter $\lambda$) on 8 tabular datasets. They also propose an extension of the algorithm by Musco & Witter, which is more amenable to high-dimension, and allows to perform experiments on MNIST and CIFAR-10.

**Questions:**

* To me, a key differentiating part of the paper lies in the freedom of choice for $\lambda$. Could the authors give intuition about sensible choice? (I am not requesting additional numerical experiments, just hints related to e.g. data distribution or test case specificities)
* Could the authors comment more about adaptive estimators and which type of prior information could be used to design efficient estimation strategies?
* Related to the previous question, I was wondering if it may be possible to use prior information to optimize the proposed bounds with respect to a sampling scheme.

**Ethical Concerns:**

["NO or VERY MINOR ethics concerns only"]

**Final Justification:**

The paper offers a new and very insightful new look at Shapley values estimation, with theoretical results which are promising for future work. The numerical experiments are conducted seriously, are well commented, and illustrates convincingly the author's work.

**Limitations:**

Yes.

**Paper Formatting Concerns:**

No formatting issue.

**Quality:**

3

**Strengths And Weaknesses:**

Even if the main and starting idea of the paper closely follows the one by Musco & Witter, I really enjoy the unification under a single formalism, with the two routes of sketched regression and approximate matrix-vector computation. The novel theoretical guarantee is interesting, in the sense that it paves the way for adaptive strategies. The new approach presented here is thus a promising point of view for improvement in Shapley values.

The paper is well-written and easy to follow (although there are some misspelling errors that should be corrected), with both novel (to the best of my knowledge) theoretical derivations, and very interesting numerical experiments. Among the strengths of the paper are these numerical findings: least-squares regression generally outperform matrix-vector-based ones, the choice of the sampling strategy is in average less influential, and the choice of the newly introduced parameter $\lambda$ seems to be important. In addition, Algorithm 2 designed for high-dimensional problems seems to perform well, and I think it deserves more space inside the main paper.

In my opinion, the only real weakness of the paper lies in the absence of clear prescriptions in practical applications: choice of sampling scheme, and choice of $\lambda$.

---

> ### Author Rebuttal · Authors · 2025-07-30
>
> We thank the reviewer for their time in reviewing the paper and for stating their main concern clearly.
> In the rebuttal below, we will address your questions, while highlighting prescriptions for practical applications.
>
> **Weaknesses.**
>
> **(1) Prescriptions for practical applications.**
>
> Based on our theoretical and experimental results, the following points are of relevance to practical applications:
>
> a. We can use the knowledge of how the error scales with the number of samples (given by our theoretical analysis) to get a fairly good estimate of the error (as elaborated in our response to reviewer TpwW). This in turn can be used to prescribe the number of samples needed for a specified error.
>
> b. In practice, the best estimators tend to be those employing leverage score sampling without replacement and using least squares estimator; however, this remains dataset and model dependent.
>
> **Questions.**
>
> **(1) Choice of $\lambda$.**
>
> Unfortunately, while our analysis reveals that $\lambda$ should be chosen to try to minimize the size of $b_\lambda$ or $P_{U} b_\lambda$, it is currently unclear to us how $\lambda$ should be chosen a priori, particularly in a model agnostic way. Our numerical experiment suggests that $\lambda = \alpha$ (what is used in KernelSHAP and LeverageSHAP) may be better than $\lambda = 0$ (what is used in unbiased KernelSHAP), but we do not view this as a comprehensive experiment.
>
> More broadly, variance reduction techniques, such as the one proposed in the supplementary material and concurrent work like "Regression-adjusted Monte Carlo Estimators for Shapley Values and Probabilistic Values" seem to be worthy of future study. We expect that the best techniques will be model dependent, but that they may substantially improve the performance of  Shapley value estimators.
>
> **(2) Adaptive Estimators /  Optimizing Bounds with Respect to a Sampling Scheme.**
>
> Yes, this is a good idea, and is something we have been working on. As our analysis reveals, the algorithms converge in a predictable way with $m$ (i.e. the convergence is the Monte--Carlo rate $\sim 1/\sqrt{m}$). We have found it works quite well to use the estimate from a larger value of $m$ to approximate the error at some $m_0 \ll m$. As long as $m\gg m_0$, the estimate using $m$ samples is a good proxy for the true solution, relative to the size of the error using $m_0$ samples. We can then estimate the error at step $m$ from the estimate of the error at $m_0$ and the known rate of convergence.
>
> In addition to improving the bounds for a specific distribution, it is possible to use prior information to obtain better/adaptive estimators. Specifically, we can use the previous samples to estimate the vector $b_\lambda$ or $P_{U} b_\lambda$, and adapt our sampling distribution accordingly. The algorithm and analysis for such an approach would be more involved since one would ideally like to reuse the previous samples to estimate the Shapley values. While it may not be possible to improve the worst-case performance with such a strategy, it could potentially lead to better performance in practice.

---

> > ### Comment · Reviewer_TpwW · 2025-08-04
> >
> > Thank you for your detailed answers.
> >
> > Concerning the practical prescriptions, I suggest you add a paragraph somewhere summarizing your answer, in order to guide the readers.
> >
> > Your last point on adaptive estimators is in my opinion an exciting future research direction, with potentially large practical impact!

---

> > > ### Author Response · Authors · 2025-08-04
> > >
> > > We will add a summary to the paper. thanks for the feedback!

---

### Official Review · Reviewer_soJB · 2025-06-30

**Clarity:** 2
**Significance:** 3
**Originality:** 4
**Rating:** 5
**Confidence:** 2

**Summary:**

The paper provides a general framework for analysing Shapley value estimation algorithms, in particular for the context of explainable ML.
The logic of the first part is as follows: (a) the Shapley value is the solution of a linear system of equation, hence can be seen as the solution of a regression (like in SHAP), (b) this regression formulation can be transformed using a whitening/centering change of variable (c) seing the generation of rows as the multiplication by a random matrix, one get two possible reinterpretations of the initial problem:  Sketch Regression and Approximate Vector Multiplication. The second part provides  a sample complexity guarantee for those two interpretations. The  last part is dedicated to experiments on SHAP datasets and on high-dimension datasets, with several sampling schemes being tested.

**Questions:**

* Why does the reformulation of the regression unlocks this unifying framework? Where does it helps?
* what do we learn  from the experimental section?
* It seems the introduction of lambda did not bring anything interesting, why did you choose to keep it?

**Ethical Concerns:**

["NO or VERY MINOR ethics concerns only"]

**Final Justification:**

I will keep my positive score. The authors provided satisfying answers to my questions and suggestions.

**Limitations:**

yes

**Quality:**

3

**Strengths And Weaknesses:**

# Main remarks
+ While the analysis borrows from "Provably Accurate Shapley Value Estimation via Leverage Score Sampling", the results are new to me and not trivial. They unify several schemes into one theorem.

+ Overall, the paper is written with care, and I could answer most of my questions by looking at the appendix.

- I wish some aspects were more detailed, for example, line 127 could have been a full paragraph. Some key tricks/observations that make this work (for example Lemma A5) could be hinted in the main text. The difference between sketch regression and approximate vector multiplication could be better explained.  The value brought by the experimental section should be clarified

- Table 1 is not easy to read (with H being introduced in the legend), and not really commented.  Also, having the kernel definition in the table would help

+ I did not read all the appendix carefully, but what I looked at was sound. In the appendix, the authors mention a correction made to the main text.

# Minor remarks

- I find the formulation of theorem 2.2  too informal (mixing of O, \implies and English)

- Algorithm 1 not readable, I do not know what maxval is

l 70: knowledge of knowing

l 67: seems to be a typo in the combinatorial term...

l 129 whose columns *form*

l 221: typo in the mse formula

paired sampling: explain in a few sentence what it is

l 282: there is an issue with this sentence

l 127: could be a full paragraph

l 265: typo

---

> ### Author Rebuttal · Authors · 2025-07-30
>
> We thank the reviewer for their careful review, highlighting typos, and giving us the opportunity to clarify.
>
> We will fix all the typos pointed out by the reviewer in the camera-ready version of the manuscript. We will also include a more formal statement of Theorem 2.2 and improve the readability of Table 1 and Algorithm 1. We thank the reviewer for spotting the typo in Algorithm 1; maxval should have only be defined in the context of Algorithm 2 (sampling without replacement), which is the maximum number of samples before which we introduce the Poisson approximation for the sampling distribution.
>
> **(1) Proof Sketch.**
>
> In response to the reviewer's suggestion (as well as the suggestion of reviewer 7LtQ), we will now be including a proof sketch in the camera-ready version of our paper. It will be included in the main text if space permits, or if not, at the beginning of Appendix A. We hope this would be helpful in providing an overview of the key ideas used in deriving the main theoretical results of our study.
>
> **(2) Regression Reformulation.**
>
> The main value in the regression formulation to our paper, is that it turns Shapley value computation into a standard linear algebra problem. This is what enables the efficient implementation of KernelSHAP in the SHAP library. In the present work, it also opens a path that allows us to apply existing theory from randomized numerical linear algebra in order to analyze existing methods like KernelSHAP and unbiased KernelSHAP.
>
> **(3) Experimental Section.**
>
> The experimental section showcases the ability to estimate Shapley values in high dimensions and it compares the different estimators used in practice. For the datasets and models that we tested, we find that the performance of different methods are comparable, with a slight advantage noticed towards the leverage scores sampling, without replacement, least squares estimator.
>
> **(4) Benefits of $\lambda$.**
> The main reason we have introduced $\lambda$ is so that our framework encompasses key past estimators like KernelSHAP, LeverageSHAP, and unbiased KernelSHAP. However, it does raise some questions about the "best choice" of $\lambda$, which we discuss in our response to Reviewer TpwW.

---

> > ### Comment · Reviewer_soJB · 2025-08-04
> >
> > Thank you for your answers, I will keep my (positive) score.

---

### Official Review · Reviewer_7LtQ · 2025-07-02

**Clarity:** 3
**Significance:** 3
**Originality:** 3
**Rating:** 4
**Confidence:** 3

**Summary:**

This work provides the first theoretical guarantee for popular estimators such as KernelSHAP in the context of Shapley value estimation. In particular, the authors provide a unified framework that encompasses many existing randomized estimators for Shapley values and prove the corresponding sample-complexity guarantees.

**Questions:**

1. I understand that due to the space constraint, lots of the details are hidden in the Appendix. However, it'll be beneficial to have a proof sketch on the main results (Theorem 2.2) for readers to grasp the high-level idea and how the presented framework helps the final proof.
2. In the discussion around Table 1, it states that the bounds for modified row-norm sampling are no worse than leverage scores and kernel weights in the worst case, but may be almost a factor of $\sqrt{d}$ better in some cases. However, in the experiments, this phenomenon does not seem to show up in practice. It'll be helpful to elaborate on this (e.g., under what scenarios will one be better/worse than others) in the next iteration.
3. Some experiments on other models can be included to go beyond tree-based models.

**Ethical Concerns:**

["NO or VERY MINOR ethics concerns only"]

**Final Justification:**

All my concerns are addressed by the authors' rebuttal, and I maintain my score since my concerns are all minor.

**Limitations:**

yes

**Quality:**

3

**Strengths And Weaknesses:**

**Strengths**
1. The scope and the research question are clearly stated and well-motivated.
2. The writing is rigorous and clear, with careful discussion about how the present work is different from the past literature.
3. The contribution is sound: the first provable non-asymptotic sample complexity bounds for popular Shapley value estimators. Moreover, the general framework might be of independent interest and can provide guidance on design to other estimators.

**Weaknesses**
1. As also pointed out by the authors, the sample complexity bounds depend on instance-specific norms, which are hard to compute in practice.
2. Experiments are mostly limited to XGBoost. Some generalization to neural networks or other black-box models is not explored, even qualitatively.

**Writing**
Only small comments on typos:
1. On line 119, "In Section 3, we extensive experimental evaluation..." is there a verb (e.g., conduct) missing?
2. On line 129, it seems like "whose columns *for* an" should be "whose columns *form* an"
3. On line 176, "$k(S)$.This" lacks a space.

---

> ### Author Rebuttal · Authors · 2025-07-30
>
> We thank the reviewer for carefully reviewing our paper and for also highlighting typos. We will fix the typos that the reviewer has pointed out in the camera-ready version. We address the questions/comments of the reviewer below.
>
> **Questions.**
>
> **(1) Proof Sketch.**
>
> We agree with the reviewer that it would be nice to include a proof sketch in order to present a high-level picture of how our sample complexity bounds are derived. We will include such a proof sketch in the camera-ready version of our manuscript. It will be included in the main text if space permits, or if not, at the beginning of Appendix A.
>
> **(2) Leverage Scores and Modified Row-Norm Sampling Comparison.**
>
> As the reviewer pointed out, modified row-norm sampling performs almost as good as leverage scores in the worst case, while they can perform better a factor of $\sqrt{d}$ in some cases. Whether this advantage in the performance is seen in practice depends on the model at hand.
>
> For the models that we tested in the experiments, we generally find that leverage scores do slightly better than modified row-norm sampling. On the other hand, to show that we can, in principle, obtain a large advantage for modified row-norm sampling, we have constructed a toy model in Appendix E, where we explicitly demonstrate a factor of $\sqrt{d}$ advantage of modified row-norm sampling over leverage scores.
>
> The key idea used for constructing the model in Appendix E is that modified row-norm sampling and leverage scores place different importance on subsets $S \subset [d]$ of features, and is determined solely by the size of the subsets. Specifically, the weights on subsets of small and large size (compared to $d$) is smaller for the modified row-norm sampling distribution as compared to leverage scores (see Theorem A.6). If we have models for which the entries of the vector $b_\lambda$ or $P_{U} b_\lambda$ corresponding to subsets of moderate size is small, we can expect to see an advantage of modified row-norm sampling over leverage scores. As to whether there are models that are used in practice (such as neural networks) where such a condition holds, and consequently, gives an advantage for modified row-norm sampling over leverage scores, needs to be studied in more detail in the future. We note that a similar discussion holds for comparison of kernel weights with leverage scores, and can be found in Appendix E. We can elaborate on these details in the camera-ready version of the manuscript, and also highlight the open question of whether an advantage of modified row-norm sampling over leverage scores can be obtained for models used in practice.
>
> **(3) Experiments Beyond Tree Models.**
>
> In the present version, estimates are limited to XGBoost since it is possible to compute exactly the true Shapley values for the model using TreeSHAP algorithm.
>
> It is possible to compute results for neural networks too, but only when the true Shapley values can be computed efficiently ($d\approx 15$ and smaller). Below, we present experimental results for a neural network trained on the Adult dataset.
> We measure the error for both least squares and matrix-vector estimators, and samples drawn with and without replacement using the leverage score distribution.
>
> | Num Samples | With Replacement | With Replacement | Without Replacement | Without Replacement |
> |-------------|------------------|------------------|---------------------|---------------------|
> |             | Least Squares    | Matrix Vector    | Least Squares       | Matrix Vector       |
> | 410         | 0.099            | 0.57             | 0.097               | 2.8                 |
> | 1229        | 0.018            | 0.13             | 0.014               | 0.8                 |
> | 2458        | 0.0099           | 0.083            | 0.011               | 0.36                |
> | 3696        | 0.01             | 0.057            | 0.0079              | 0.14                |
>
> We find the results to be consistent with our experiments on trees, with least squares approximation generally performing better than matrix vector estimator.

---

> > ### Comment · Reviewer_7LtQ · 2025-08-01
> >
> > Thanks for the detailed response, and indeed, the discussion around (2) is insightful. I have no further questions. Great work by the way : ).

---

### Official Review · Reviewer_Z2FN · 2025-07-03

**Clarity:** 4
**Significance:** 3
**Originality:** 4
**Rating:** 4
**Confidence:** 3

**Summary:**

The paper presents a new, unified framework for analyzing and developing randomized algorithms to estimate Shapley values, which are crucial for explaining the outputs of machine learning models. The exact computation of Shapley values is often too computationally expensive, necessitating the use of estimation methods. The contributions include a framework that encompasses many existing randomized estimators for Shapley values, theoretical guarantees for the popular KernelSHAP method, and implementation improvements that enhance the scalability of Shapley value estimation to high-dimensional data.

**Questions:**

1. One question for the author concerns the implications of the proposed method for Shapley value-based data valuation.
2. Can you provide some intuition how sampling with replacement works better than without replacement in certain settings and worse in others?
3. Figure 3 2(a) and (b) remove the top 100 features altogether, and as a result, there are only two data points for each estimator. Can you provide results for individuals one by one?

**Ethical Concerns:**

["NO or VERY MINOR ethics concerns only"]

**Final Justification:**

The paper is a novel contribution to the space of Shapley value estimation. I am leaning toward accepting the paper.

**Limitations:**

Yes

**Quality:**

4

**Strengths And Weaknesses:**

Strengths:

- The presentation is clear and flows well.
- The related work is carefully discussed.
- The idea of using randomized sketching to unify different Shapley estimators is novel.

Weakness:
- The improvement of the proposed estimators measured in terms of AUC deletion is marginal.

---

> ### Author Rebuttal · Authors · 2025-07-30
>
> We thank the reviewer for their time in carefully reviewing our paper.
> We answer the reviewer's questions below:
>
> **(1) Implications.**
>
> The KernelSHAP method is widely used to approximate Shapley values (e.g. the SHAP library has 24k stars on Github).
> However, until our work, there were no strong theoretical bounds guaranteeing the accuracy of KernelSHAP. While the method was observed to work well in practice, the lack of theoretical guarantees represented a major gap in the explainability pipeline. By providing theoretical guarantees for the method, our work  further justifies the use of such libraries for providing model explainability in critical fields.
>
> Moreover, by approximating values of $||b_\lambda||$ and $||Ub_\lambda||$, it is  possible to estimate the number of samples that will provide a specific $\epsilon, \delta$ error bound. In future work, these bounds could be made precise by analyzing the behavior of $||b_\lambda||$ as the number of samples varies. This also opens up the possibility of adaptively choosing the distribution based on the samples that we have seen so far so as to improve the performance (which was also highlighted by reviewer TpwW), and can be explored in future work.
>
> **(2) Comparison Between Estimator Performance (sampling with/without replacement).**
>
> Based on our theoretical analysis for sampling with and without replacement in the unpaired case, we expect sampling without replacement to perform as well as or better than sampling with replacement. Experimentally, for the case of paired sampling (where we always sample a subset along with its complement), we find that the sampling with and without replacement perform very similarly for the least squares estimator. However, sampling with replacement outperforms sampling without replacement for matrix multiplication estimator in the paired case: we leave the problem developing bounds for the paired case for future research.
>
> **(3) Fine-Grained Results.**
>
> We believe the information that the reviewer is looking for can be found in Appendix  H (Figures 6 and 7). Indeed, since these are Shapley estimators, we do not expect the curves to vary dramatically; we verify that indeed they converge to a single curve. In this sense, sample efficiency positively impacts faithfulness.

---

### Decision · Program_Chairs · 2025-09-17

**Decision:**

Accept (poster)

**Comment:**

Exploiting the fact that the Shapley value is the solution of a (large) linear regression problem, the authors applied matrix sketching techniques to unify many existing estimation algorithms for the Shapley value. In particular, the authors provided the first theoretical justification for the popular KernelSHAP method and proposed a new variant that mixes KernelSHAP and LeverageSHAP. Numerical experiments were conducted to verify the theoretical results.

The reviewers found the unifying framework, the theoretical justifications and the numerical insights interesting and significant. The reviewers also appreciated the writing and presentation of the paper (barring some minor corrections). The authors' response helped clarify most of the reviewers' questions and resulted in a uniformly positive recommendation, which I concur. This work adds significant novel insights into the existing literature on Shapley value estimation, and should be interesting to the community and possibly inspire further developments.

In the final version, please incorporate the reviewers' suggestions and some of the details in the rebuttal.